# No-Regret Online Reinforcement Learning with Adversarial Losses and Transitions

**Tiancheng Jin** *
University of Southern California
tiancheng.jin@usc.edu

**Junyan Liu** *
University of California, San Diego
jul037@ucsd.edu

**Chloé Rouyer**
University of Copenhagen
chloe@di.ku.dk

**William Chang**
University of California, Los Angeles
chang314@g.ucla.edu

**Chen-Yu Wei**
MIT Institute for Data, Systems, and Society
chenyuw@mit.edu

**Haipeng Luo**
University of Southern California
haipengl@usc.edu

## Abstract

Existing online learning algorithms for adversarial Markov Decision Processes achieve $\mathcal{O}(\sqrt{T})$ regret after $T$ rounds of interactions even if the loss functions are chosen arbitrarily by an adversary, with the caveat that the transition function has to be fixed. This is because it has been shown that adversarial transition functions make no-regret learning impossible. Despite such impossibility results, in this work, we develop algorithms that can handle both adversarial losses and adversarial transitions, with regret increasing smoothly in the degree of maliciousness of the adversary. More concretely, we first propose an algorithm that enjoys $\widetilde{\mathcal{O}}(\sqrt{T} + C^{\mathsf{P}})$ regret where $C^{\mathsf{P}}$ measures how adversarial the transition functions are and can be at most $\mathcal{O}(T)$. While this algorithm itself requires knowledge of $C^{\mathsf{P}}$, we further develop a black-box reduction approach that removes this requirement. Moreover, we also show that further refinements of the algorithm not only maintains the same regret bound, but also simultaneously adapts to easier environments (where losses are generated in a certain stochastically constrained manner as in Jin et al. [2021]) and achieves $\widetilde{\mathcal{O}}(U + \sqrt{UC^{\mathsf{L}}} + C^{\mathsf{P}})$ regret, where $U$ is some standard gap-dependent coefficient and $C^{\mathsf{L}}$ is the amount of corruption on losses.

## 1  Introduction

Markov Decision Processes (MDPs) are widely-used models for reinforcement learning. They are typically studied under a fixed transition function and a fixed loss function, which fails to capture scenarios where the environment is time-evolving or susceptible to adversarial corruption. This motivates many recent studies to overcome these challenges. In particular, one line of research, originated from Even-Dar et al. [2009] and later improved or generalized by e.g. Neu et al. [2010a,b], Zimin and Neu [2013], Rosenberg and Mansour [2019a], Jin et al. [2020], takes inspiration from the online learning literature and considers interacting with a sequence of $T$ MDPs, each with an adversarially chosen loss function. Despite facing such a challenging environment, the learner can still ensure $\mathcal{O}(\sqrt{T})$ regret (ignoring other dependence; same below) as shown by these works, that is, the learner's average performance is close to that of the best fixed policy up to $\mathcal{O}(1/\sqrt{T})$.

---

*Equal contribution.

37th Conference on Neural Information Processing Systems (NeurIPS 2023).

However, one caveat of these studies is that they all still require the MDPs to have to the same transition function. This is not for no reason — Abbasi Yadkori et al. [2013] shows that even with full information feedback, achieving sub-linear regret with adversarial transition functions is computationally hard, and Tian et al. [2021] complements this result by showing that under the more challenging bandit feedback, this goal becomes even information-theoretically impossible without paying exponential dependence on the episode length.

To get around such impossibility results, one natural idea is to allow the regret to depend on some measure of maliciousness $C^{\mathsf{P}}$ of the transition functions, which is $0$ when the transitions remain the same over time and $\mathcal{O}(T)$ in the worst case when they are completely arbitrary. We review several such attempts at the end of this section, and point out here that they all suffer one issue: even when $C^{\mathsf{P}} = 0$, the algorithms developed in these works all suffer *linear regret* when the loss functions are completely arbitrary, while, as mentioned above, $\mathcal{O}(\sqrt{T})$ regret is achievable in this case. This begs the question: *when learning with completely adversarial MDPs, is $\mathcal{O}(\sqrt{T} + C^{\mathsf{P}})$ regret achievable?*

In this work, we not only answer this question affirmatively, but also show that one can perform even better sometimes. More concretely, our results are as follows.

1. In Section 3, we develop a variant of the UOB-REPS algorithm [Jin et al., 2020], achieving $\widetilde{\mathcal{O}}(\sqrt{T} + C^{\mathsf{P}})$ regret in completely adversarial environments when $C^{\mathsf{P}}$ is known. The algorithmic modifications we propose include an enlarged confidence set, using the log-barrier regularizer, and a novel *amortized* bonus term that leads to a critical "change of measure" effect in the analysis.

2. We then remove the requirement on the knowledge of $C^{\mathsf{P}}$ in Section 4 by proposing a *black-box reduction* that turns any algorithm with $\widetilde{\mathcal{O}}(\sqrt{T} + C^{\mathsf{P}})$ regret under known $C^{\mathsf{P}}$ into another algorithm with the same guarantee (up to logarithmic factors) even if $C^{\mathsf{P}}$ is unknown. Our reduction improves that of Wei et al. [2022] by allowing adversarial losses, which presents extra challenges as discussed in Pacchiano et al. [2022]. The idea of our reduction builds on top of previous adversarial model selection framework (a.k.a. Corral [Agarwal et al., 2017, Foster et al., 2020, Luo et al., 2022]), but is even more general and is of independent interest: it shows that the requirement from previous work on having a *stable* input algorithm is actually redundant, since our method can turn *any* algorithm into a stable one.

3. Finally, in Section 5 we also further refine our algorithm so that it simultaneously adapts to the maliciousness of the loss functions and achieves $\widetilde{\mathcal{O}}(\min\{\sqrt{T}, U + \sqrt{UC^{\mathsf{L}}}\} + C^{\mathsf{P}})$ regret, where $U$ is some standard gap-dependent coefficient and $C^{\mathsf{L}} \leq T$ is the amount of corruption on losses (this result unfortunately requires the knowledge of $C^{\mathsf{P}}$, but not $U$ or $C^{\mathsf{L}}$). This generalizes the so-called *best-of-both-worlds* guarantee of Jin and Luo [2020], Jin et al. [2021], Dann et al. [2023] from $C^{\mathsf{P}} = 0$ to any $C^{\mathsf{P}}$, and is achieved by combining the ideas from Jin et al. [2021] and Ito [2021] with a novel *optimistic transition* technique. In fact, this technique also leads to improvement on the dependence of episode length even when $C^{\mathsf{P}} = 0$.

**Related Work** Here, we review how existing studies deal with adversarially chosen transition functions and how our results compare to theirs. The closest line of research is usually known as *corruption robust* reinforcement learning [Lykouris et al., 2019, Chen et al., 2021, Zhang et al., 2021, Wei et al., 2022], which assumes a ground truth MDP and measures the maliciousness of the adversary via the amount of corruption to the ground truth — the amount of corruption is essentially our $C^{\mathsf{L}}$, while the amount of corruption is essentially our $C^{\mathsf{P}}$ (these will become clear after we provide their formal definitions later). Naturally, the regret in these works is defined as the difference between the learner's total loss and that of the best policy *with respect to the ground truth MDP*, in which case $\widetilde{\mathcal{O}}(\sqrt{T} + C^{\mathsf{P}} + C^{\mathsf{L}})$ regret is unavoidable and is achieved by the state-of-the-art [Wei et al., 2022].

On the other hand, following the canonical definition in online learning, we define regret *with respect to the corrupted MDPs*, in which case $\widetilde{\mathcal{O}}(\sqrt{T} + C^{\mathsf{P}})$ is achievable as we show (regardless how large $C^{\mathsf{L}}$ is). To compare these results, note that the two regret definitions differ from each other by an amount of at most $\mathcal{O}(C^{\mathsf{P}} + C^{\mathsf{L}})$. Therefore, *our result implies that of Wei et al. [2022], but not vice versa* — what Wei et al. [2022] achieves in our definition of regret is again $\widetilde{\mathcal{O}}(\sqrt{T} + C^{\mathsf{P}} + C^{\mathsf{L}})$, which is never better than ours and could be $\Omega(T)$ even when $C^{\mathsf{P}} = 0$.

In fact, our result also improves upon that of Wei et al. [2022] in terms of the gap-dependent refinement — their refined bound is $\widetilde{\mathcal{O}}(\min\{\sqrt{T}, G\} + C^{\mathsf{P}} + C^{\mathsf{L}})$ for some gap-dependent measure

$G$ that is known to be no less than our gap-dependent measure $U$; on the other hand, based on earlier discussion, our refined bound *in their regret definition* is $\widetilde{\mathcal{O}}(\min\{\sqrt{T}, U + \sqrt{UC^{\mathsf{L}}}\} + C^{\mathsf{P}} + C^{\mathsf{L}}) = \widetilde{\mathcal{O}}(\min\{\sqrt{T}, U\} + C^{\mathsf{P}} + C^{\mathsf{L}})$ and thus better. The caveat is that, as mentioned, for this refinement our result requires the knowledge of $C^{\mathsf{P}}$, but Wei et al. [2022] does not.[2] However, we emphasize again that for the gap-independent bound, our result does not require knowledge of $C^{\mathsf{P}}$ and is achieved via an even more general black-box reduction compared to the reduction of Wei et al. [2022].

Finally, we mention that another line of research, usually known as *non-stationary reinforcement learning*, also allows arbitrary transition/loss functions and measures the difficulty by either the number of changes in the environment [Auer et al., 2008, Gajane et al., 2018] or some smoother measure such as the total variation across time [Wei and Luo, 2021, Cheung et al., 2023]. These results are less comparable to ours since their regret (known as dynamic regret) measures the performance of the learner against the best *sequence* of policies, while ours (known as static regret) measures the performance against the best *fixed* policy.

## 2 Preliminaries

We consider the problem of sequentially learning $T$ episodic MDPs, all with the same state space $S$ and action space $A$. We assume without loss of generality (similarly to Jin et al. [2021]) that the state space $S$ has a layered structure and is partitioned into $L + 1$ subsets $S_0, \ldots, S_L$ such that $S_0$ and $S_L$ only contain the initial state $s_0$ and the terminal state $s_L$ respectively, and transitions are only possible between consecutive layers. For notational convenience, for any $k < L$, we denote the set of tuples $S_k \times A \times S_{k+1}$ by $W_k$. We also denote by $k(s)$ the layer to which state $s \in S$ belongs.

Ahead of time, knowing the learner's algorithm, the environment decides the transition functions $\{P_t\}_{t=1}^T$ and the loss functions $\{\ell_t\}_{t=1}^T$ for these $T$ MDPs in an arbitrary manner (unknown to the learner). Then the learner sequentially interacts with these $T$ MDPs: for each episode $t = 1, \ldots, T$, the learner first decides a stochastic policy $\pi_t : S \times A \to [0, 1]$ where $\pi_t(a|s)$ is the probability of taking action $a$ when visiting state $s$; then, starting from the initial state $s_{t,0} = s_0$, for each $k = 0, \ldots, L-1$, the learner repeatedly selects an action $a_{t,k}$ sampled from $\pi_t(\cdot|s_{t,k})$, suffers loss $\ell_t(s_{t,k}, a_{t,k}) \in [0, 1]$, and transits to the next state $s_{t,k+1}$ sampled from $P_t(\cdot|s_{t,k}, a_{t,k})$ (until reaching the terminal state); finally, the learner observes the losses of those visited state-action pairs (a.k.a. bandit feedback).

Let $\ell_t(\pi) = \mathbb{E}\big[\sum_{h=0}^{L-1} \ell_t(s_h, a_h)|P_t, \pi\big]$ be the expected loss of executing policy $\pi$ in the $t$-th MDP (that is, $\{(s_h, a_h)\}_{h=0}^{L-1}$ is a stochastic trajectory generated according to transition $P_t$ and policy $\pi$). Then, the regret of the learner against any policy $\pi$ is defined as $\text{Reg}_T(\pi) = \mathbb{E}\big[\sum_{t=1}^T \ell_t(\pi_t) - \ell_t(\pi)\big]$. We denote by $\mathring{\pi}$ one of the optimal policies in hindsight such that $\text{Reg}_T(\mathring{\pi}) = \max_\pi \text{Reg}_T(\pi)$ and use $\text{Reg}_T \triangleq \text{Reg}_T(\mathring{\pi})$ as a shorthand.

**Maliciousness Measure of the Transitions** If the transition functions are all the same, Jin et al. [2020] shows that $\text{Reg}_T = \widetilde{\mathcal{O}}(L|S|\sqrt{|A|T})$ is achievable, no matter how the loss functions are decided. However, when the transition functions are also arbitrary, Tian et al. [2021] shows that $\text{Reg}_T = \Omega(\min\{T, \sqrt{2^L T}\})$ is unavoidable. Therefore, a natural goal is to allow the regret to smoothly increase from order $\sqrt{T}$ to $T$ when some maliciousness measure of the transitions increases. Specifically, the measure we use is

$$C^{\mathsf{P}} \triangleq \min_{P' \in \mathcal{P}} \sum_{t=1}^T \sum_{k=0}^{L-1} \max_{(s,a) \in S_k \times A} \|P_t(\cdot|s, a) - P'(\cdot|s, a)\|_1, \tag{1}$$

where $\mathcal{P}$ denotes the set of all valid transition functions. Let $P$ be the transition that realizes the minimum in this definition. Then $C^{\mathsf{P}}$ can be regarded as the same *corruption* measure used in Chen et al. [2021]; there, it is assumed that a ground truth MDP with transition $P$ exists, and the adversary corrupts it arbitrarily in each episode to obtain $P_t$, making $C^{\mathsf{P}}$ the total amount of corruption measured in a certain norm. For simplicity, in the rest of this paper, we will also take this perspective and call

---

[2]When $C^{\mathsf{P}} + C^{\mathsf{L}}$ is known, Lykouris et al. [2019] also achieves $\widetilde{\mathcal{O}}(\min\{\sqrt{T}, U\} + C^{\mathsf{P}} + C^{\mathsf{L}})$ regret, but similar to earlier discussions on gap-independent bounds, their regret definition is weaker than ours.

$C^{\mathsf{P}}$ the transition corruption. We also use $C_t^{\mathsf{P}} = \sum_{k=0}^{L-1} \max_{(s,a) \in S_k \times A} \|P_t(\cdot|s,a) - P(\cdot|s,a)\|_1$ to denote the per-round corruption (so $C^{\mathsf{P}} = \sum_{t=1}^{T} C_t^{\mathsf{P}}$). It is clear that $C^{\mathsf{P}} = 0$ when the transition stays the same for all MDPs, while in the worst case it is at most $2TL$. Our goal is to achieve $\mathrm{Reg}_T = \mathcal{O}(\sqrt{T} + C^{\mathsf{P}})$ (ignoring other dependence), which smoothly interpolates between the result of Jin et al. [2020] for $C^{\mathsf{P}} = 0$ and that of Tian et al. [2021] for $C^{\mathsf{P}} = \mathcal{O}(T)$.

**Enlarged Confidence Set**  A central technique to deal with unknown transitions is to maintain a shrinking confidence set that contains the ground truth with high probability [Rosenberg and Mansour, 2019a,b, Jin et al., 2020]. With a properly enlarged confidence set, the same idea extends to the case with adversarial transitions [Lykouris et al., 2019]. Specifically, all our algorithms deploy the following transition estimation procedure. It proceeds in epochs, indexed by $i = 1, 2, \cdots$, and each epoch $i$ includes some consecutive episodes. An epoch ends whenever we encounter a state-action pair whose total number of visits doubles itself when compared to the beginning of that epoch. At the beginning of each epoch $i$, we calculate an empirical transition $\bar{P}_i$ as:

$$\bar{P}_i(s'|s,a) = m_i(s,a,s')/m_i(s,a), \quad \forall(s,a,s') \in W_k,\ k = 0,\ldots L-1, \tag{2}$$

where $m_i(s,a)$ and $m_i(s,a,s')$ are the total number of visits to $(s,a)$ and $(s,a,s')$ prior to epoch $i$.[3] In addition, we calculate the following transition confidence set.

**Definition 2.1.** *(Confidence Set of Transition Functions) Let $\delta \in (0,1)$ be a confidence parameter. With known corruption $C^{\mathsf{P}}$, we define the confidence set of transition functions for epoch $i$ as*

$$\mathcal{P}_i = \Big\{ P \in \mathcal{P} : \big|P(s'|s,a) - \bar{P}_i(s'|s,a)\big| \le B_i(s,a,s'),\ \forall(s,a,s') \in W_k, k = 0,\ldots,L-1 \Big\}, \tag{3}$$

*where the confidence interval $B_i(s,a,s')$ is defined, with $\iota = |S||A|T/\delta$ as*

$$B_i(s,a,s') = \min\left\{ 1, 16\sqrt{\frac{\bar{P}_i(s'|s,a)\log(\iota)}{m_i(s,a)}} + 64 \cdot \frac{C^{\mathsf{P}} + \log(\iota)}{m_i(s,a)} \right\}. \tag{4}$$

Note that the confidence interval is enlarged according to how large the corruption $C^{\mathsf{P}}$ is. We denote by $\mathcal{E}_{\mathrm{CON}}$ the event that $P \in \mathcal{P}_i$ for all epoch $i$, which is guaranteed to happen with high-probability.

**Lemma 2.2.** *With probability at least $1 - 2\delta$, the event $\mathcal{E}_{\mathrm{CON}}$ holds.*

**Occupancy Measure and Upper Occupancy Bound**  Similar to previous work [Rosenberg and Mansour, 2019a, Jin et al., 2020], given a stochastic policy $\pi$ and a transition function $\bar{P}$, we define the occupancy measure $q^{\bar{P},\pi}$ as the mapping from $S \times A \times S$ to $[0,1]$ such that $q^{\bar{P},\pi}(s,a,s')$ is the probability of visiting $(s,a,s')$ when executing policy $\pi$ in an MDP with transition $\bar{P}$. Further define $q^{\bar{P},\pi}(s,a) = \sum_{s' \in S} q^{\bar{P},\pi}(s,a,s')$ (the probability of visiting $(s,a)$) and $q^{\bar{P},\pi}(s) = \sum_{a \in A} q^{\bar{P},\pi}(s,a)$ (the probability of visiting $s$). Given an occupancy measure $q$, the corresponding policy that defines it, denoted by $\pi^q$, can be extracted via $\pi^q(a|s) \propto q(s,a)$.

Importantly, the expected loss $\ell_t(\pi)$ defined earlier equals $\langle q^{P_t,\pi}, \ell_t \rangle$, making our problem a variant of online linear optimization and enabling the usage of standard algorithmic frameworks such as Online Mirror Descent (OMD) or Follow-the-Regularized-Leader (FTRL). These frameworks operate over a set of occupancy measures in the form of either $\Omega(\bar{P}) = \{q^{\bar{P},\pi} : \pi \text{ is a stochastic policy}\}$ for some transition function $\bar{P}$, or $\Omega(\bar{\mathcal{P}}) = \{q^{\bar{P},\pi} : \bar{P} \in \bar{\mathcal{P}}, \pi \text{ is a stochastic policy}\}$ for some set of transition functions $\bar{\mathcal{P}}$.

Following Jin et al. [2020], to handle partial feedback on the loss function $\ell_t$, we need to construct loss estimators $\widehat{\ell}_t$ using the (efficiently computable) *upper occupancy bound* $u_t$:

$$\widehat{\ell}_t(s,a) = \frac{\mathbb{I}_t(s,a)\ell_t(s,a)}{u_t(s,a)}, \text{ where } u_t(s,a) = \max_{\widehat{P} \in \mathcal{P}_{i(t)}} q^{\widehat{P},\pi_t}(s,a), \tag{5}$$

$\mathbb{I}_t(s,a)$ is 1 if $(s,a)$ is visited during episode $t$ (so that $\ell_t(s,a)$ is revealed), and 0 otherwise, and $i(t)$ denotes the epoch index to which episode $t$ belongs. We also define $u_t(s) = \sum_{a \in A} u_t(s,a)$.

---

[3]When $m_i(s,a) = 0$, we simply let the transition function to be uniform, that is, $\bar{P}_i(s'|s,a) = 1/|S_{k(s')}|$.

# 3  Achieving $\mathcal{O}(\sqrt{T} + C^{\mathsf{P}})$ with Known $C^{\mathsf{P}}$

As the first step, we develop an algorithm that achieves our goal when $C^{\mathsf{P}}$ is known. To introduce our solution, we first briefly review the UOB-REPS algorithm of Jin et al. [2020] (designed for $C^{\mathsf{P}} = 0$) and point out why simply using the enlarged confidence set Eq. (3) when $C^{\mathsf{P}} \neq 0$ is far away from solving the problem. Specifically, UOB-REPS maintains a sequence of occupancy measures $\{\widehat{q}_t\}_{t=1}^T$ via OMD: $\widehat{q}_{t+1} = \operatorname{argmin}_{q \in \Omega(\mathcal{P}_{i(t+1)})} \eta \langle q, \widehat{\ell}_t \rangle + D_\phi(q, \widehat{q}_t)$. Here, $\eta > 0$ is a learning rate, $\widehat{\ell}_t$ is the loss estimator defined in Eq. (5), $\phi$ is the negative entropy regularizer, and $D_\phi$ is the corresponding Bregman divergence.[4] With $\widehat{q}_t$ at hand, in episode $t$, the learner simply executes $\pi_t = \pi^{\widehat{q}_t}$. Standard analysis of OMD ensures a bound on the estimated regret $\text{REG} = \mathbb{E}[\sum_t \langle \widehat{q}_t - q^{P,\mathring{\pi}}, \widehat{\ell}_t \rangle]$, and the rest of the analysis of Jin et al. [2020] boils down to bounding the difference between $\text{REG}$ and $\text{Reg}_T$.

**First Issue**  This difference between $\text{REG}$ and $\text{Reg}_T$ leads to the first issue when one tries to analyze UOB-REPS against adversarial transitions — it contains the following bias term that measures the difference between the optimal policy's estimated loss and its true loss:

$$\mathbb{E}\left[\sum_{t=1}^T \left\langle q^{P,\mathring{\pi}}, \widehat{\ell}_t - \ell_t \right\rangle\right] = \mathbb{E}\left[\sum_{t=1}^T \sum_{s,a} q^{P,\mathring{\pi}}(s,a)\ell_t(s,a)\left(\frac{q^{P_t,\pi_t}(s,a) - u_t(s,a)}{u_t(s,a)}\right)\right]. \quad (6)$$

When $C^{\mathsf{P}} = 0$, we have $P = P_t$, and thus under the high probability event $\mathcal{E}_{\text{CON}}$ and by the definition of upper occupancy bound, we know $q^{P_t,\pi_t}(s,a) \leq u_t(s,a)$, making Eq. (6) negligible. However, this argument breaks when $C^{\mathsf{P}} \neq 0$ and $P \neq P_t$. In fact, $P_t$ can be highly different from any transitions in $\mathcal{P}_{i(t)}$ with respect to which $u_t$ is defined, making Eq. (6) potentially huge.

**Solution: Change of Measure via Amortized Bonuses**  Given that $q^{P,\pi_t}(s,a) \leq u_t(s,a)$ does still hold with high probability, Eq. (6) is (approximately) bounded by

$$\mathbb{E}\left[\sum_{t=1}^T \sum_{s,a} q^{P,\mathring{\pi}}(s,a)\frac{|q^{P_t,\pi_t}(s,a) - q^{P,\pi_t}(s,a)|}{u_t(s,a)}\right] = \mathbb{E}\left[\sum_{t=1}^T \sum_s q^{P,\mathring{\pi}}(s)\frac{|q^{P_t,\pi_t}(s) - q^{P,\pi_t}(s)|}{u_t(s)}\right]$$

which is at most $\mathbb{E}\left[\sum_{t=1}^T \sum_s q^{P,\mathring{\pi}}(s)\frac{C_t^{\mathsf{P}}}{u_t(s)}\right]$ since $\left|q^{P_t,\pi_t}(s) - q^{P,\pi_t}(s)\right|$ is bounded by the per-round corruption $C_t^{\mathsf{P}}$ (see Corollary D.3.6). While this quantity is potentially huge, if we could "change the measure" from $q^{P,\mathring{\pi}}$ to $\widehat{q}_t$, then the resulting quantity $\mathbb{E}\left[\sum_{t=1}^T \sum_s \widehat{q}_t(s)\frac{C_t^{\mathsf{P}}}{u_t(s)}\right]$ is at most $|S|C^{\mathsf{P}}$ since $\widehat{q}_t(s) \leq u_t(s)$ by definition. The general idea of such a change of measure has been extensively used in the online learning literature (see Luo et al. [2021] in a most related context) and can be realized by changing the loss fed to OMD from $\widehat{\ell}_t$ to $\widehat{\ell}_t - b_t$ for some bonus term $b_t$, which, in our case, should satisfy $b_t(s,a) \approx \frac{C_t^{\mathsf{P}}}{u_t(s)}$. However, the challenge here is that $C_t^{\mathsf{P}}$ is *unknown*!

Our solution is to introduce a type of efficiently computable *amortized bonuses* that do not change the measure per round, but do so overall. Specifically, our amortized bonus $b_t$ is defined as

$$b_t(s,a) = \begin{cases} \frac{4L}{u_t(s)} & \text{if } \sum_{\tau=1}^t \mathbb{I}\{\lceil \log_2 u_\tau(s)\rceil = \lceil \log_2 u_t(s)\rceil\} \leq \frac{C^{\mathsf{P}}}{2L}, \\ 0 & \text{else,} \end{cases} \quad (7)$$

which we also write as $b_t(s)$ since it is independent of $a$. To understand this definition, note that $-\lceil \log_2 u_t(s)\rceil$ is exactly the unique integer $j$ such that $u_t(s)$ falls into the bin $(2^{-j-1}, 2^{-j}]$. Therefore, the expression $\sum_{\tau=1}^t \mathbb{I}\{\lceil \log_2 u_\tau(s)\rceil = \lceil \log_2 u_t(s)\rceil\}$ counts, among all previous rounds $\tau = 1, \ldots, t$, how many times we have encountered a $u_\tau(s)$ value that falls into the same bin as $u_t(s)$. If this number does not exceed $\frac{C^{\mathsf{P}}}{2L}$, we apply a bonus of $\frac{4L}{u_t(s)}$, which is (two times of) the maximum possible value of the unknown quantity $\frac{C_t^{\mathsf{P}}}{u_t(s)}$; otherwise, we do not apply any bonus. The idea is that by enlarging the bonus to it maximum value and stopping it after enough times, even though each $b_t(s)$ might be quite different from $\frac{C_t^{\mathsf{P}}}{u_t(s)}$, overall they behave similarly after $T$ episodes:

---

[4]The original loss estimator of Jin et al. [2020] is slightly different, but that difference is only for the purpose of obtaining a high probability regret guarantee, which we do not consider in this work for simplicity.

---

**Algorithm 1** Algorithm for Adversarial Transitions (with Known $C^P$)

---

**Input:** confidence parameter $\delta \in (0,1)$, learning rate $\eta > 0$.

**Initialize:** epoch index $i = 1$; counters $m_1(s,a) = m_1(s,a,s') = m_0(s,a) = m_0(s,a,s') = 0$ for all $(s,a,s')$; empirical transition $\bar{P}_1$ and confidence width $B_1$ based on Eq. (2) and Eq. (4); occupancy measure $\widehat{q}_1(s,a,s') = \frac{1}{|S_k||A||S_{k+1}|}$ for all $(s,a,s')$; and initial policy $\pi_1 = \pi^{\widehat{q}_1}$.

**for** $t = 1, \ldots, T$ **do**

    Execute policy $\pi_t$ and obtain trajectory $(s_{t,k}, a_{t,k})$ for $k = 0, \ldots, L-1$.

    Construct loss estimator $\widehat{\ell}_t$ as defined in Eq. (5).

    Update $b_t(s)$ for all $s$ based on Eq. (7).

    Increase counters: for each $k < L$, $m_i(s_{t,k}, a_{t,k}, s_{t,k+1}) \overset{+}{\leftarrow} 1$, $m_i(s_{t,k}, a_{t,k}) \overset{+}{\leftarrow} 1$.

    **if** $\exists k, \; m_i(s_{t,k}, a_{t,k}) \geq \max\{1, 2m_{i-1}(s_{t,k}, a_{t,k})\}$ **then**         ▷ entering a new epoch

        Increase epoch index $i \overset{+}{\leftarrow} 1$.

        Initialize new counters: $\forall (s,a,s')$, $m_i(s,a,s') = m_{i-1}(s,a,s'), m_i(s,a) = m_{i-1}(s,a)$.

        Update confidence set $\mathcal{P}_i$ based on Eq. (3).

    Let $D_\phi(\cdot, \cdot)$ be the Bregman divergence with respect to log barrier (Eq. (11)) and compute

$$\widehat{q}_{t+1} = \operatorname*{argmin}_{q \in \Omega(\mathcal{P}_i)} \eta \left\langle q, \widehat{\ell}_t - b_t \right\rangle + D_\phi(q, \widehat{q}_t). \tag{8}$$

    Update policy $\pi_{t+1} = \pi^{\widehat{q}_{t+1}}$.

---

**Lemma 3.1.** *The amortized bonus defined in* Eq. (7) *satisfies* $\sum_{t=1}^{T} \frac{C_t^P}{u_t(s)} \leq \sum_{t=1}^{T} b_t(s)$ *and* $\sum_{t=1}^{T} \widehat{q}_t(s) b_t(s) = \mathcal{O}(C^P \log T)$ *for any* $s$.

Therefore, the problematic term Eq. (6) is at most $\mathbb{E}[\sum_t \langle q^{P,\mathring{\pi}}, b_t \rangle]$, which, if "converted" to $\mathbb{E}[\sum_t \langle \widehat{q}_t, b_t \rangle]$ (change of measure), is nicely bounded by $\mathcal{O}(|S|C^P \log T)$. As mentioned, such a change of measure can be realized by feeding $\widehat{\ell}_t - b_t$ instead of $\widehat{\ell}_t$ to OMD, because now standard analysis of OMD ensures a bound on $\text{REG} = \mathbb{E}[\sum_t \langle \widehat{q}_t - q^{P,\mathring{\pi}}, \widehat{\ell}_t - b_t \rangle]$, which, compared to the earlier definition of REG, leads to a difference of $\mathbb{E}[\sum_t \langle \widehat{q}_t - q^{P,\mathring{\pi}}, b_t \rangle]$ (see Appendix A.5 for details).

**Second Issue** The second issue comes from analyzing REG (which exists even if no bonuses are used). Specifically, standard analysis of OMD requires bounding a "stability" term, which, for the negative entropy regularizer, is in the form of $\mathbb{E}[\sum_t \sum_{s,a} \widehat{q}_t(s,a) \widehat{\ell}_t(s,a)^2] = \mathbb{E}[\sum_t \sum_{s,a} \widehat{q}_t(s,a) \frac{q^{P_t, \pi_t}(s,a) \ell_t(s,a)^2}{u_t(s,a)^2}] \leq \mathbb{E}[\sum_t \sum_{s,a} \frac{q^{P_t, \pi_t}(s,a)}{u_t(s,a)}]$. Once again, when $C^P = 0$ and $P_t = P$, we have $q^{P_t, \pi_t}$ bounded by $u_t(s,a)$ with high probability, and thus the stability term is $\mathcal{O}(T|S||A|)$; but this breaks if $C^P \neq 0$ and $P_t$ can be arbitrarily different from transitions in $\mathcal{P}_{i(t)}$.

**Solution: Log-Barrier Regularizer** Resolving this second issue, however, is relatively straightforward — it suffices to switch the regularizer from negative entropy to log-barrier: $\phi(q) = -\sum_{k=0}^{L-1} \sum_{(s,a,s') \in W_k} \log q(s,a,s')$, which is first used by Lee et al. [2020] in the context of learning adversarial MDPs but dates back to earlier work such as Foster et al. [2016] for multi-armed bandits. An important property of log-barrier is that it leads to a smaller stability term in the form of $\mathbb{E}[\sum_t \sum_{s,a} \widehat{q}_t(s,a)^2 \widehat{\ell}_t(s,a)^2]$ (with an extra $\widehat{q}_t(s,a)$), which is at most $\mathbb{E}[\sum_t \sum_{s,a} q^{P_t, \pi_t}(s,a) \ell_t(s,a)^2] = \mathcal{O}(TL)$ since $\widehat{q}_t(s,a) \leq u_t(s,a)$. In fact, this also helps control the extra stability term when bonuses are used, which is in the form of $\mathbb{E}[\sum_t \sum_{s,a} \widehat{q}_t(s,a)^2 b_t(s,a)^2]$ and is at most $4L\mathbb{E}[\sum_t \langle \widehat{q}_t, b_t \rangle] = \mathcal{O}(L|S|C^P \log T)$ according to Lemma 3.1.

Putting these two ideas together leads to our final algorithm (see Algorithm 1). We prove the following regret bound in Appendix A, which recovers that of Jin et al. [2020] when $C^P = 0$ and increases linearly in $C^P$ as desired.

**Theorem 3.2.** *With $\delta = 1/T$ and $\eta = \min\left\{\sqrt{\frac{|S|^2|A|\log(\iota)}{LT}}, \frac{1}{8L}\right\}$, Algorithm 1 ensures*

$$\text{Reg}_T = \mathcal{O}\left(L|S|\sqrt{|A|T\log(\iota)} + L|S|^4|A|\log^2(\iota) + C^{\mathsf{P}}L|S|^4|A|\log(\iota)\right).$$

# 4 Achieving $\mathcal{O}(\sqrt{T} + C^{\mathsf{P}})$ with Unknown $C^{\mathsf{P}}$

In this section, we address the case when the amount of corruption is unknown. We develop a black-box reduction which turns an algorithm that only deals with known $C^{\mathsf{P}}$ to one that handles unknown $C^{\mathsf{P}}$. This is similar to Wei et al. [2022] but additionally handles adversarial losses using a different approach. A byproduct of our reduction is that we develop an entirely *black-box* model selection approach for adversarial online learning problems, as opposed to the *gray-box* approach developed by the "Corral" literature [Agarwal et al., 2017, Foster et al., 2020, Luo et al., 2022] which requires checking if the base algorithm is *stable*. To achieve this, we essentially develop another layer of reduction that turns any standard algorithm with sublinear regret into a stable algorithm. This result itself might be of independent interest and useful for solving other model selection problems.

More specifically, our reduction has two layers. The bottom layer is where our novelty lies: it takes as input an arbitrary corruption-robust algorithm that operates under known $C^{\mathsf{P}}$ (e.g., the one we developed in Section 3), and outputs a *stable* corruption-robust algorithm (formally defined later) that still operates under known $C^{\mathsf{P}}$. The top layer, on the other hand, follows the standard Corral idea and takes as input a stable algorithm that operates under known $C^{\mathsf{P}}$, and outputs an algorithm that operates under unknown $C^{\mathsf{P}}$. Below, we explain these two layers of reduction in details.

**Bottom Layer (from an Arbitrary Algorithm to a Stable Algorithm)** The input of the bottom layer is an arbitrary corruption-robust algorithm, formally defined as:

**Definition 4.1.** *An adversarial MDP algorithm is corruption-robust if it takes $\theta$ (a guess on the corruption amount) as input, and achieves the following regret for any random stopping time $t' \leq T$:*

$$\max_{\pi} \mathbb{E}\left[\sum_{t=1}^{t'}(\ell_t(\pi_t) - \ell_t(\pi))\right] \leq \mathbb{E}\left[\sqrt{\beta_1 t'} + (\beta_2 + \beta_3\theta)\mathbb{I}\{t' \geq 1\}\right] + \Pr[C_{1:t'}^{\mathsf{P}} > \theta]LT$$

*for problem-dependent constants and $\log(T)$ factors $\beta_1 \geq L^2, \beta_2 \geq L, \beta_3 \geq 1$, where $C_{1:t'}^{\mathsf{P}} = \sum_{\tau=1}^{t'} C_\tau^{\mathsf{P}}$ is the total corruption up to time $t'$.*

While the regret bound in Definition 4.1 might look cumbersome, it is in fact fairly reasonable: if the guess $\theta$ is not smaller than the true corruption amount, the regret should be of order $\sqrt{t'} + \theta$; otherwise, the regret bound is vacuous since $LT$ is its largest possible value. The only extra requirement is that the algorithm needs to be *anytime* (i.e., the regret bound holds for any stopping time $t'$), but even this is known to be easily achievable by using a doubling trick over a fixed-time algorithm. It is then clear that our algorithm in Section 3 (together with a doubling trick) indeed satisfies Definition 4.1.

As mentioned, the output of the bottom layer is a stable robust algorithm. To characterize stability, we follow Agarwal et al. [2017] and define a new learning protocol that abstracts the interaction between the output algorithm of the bottom layer and the master algorithm from the top layer:

**Protocol 1.** In every round $t$, before the learner makes a decision, a probability $w_t \in [0,1]$ is revealed to the learner. After making a decision, the learner sees the desired feedback from the environment with probability $w_t$, and sees nothing with probability $1 - w_t$.

In such a learning protocol, Agarwal et al. [2017] defines a stable algorithm as one whose regret smoothly degrades with $\rho_T = \frac{1}{\min_{t \in [T]} w_t}$. For our purpose here, we additionally require that the dependence on $C^{\mathsf{P}}$ in the regret bound is linear, which results in the following definition:

**Definition 4.2** ($\frac{1}{2}$-stable corruption-robust algorithm)**.** *A $\frac{1}{2}$-stable corruption-robust algorithm is one that, with prior knowledge on $C^{\mathsf{P}}$, achieves $\text{Reg}_T \leq \mathbb{E}\left[\sqrt{\beta_1\rho_T T} + \beta_2\rho_T\right] + \beta_3 C^{\mathsf{P}}$ under Protocol 1 for problem-dependent constants and $\log(T)$ factors $\beta_1 \geq L^2, \beta_2 \geq L$, and $\beta_3 \geq 1$.*

For simplicity, we only define and discuss the $\frac{1}{2}$-stability notion here (the parameter $\frac{1}{2}$ refers to the exponent of $T$), but our result can be straightforwardly extended to the general $\alpha$-stability notion for

**Algorithm 2** **ST**able **A**lgorithm **B**y **I**ndependent **L**earners and **I**nstance **SE**lection (STABILISE)

---

**Input**: $C^\mathsf{P}$ and a base algorithm satisfying Definition 4.1.
**Initialize**: $\lceil \log_2 T \rceil$ instances of the base algorithm $\mathsf{ALG}_1, \ldots, \mathsf{ALG}_{\lceil \log_2 T \rceil}$, where $\mathsf{ALG}_j$ is configured with the parameter

$$\theta = \theta_j \triangleq 2^{-j+1} C^\mathsf{P} + 16L \log(T).$$

**for** $t = 1, 2, \ldots$ **do**
    Receive $w_t$.
    **if** $w_t \leq \frac{1}{T}$ **then**
        play an arbitrary policy $\pi_t$
        **continue** (without updating any instances)
    Let $j_t$ be such that $w_t \in (2^{-j_t-1}, 2^{-j_t}]$.
    Let $\pi_t$ be the policy suggested by $\mathsf{ALG}_{j_t}$.
    Output $\pi_t$.
    If feedback is received, send it to $\mathsf{ALG}_{j_t}$ with probability $\frac{2^{-j_t-1}}{w_t}$, and discard it otherwise.

---

$\alpha \in [\frac{1}{2}, 1)$ as in Agarwal et al. [2017]. Our main result in this section is then that one can convert any corruption-robust algorithm into a $\frac{1}{2}$-stable corruption-robust algorithm:

**Theorem 4.3.** *If an algorithm is corruption robust according to Definition 4.1 for some constants* $(\beta_1, \beta_2, \beta_3)$, *then one can convert it to a* $\frac{1}{2}$-*stable corruption-robust algorithm (Definition 4.2) with constants* $(\beta_1', \beta_2', \beta_3')$ *where* $\beta_1' = \mathcal{O}(\beta_1 \log T)$, $\beta_2' = \mathcal{O}(\beta_2 + \beta_3 L \log T)$, *and* $\beta_3' = \mathcal{O}(\beta_3 \log T)$.

This conversion is achieved by a procedure that we call STABILISE (see Algorithm 2 for details). The high-level idea of STABILISE is as follows. Noticing that the challenge when learning in Protocol 1 is that $w_t$ varies over time, we discretize the value of $w_t$ and instantiate one instance of the input algorithm to deal with one possible discretized value, so that it is learning in Protocol 1 but with a *fixed* $w_t$, making it straightforward to bound its regret based on what it promises in Definition 4.1.

More concretely, STABILISE instantiates $\mathcal{O}(\log_2 T)$ instances $\{\mathsf{ALG}_j\}_{j=0}^{\lceil \log_2 T \rceil}$ of the input algorithm that satisfies Definition 4.1, each with a different parameter $\theta_j$. Upon receiving $w_t$ from the environment, it dispatches round $t$ to the $j$-th instance where $j$ is such that $w_t \in (2^{-j-1}, 2^{-j}]$, and uses the policy generated by $\mathsf{ALG}_j$ to interact with the environment (if $w_t \leq \frac{1}{T}$, simply ignore this round). Based on Protocol 1, the feedback for this round is received with probability $w_t$. To *equalize* the probability of $\mathsf{ALG}_j$ receiving feedback as mentioned in the high-level idea, when the feedback is actually obtained, STABILISE sends it to $\mathsf{ALG}_j$ only with probability $\frac{2^{-j-1}}{w_t}$ (and discards it otherwise). This way, every time $\mathsf{ALG}_j$ is assigned to a round, it always receives the desired feedback with probability $w_t \cdot \frac{2^{-j-1}}{w_t} = 2^{-j-1}$. This equalization step is the key that allows us to use the original guarantee of the base algorithm (Definition 4.1) and run it as it is, without requiring it to perform extra importance weighting steps as in Agarwal et al. [2017].

The choice of $\theta_j$ is crucial in making sure that STABILISE only has $C^\mathsf{P}$ regret overhead instead of $\rho_T C^\mathsf{P}$. Since $\mathsf{ALG}_j$ only receives feedback with probability $2^{-j-1}$, the expected total corruption it experiences is on the order of $2^{-j-1} C^\mathsf{P}$. Therefore, its input parameter $\theta_j$ only needs to be of this order instead of the total corruption $C^\mathsf{P}$. This is similar to the key idea of Wei et al. [2022] and Lykouris et al. [2018]. See Appendix B.1 for more details and the full proof of Theorem 4.3.

**Top Layer (from Known $C^\mathsf{P}$ to Unknown $C^\mathsf{P}$)** With a stable algorithm and a regret guarantee in Definition 4.2, it is relatively standard to convert it to an algorithm with $\widetilde{\mathcal{O}}(\sqrt{T} + C^\mathsf{P})$ regret without knowing $C^\mathsf{P}$. Similar arguments have been made in Foster et al. [2020], and the idea is to have another specially designed OMD/FTRL-based master algorithm to choose on the fly among a set of instances of this stable base algorithm, each with a different guess on $C^\mathsf{P}$ (the probability $w_t$ in Protocol 1 is then decided by this master algorithm). We defer all details to Appendix B. The final regret guarantee is the following ($\widetilde{\mathcal{O}}(\cdot)$ hides $\log(T)$ factors).

**Theorem 4.4.** *Using an algorithm satisfying Definition 4.2 as a base algorithm, Algorithm 3 (in the appendix) ensures* $\text{Reg}_T = \widetilde{\mathcal{O}}\left(\sqrt{\beta_1 T} + \beta_2 + \beta_3 C^{\mathsf{P}}\right)$ *without knowing* $C^{\mathsf{P}}$.

# 5  Gap-Dependent Refinements with Known $C^{\mathsf{P}}$

Finally, we discuss how to further improve our algorithm so that it adapts to easier environments and enjoys a better bound when the loss functions satisfy a certain gap condition, while still maintaining the $\mathcal{O}(\sqrt{T} + C^{\mathsf{P}})$ robustness guarantee. This result unfortunately requires the knowledge of $C^{\mathsf{P}}$ because the black-box approach introduced in the last section leads to $\sqrt{T}$ regret overhead already. We leave the possibility of removing this limitation for future work.

More concretely, following prior work such as Jin and Luo [2020], we consider the following general condition: there exists a mapping $\pi^\star : S \to A$, a gap function $\Delta : S \times A \to (0, L]$, and a constant $C^{\mathsf{L}} \geq 0$, such that for any policies $\pi_1, \ldots, \pi_T$ generated by the learner, we have

$$\mathbb{E}\left[\sum_{t=1}^{T}\left\langle q^{P,\pi_t} - q^{P,\pi^\star}, \ell_t\right\rangle\right] \geq \mathbb{E}\left[\sum_{t=1}^{T}\sum_{s\neq s_L}\sum_{a\neq\pi^\star(s)} q^{P,\pi_t}(s,a)\Delta(s,a)\right] - C^{\mathsf{L}}. \tag{9}$$

It has been shown that this condition subsumes the case when the loss functions are drawn from a fixed distribution (in which case $\pi^\star$ is simply the optimal policy with respect to the loss mean and $P$, $\Delta$ is the gap function with respect to the optimal $Q$-function, and $C^{\mathsf{L}} = 0$), or further corrupted by an adversary in an arbitrary manner subject to a budget of $C^{\mathsf{L}}$; we refer the readers to Jin and Luo [2020] for detailed explanation. Our main result for this section is a novel algorithm (whose pseudocode is deferred to Appendix C due to space limit) that achieves the following best-of-both-world guarantee.

**Theorem 5.1.** *Algorithm 4 (with $\delta = 1/T^2$ and $\gamma_t$ defined as in Definition 5.2) ensures*

$$\text{Reg}_T(\mathring{\pi}) = \mathcal{O}\left(L^2|S||A|\log(\iota)\sqrt{T} + \left(C^{\mathsf{P}} + 1\right)L^2|S|^4|A|^2\log^2(\iota)\right)$$

*always, and simultaneously the following gap-dependent bound under Condition (9):*

$$\text{Reg}_T(\pi^\star) = \mathcal{O}\left(U + \sqrt{UC^{\mathsf{L}}} + \left(C^{\mathsf{P}} + 1\right)L^2|S|^4|A|^2\log^2(\iota)\right),$$

*where* $U = \frac{L^3|S|^2|A|\log^2(\iota)}{\Delta_{\text{MIN}}} + \sum_{s\neq s_L}\sum_{a\neq\pi^\star(s)}\frac{L^2|S||A|\log^2(\iota)}{\Delta(s,a)}$ *and* $\Delta_{\text{MIN}} = \min\limits_{s\neq s_L, a\neq\pi^\star(s)}\Delta(s,a)$.

Aside from having larger dependence on parameters $L$, $S$, and $A$, Algorithm 4 maintains the same $\mathcal{O}(\sqrt{T} + C^{\mathsf{P}})$ regret as before, no matter how losses/transitions are generated; additionally, the $\sqrt{T}$ part can be significantly improved to $\mathcal{O}(U + \sqrt{UC^{\mathsf{L}}})$ (which can be of order only $\log^2 T$ when $C^{\mathsf{L}}$ is small) under Condition (9). This result not only generalizes that of Jin et al. [2021], Dann et al. [2023] from $C^{\mathsf{P}} = 0$ to any $C^{\mathsf{P}}$, but in fact also improves their results by having smaller dependence on $L$ in the definition of $U$. In the rest of this section, we describe the main ideas of our algorithm.

**FTRL with Epoch Schedule**   Our algorithm follows a line of research originated from Wei and Luo [2018], Zimmert and Seldin [2019] for multi-armed bandits and uses FTRL (instead of OMD) together with a certain self-bounding analysis technique. Since FTRL does not deal with varying decision sets easily, similar to Jin et al. [2021], we restart FTRL from scratch at the beginning of each epoch $i$ (recall the epoch schedule described in Section 2). More specifically, in an episode $t$ that belongs to epoch $i$, we now compute $\widehat{q}_t$ as $\text{argmin}_q\left\langle q, \sum_{\tau=t_i}^{t-1}(\widehat{\ell}_\tau - b_\tau)\right\rangle + \phi_t(q)$, where $t_i$ is the first episode of epoch $i$, $\widehat{\ell}_t$ is the same loss estimator defined in Eq. (5), $b_t$ is the amortized bonus defined in Eq. (7) (except that $\tau = 1$ there is also changed to $\tau = t_i$ due to restarting), $\phi_t$ is a time-varying regularizer to be specified later, and the set that $q$ is optimized over is also a key element to be discussed next. As before, the learner then simply executes $\pi_t = \pi^{\widehat{q}_t}$ for this episode.

**Optimistic Transition**   An important idea from Jin et al. [2021] is that if FTRL optimizes $q$ over $\Omega(\bar{P}_i)$ (occupancy measures with respect to a fixed transition $\bar{P}_i$) instead of $\Omega(\mathcal{P}_i)$ (occupancy measures with respect to a set of plausible transitions) as in UOB-REPS, then a critical *loss-shifting* technique can be applied in the analysis. However, the algorithm lacks "optimism" when not using a confidence set, which motivates Jin et al. [2021] to instead incorporate optimism by subtracting

a bonus term BONUS from the loss estimator (not to be confused with the amortized bonus $b_t$ we propose in this work). Indeed, if we define the value function $V^{\bar{P},\pi}(s;\ell)$ as the expected loss one suffers when starting from $s$ and following $\pi$ in an MDP with transition $\bar{P}$ and loss $\ell$, then they show that the BONUS term is such that $V^{\bar{P}_i,\pi}(s;\ell - \text{BONUS}) \leq V^{P,\pi}(s;\ell)$ for any state $s$ and any loss function $\ell$, that is, the performance of any policy is never underestimated.

Instead of following the same idea, here, we propose a simpler and better way to incorporate optimism via what we call *optimistic transitions*. Specifically, for each epoch $i$, we simply define an optimistic transition function $\widetilde{P}_i$ such that $\widetilde{P}_i(s'|s,a) = \max\{0, \bar{P}_i(s'|s,a) - B_i(s,a,s')\}$ (recall the confidence interval $B_i$ defined in Eq. (4)). Since this makes $\sum_{s'} \widetilde{P}_i(s'|s,a)$ less than 1, we allocate all the remaining probability to the terminal state $s_L$ (which breaks the layer structure but does not really affect anything). This is a form of optimism because reaching the terminate state earlier can only lead to smaller loss. More formally, under the high probability event $\mathcal{E}_{\text{CON}}$, we prove $V^{\widetilde{P}_i,\pi}(s;\ell) \leq V^{P,\pi}(s;\ell)$ for any policy $\pi$, any state $s$, and any loss function $\ell$ (see Lemma C.8.3).

With such an optimistic transition, we simply perform FTRL over $\Omega(\widetilde{P}_i)$ without adding any additional bonus term (other than $b_t$), making both the algorithm and the analysis much simpler than Jin et al. [2021]. Moreover, it can also be shown that $V^{\bar{P}_i,\pi}(s;\ell - \text{BONUS}) \leq V^{\widetilde{P}_i,\pi}(s;\ell)$ (see Lemma C.8.4), meaning that while both loss estimation schemes are optimistic, ours is tighter than that of Jin et al. [2021]. This eventually leads to the aforementioned improvement in the $U$ definition.

**Time-Varying Log-Barrier Regularizers** The final element to be specified in our algorithm is the time-varying regularizer $\phi_t$. Recall from discussions in Section 3 that using log-barrier as the regularizer is critical for bounding some stability terms in the presence of adversarial transitions. We thus consider the following log-barrier regularizer with an adaptive learning rate $\gamma_t : S \times A \to \mathbb{R}_+$: $\phi_t(q) = -\sum_{s \neq s_L}\sum_{a \in A}\gamma_t(s,a) \cdot \log q(s,a)$. The learning rate design requires combining the loss-shifting idea of [Jin et al., 2021] and the idea from [Ito, 2021], the latter of which is the first work to show that with adaptive learning rate tuning, the log-barrier regularizer leads to near-optimal best-of-both-world gaurantee for multi-armed bandits.

More specifically, following the same loss-shifting argument of Jin et al. [2021], we first observe that our FTRL update can be equivalently written as

$$\widehat{q}_t = \underset{q \in \Omega(\widetilde{P}_i)}{\operatorname{argmin}} \left\langle q, \sum_{\tau=t_i}^{t-1}(\widehat{\ell}_\tau - b_\tau)\right\rangle + \phi_t(q) = \underset{x \in \Omega(\widetilde{P}_i)}{\operatorname{argmin}} \left\langle q, \sum_{\tau=t_i}^{t-1}(g_\tau - b_\tau)\right\rangle + \phi_t(q),$$

where $g_\tau(s,a) = Q^{\widetilde{P}_i,\pi_\tau}(s,a;\widehat{\ell}_\tau) - V^{\widetilde{P}_i,\pi_\tau}(s;\widehat{\ell}_\tau)$ for any state-action pair $(s,a)$ ($Q$ is the standard $Q$-function; see Appendix C for formal definition). With this perspective, we follow the idea of Ito [2021] and propose the following learning rate schedule:

**Definition 5.2.** *(Adaptive learning rate for log-barrier) For any $t$, if it is the starting episode of an epoch, we set $\gamma_t(s,a) = 256L^2|S|$; otherwise, we set $\gamma_{t+1}(s,a) = \gamma_t(s,a) + \frac{D\nu_t(s,a)}{2\gamma_t(s,a)}$ where $D = 1/\log(\iota)$, $\nu_t(s,a) = q^{\widetilde{P}_{i(t)},\pi_t}(s,a)^2 \left(Q^{\widetilde{P}_{i(t)},\pi_t}(s,a;\widehat{\ell}_t) - V^{\widetilde{P}_{i(t)},\pi_t}(s;\widehat{\ell}_t)\right)^2$, and $i(t)$ is the epoch index to which episode $t$ belongs.*

Such a learning rate schedule is critical for the analysis in obtaining a certain self-bounding quantity and eventually deriving the gap-dependent bound. This concludes the design of our algorithm; see Appendix C for more details.

# 6  Conclusions

In this work, we propose online RL algorithms that can handle both adversarial losses and adversarial transitions, with regret gracefully degrading in the degree of maliciousness of the adversary. Specifically, we achieve $\widetilde{\mathcal{O}}(\sqrt{T} + C^{\mathsf{P}})$ regret where $C^{\mathsf{P}}$ measures how adversarial the transition functions are, even when $C^{\mathsf{P}}$ is unknown. Moreover, we show that further refinements of the algorithm not only maintain the same regret bound, but also simultaneously adapt to easier environments, with the caveat that $C^{\mathsf{P}}$ must be known ahead of time. We leave how to further remove this restriction as a key future direction.

## Acknowledgments and Disclosure of Funding

TJ and HL are supported by NSF Award IIS-1943607 and a Google Research Scholar award. CR acknowledges partial support by the Independent Research Fund Denmark, grant number 9040-00361B.

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

# Appendix

**Important Notations** Throughout the appendix, we denote the transition associated with the occupancy measure $\widehat{q}_t$ by $\widehat{P}_t$ so that $\widehat{q}_t = q^{\widehat{P}_t, \pi_t}$. Recall $i(t)$ is the epoch index to which episode $t$ belongs. Importantly, the notation $m_i(s, a)$ (and similarly $m_i(s, a, a')$), which is a changing variable in the algorithms, denotes the initial value of this counter in all analysis, that is, the total number of visits to $(s, a)$ *prior* to epoch $i$. Finally, for convenience, we define $\widehat{m}_i(s, a) = \max \left\{ m_i(s, a), C^{\mathsf{P}} + \log(\iota) \right\}$, and it can be verified that the confidence interval, defined in Eq. (4) as

$$B_i(s, a, s') = \min \left\{ 1, 16 \sqrt{\frac{\bar{P}_i(s'|s, a) \log(\iota)}{m_i(s, a)}} + 64 \cdot \frac{C^{\mathsf{P}} + \log(\iota)}{m_i(s, a)} \right\},$$

can be equivalently written as

$$B_i(s, a, s') = \min \left\{ 1, 16 \sqrt{\frac{\bar{P}_i(s'|s, a) \log(\iota)}{\widehat{m}_i(s, a)}} + 64 \cdot \frac{C^{\mathsf{P}} + \log(\iota)}{\widehat{m}_i(s, a)} \right\}$$

since whenever $\widehat{m}_i(s, a) \neq m_i(s, a)$, the two definitions both lead to a value of 1.

## A  Omitted Details for Section 3

In this section, we provide more details of the modified UOB-REPS algorithm as shown in Algorithm 1. In the algorithm, the occupancy measure for each episode $t$ is computed as

$$\widehat{q}_{t+1} = \operatorname*{argmin}_{q \in \Omega\left(\mathcal{P}_{i(t+1)}\right)} \eta \left\langle q, \widehat{\ell}_t - b_t \right\rangle + D_\phi(q, \widehat{q}_t), \tag{10}$$

where $\eta > 0$ is the learning rate and $D_\phi(q, q')$ is the Bregman divergence defined as

$$D_\phi(q, q') = \phi(q) - \phi(q') - \left\langle \nabla\phi(q'), q - q' \right\rangle. \tag{11}$$

The Bregman divergence is induced by the log-barrier regularizer $\phi$ given as:

$$\phi(q) = \sum_{k=0}^{L-1} \sum_{s \in S_k} \sum_{a \in A} \sum_{s' \in S_{k+1}} \log\left(\frac{1}{q(s, a, s')}\right). \tag{12}$$

We note that the loss estimator is constructed based upon upper occupancy bound $u_t$, which can be efficiently computed by COMP-UOB [Jin et al., 2020].

In the rest of this section, we prove Theorem 3.2 which shows that the expected regret is bounded by

$$\mathcal{O}\left( L|S|\sqrt{|A|T \log(\iota)} + L|S|^4|A| \log^2(\iota) + C^{\mathsf{P}} L|S|^4|A| \log(\iota) \right).$$

Our analysis starts from a regret decomposition similar to Jin et al. [2020], but with the amortized bonuses taken into account:

$$\mathrm{Reg}_T = \mathbb{E}\left[ \sum_{t=1}^{T} \left\langle q^{P_t, \pi_t} - q^{P_t, \mathring{\pi}}, \ell_t \right\rangle \right]$$

$$= \underbrace{\mathbb{E}\left[ \sum_{t=1}^{T} \left\langle q^{P_t, \pi_t} - q^{\widehat{P}_t, \pi_t}, \ell_t \right\rangle \right]}_{\text{ERROR}} + \underbrace{\mathbb{E}\left[ \sum_{t=1}^{T} \left\langle q^{\widehat{P}_t, \pi_t}, \ell_t - \widehat{\ell}_t \right\rangle \right]}_{\text{BIAS}_1}$$

$$+ \underbrace{\mathbb{E}\left[ \sum_{t=1}^{T} \left\langle q^{\widehat{P}_t, \pi_t} - q^{P, \mathring{\pi}}, \widehat{\ell}_t - b_t \right\rangle \right]}_{\text{REG}} + \underbrace{\mathbb{E}\left[ \sum_{t=1}^{T} \left\langle q^{P, \mathring{\pi}}, \widehat{\ell}_t - \ell_t \right\rangle + \sum_{t=1}^{T} \left\langle q^{\widehat{P}_t, \pi_t} - q^{P, \mathring{\pi}}, b_t \right\rangle \right]}_{\text{BIAS}_2}$$

$$+ \mathbb{E}\left[ \sum_{t=1}^{T} \left\langle q^{P, \mathring{\pi}} - q^{P_t, \mathring{\pi}}, \ell_t \right\rangle \right],$$

where the last term can be directly bounded by $\mathcal{O}\left(LC^{\mathsf{P}}\right)$ using Corollary D.3.7, and ERROR, BIAS$_1$, REG, and BIAS$_2$ are analyzed in Appendix A.2, Appendix A.3, Appendix A.4 and Appendix A.5 respectively.

## A.1 Equivalent Definition of Amortized Bonuses

For ease of exposition, we show an alternative definition of the amortized bonus $b_t(s)$, which is equivalent to Eq. (7) but is useful for our following analysis. Recall from Eq. (7) that

$$b_t(s) = b_t(s,a) = \begin{cases} \frac{4L}{u_t(s)} & \text{if } \sum_{\tau=1}^{t} \mathbb{I}\{\lceil \log_2 u_\tau(s) \rceil = \lceil \log_2 u_t(s) \rceil\} \leq \frac{C^{\mathsf{P}}}{2L}, \\ 0 & \text{else,} \end{cases} \tag{13}$$

In Eq. (13), the value of $b_t(s)$ relies on the cumulative sum of $\mathbb{I}\{\lceil \log_2 u_\tau(s) \rceil = \lceil \log_2 u_t(s) \rceil\}$ and whether the cumulative sum exceeds the threshold $\frac{C^{\mathsf{P}}}{2L}$. To reconstruct this, we define $y_t(s)$ as the unique integer value $j = -\lceil \log_2 u_t(s) \rceil$ such that $u_t(s) \in (2^{-j-1}, 2^{-j}]$ and define $z_t^j(s)$ as the total number of times from episode $\tau = 1$ to $\tau = t$ where $y_\tau(s) = j$ holds. With these definitions, the condition under which $b_t(s) = \frac{4L}{u_t(s)}$ can be represented by $\sum_j \mathbb{I}\{y_t(s) = j\}\mathbb{I}\left\{z_t^j(s) \leq C^{\mathsf{P}}/2L\right\}$ where the summation is over all integers that $-\lceil \log_2 u_t(s) \rceil$ can take. Notice that since $u_t(s) \geq \frac{1}{|S|T}$ holds for all $s$ and $t$ (see Lemma D.2.8), the largest possible integer value we need to consider is $\lceil \log_2 (|S|T) \rceil$. Therefore, $b_t(s)$ can be equivalently defined as:

$$b_t(s) = b_t(s,a) = \frac{4L}{u_t(s)} \sum_{j=0}^{\lceil \log_2(|S|T) \rceil} \mathbb{I}\{y_t(s) = j\}\mathbb{I}\left\{z_t^j(s) \leq \frac{C^{\mathsf{P}}}{2L}\right\}, \forall s \in S, \forall t \in \{1, \ldots, T\}. \tag{14}$$

## A.2 Bounding ERROR

**Lemma A.2.1** (Bound of ERROR). *Algorithm 1 ensures*

$$\text{ERROR} = \mathcal{O}\left(L|S|\sqrt{|A|T\log(\iota)} + L|S|^2|A|\log(\iota)\log(T) + C^{\mathsf{P}}L|S||A|\log(T) + \delta LT\right).$$

*Proof.* For this proof, we consider two cases, whether the event $\mathcal{E}_{\text{CON}} \wedge \mathcal{E}_{\text{EST}}$ holds or not, where $\mathcal{E}_{\text{CON}}$, defined above Eq. (4), and $\mathcal{E}_{\text{EST}}$, defined in Proposition D.5.2, are both high probability events.

Suppose that $\mathcal{E}_{\text{CON}} \wedge \mathcal{E}_{\text{EST}}$ holds. We have

$$\sum_{t=1}^{T} \left\langle q^{P_t, \pi_t} - q^{\widehat{P}_t, \pi_t}, \ell_t \right\rangle$$

$$\leq \sum_{t=1}^{T} \sum_{k=0}^{L-1} \sum_{s \in S_k} \sum_{a \in A} \left| q^{P_t, \pi_t}(s,a) - q^{\widehat{P}_t, \pi_t}(s,a) \right|$$

$$= \sum_{t=1}^{T} \sum_{k=0}^{L-1} \sum_{s \in S_k} \left| q^{\widehat{P}_t, \pi_t}(s) - q^{P, \pi_t}(s) \right|$$

$$\leq \sum_{t=1}^{T} \sum_{k=0}^{L-1} \sum_{s \in S_k} \sum_{h=0}^{k-1} \sum_{u \in S_h} \sum_{v \in A} \sum_{w \in S_{h+1}} q^{P_t, \pi_t}(u,v) \left| \widehat{P}_t(w|u,v) - P_t(w|u,v) \right| q^{\widehat{P}_t, \pi_t}(s|w)$$

$$\leq L \sum_{t=1}^{T} \sum_{h=0}^{L-1} \sum_{u \in S_h} \sum_{v \in A} q^{P_t, \pi_t}(u,v) \left\| \widehat{P}_t(\cdot|u,v) - P_t(\cdot|u,v) \right\|_1$$

$$\leq L \sum_{t=1}^{T} \sum_{h=0}^{L-1} \sum_{u \in S_h} \sum_{v \in A} q^{P_t, \pi_t}(u,v) \left( \|P_t(\cdot|u,v) - P(\cdot|u,v)\|_1 + \left\| \widehat{P}_t(\cdot|u,v) - P(\cdot|u,v) \right\|_1 \right)$$

$$\leq L \sum_{t=1}^{T} \sum_{h=0}^{L-1} \sum_{u \in S_h} \sum_{v \in A} q^{P_t, \pi_t}(u,v) \left\| \widehat{P}_t(\cdot|u,v) - P(\cdot|u,v) \right\|_1 + L \sum_{t=1}^{T} \sum_{h=0}^{L-1} C_{t,h}^{\mathsf{P}}$$

$$= L \sum_{t=1}^{T} \sum_{h=0}^{L-1} \sum_{u \in S_h} \sum_{v \in A} q^{P_t, \pi_t}(u,v) \left\| \widehat{P}_t(\cdot|u,v) - P(\cdot|u,v) \right\|_1 + LC^{\mathsf{P}}, \tag{15}$$

where the first step follows from the fact that $\ell_t(s,a) \in [0,1]$ for any $(s,a)$; the third step applies Lemma D.3.3; the fourth step rearranges the summation and uses the fact that $\sum_{s,a} q^{\widehat{P}_t,\pi_t}(s,a|w) \leq L$; and in the sixth step we define $C^{\mathsf{P}}_{t,h} = \max_{s \in S_h, a \in A} \|P_t(\cdot|s,a) - P(\cdot|s,a)\|_1$, so that $C^{\mathsf{P}} = \sum_{t=1}^{T} \sum_{h=0}^{L-1} C^{\mathsf{P}}_{t,h}$. We continue to bound the first term above as:

$$\sum_{t=1}^{T} \sum_{h=0}^{L-1} \sum_{u \in S_h} \sum_{v \in A} q^{P_t,\pi_t}(u,v) \left\| \widehat{P}_t(\cdot|u,v) - P(\cdot|u,v) \right\|_1$$

$$\leq \sum_{t=1}^{T} \sum_{h=0}^{L-1} \sum_{u \in S_h} \sum_{v \in A} q^{P_t,\pi_t}(u,v) \left( \left\| \widehat{P}_t(\cdot|u,v) - \bar{P}_{i(t)}(\cdot|u,v) \right\|_1 + \left\| \bar{P}_{i(t)}(\cdot|u,v) - P(\cdot|u,v) \right\|_1 \right)$$

$$\leq \sum_{h=0}^{L-1} \sum_{t=1}^{T} \sum_{u \in S_h} \sum_{v \in A} q^{P_t,\pi_t}(u,v) \, \mathcal{O} \left( \sqrt{\frac{|S_{h+1}|\log(\iota)}{\widehat{m}_{i(t)}(u,v)}} + \frac{C^{\mathsf{P}} + |S_{h+1}|\log(\iota)}{\widehat{m}_{i(t)}(u,v)} \right)$$

$$\leq \mathcal{O} \left( \sum_{h=0}^{L-1} \left( \sqrt{|S_h||S_{h+1}||A|T\log(\iota)} + |S_h||A|\log(T) \left( |S_{h+1}|\log(\iota) + C^{\mathsf{P}} \right) + \log(\iota) \right) \right)$$

$$\leq \mathcal{O} \left( \sum_{h=0}^{L-1} \left( (|S_h| + |S_{h+1}|) \sqrt{|A|T\log(\iota)} + |S_h||A|\log(T) \left( |S_{h+1}|\log(\iota) + C^{\mathsf{P}} \right) + \log(\iota) \right) \right)$$

$$\leq \mathcal{O} \left( |S|\sqrt{|A|T\log(\iota)} + |S|^2|A|\log(\iota)\log(T) + C^{\mathsf{P}}|S||A|\log(T) \right), \tag{16}$$

where the first step uses the triangle inequality; the second step applies Corollary D.2.6 to bound two norm terms based on the fact that $\widehat{P}_t, \bar{P}_{i(t)} \in \mathcal{P}_{i(t)}$; the third step follows the definition of $\mathcal{E}_{\mathrm{EST}}$ from Proposition D.5.2; the fourth step uses the AM-GM inequality. Putting the result of Eq. (16) into Eq. (15) yields the first three terms of the claimed bound.

Now suppose that $\mathcal{E}_{\mathrm{CON}} \wedge \mathcal{E}_{\mathrm{EST}}$ does not hold. We trivially bound $\sum_{t=1}^{T} \left\langle q^{P_t,\pi_t} - q^{\widehat{P}_t,\pi_t}, \ell_t \right\rangle$ by $LT$. As the probability that this case occurs is at most $\mathcal{O}(\delta)$, this case contributes to at most $\mathcal{O}(\delta LT)$ regret. Finally, combining these two cases and applying Lemma D.1.1 finishes the proof. $\square$

## A.3 Bounding BIAS$_1$

**Lemma A.3.1** (Bound of BIAS$_1$). *Algorithm 1 ensures*

$$\mathrm{BIAS}_1 = \mathcal{O} \left( L|S|\sqrt{|A|T\log(\iota)} + |S|^4|A|\log^2(\iota) + C^{\mathsf{P}}L|S|^4|A|\log(\iota) + \delta LT \right).$$

*Proof.* We can rewrite BIAS$_1$ as

$$\mathrm{BIAS}_1 = \sum_{t=1}^{T} \mathbb{E} \left[ \left\langle q^{\widehat{P}_t,\pi_t}, \ell_t - \widehat{\ell}_t \right\rangle \right] = \sum_{t=1}^{T} \mathbb{E} \left[ \left\langle q^{\widehat{P}_t,\pi_t}, \mathbb{E}_t \left[ \ell_t - \widehat{\ell}_t \right] \right\rangle \right],$$

where the second step uses the law of total expectation and the fact that $q^{\widehat{P}_t,\pi_t}$ is deterministic given all the history up to $t$. For any state-action pair $(s,a)$, we have

$$\mathbb{E}_t \left[ \ell_t(s,a) - \widehat{\ell}_t(s,a) \right] = \ell_t(s,a) \left( 1 - \frac{q^{P_t,\pi_t}(s,a)}{u_t(s,a)} \right) = \ell_t(s,a) \left( \frac{u_t(s,a) - q^{P_t,\pi_t}(s,a)}{u_t(s,a)} \right).$$

Then, we can further rewrite and bound BIAS$_1$ as:

$$\mathrm{BIAS}_1 = \mathbb{E} \left[ \sum_{t=1}^{T} \sum_{s,a} q^{\widehat{P}_t,\pi_t}(s,a)\ell_t(s,a) \left( \frac{u_t(s,a) - q^{P_t,\pi_t}(s,a)}{u_t(s,a)} \right) \right]$$

$$\leq \mathbb{E} \left[ \sum_{t=1}^{T} \sum_{s,a} q^{\widehat{P}_t,\pi_t}(s,a) \left( \frac{|u_t(s,a) - q^{P_t,\pi_t}(s,a)|}{u_t(s,a)} \right) \right]$$

$$\leq \mathbb{E}\left[\sum_{t=1}^{T}\sum_{s\in S}\left|u_t(s)-q^{P_t,\pi_t}(s)\right|\right],$$

where the third step follows from the fact that $q^{\widehat{P}_t,\pi_t}(s,a)\leq u_t(s,a)$ according to the definition of the upper occupancy measure.

Similar to the proof of Lemma A.2.1, we first consider the case when the high probability event $\mathcal{E}_{\mathrm{CON}}\wedge\mathcal{E}_{\mathrm{EST}}$ holds:

$$\sum_{s\in S}\left|u_t(s)-q^{P_t,\pi_t}(s)\right|$$

$$\leq \sum_{s\in S}\left|u_t(s)-q^{P,\pi_t}(s)\right|+\sum_{s\in S}\left|q^{P,\pi_t}(s)-q^{P_t,\pi_t}(s)\right|$$

$$\leq \sum_{s\in S}\left|u_t(s)-q^{P,\pi_t}(s)\right|+LC_t^{\mathsf{P}}$$

$$\leq \mathcal{O}\left(LC_t^{\mathsf{P}}+\sum_{s\in S}\sum_{k=0}^{k(s)-1}\sum_{(u,v,w)\in W_k}q^{P,\pi_t}(u,v)\sqrt{\frac{P(w|u,v)\log(\iota)}{\widehat{m}_{i(t)}(s,a)}}q^{P,\pi_t}(s|w)\right)$$

$$+\mathcal{O}\left(|S|^3\sum_{s,a}q^{P,\pi_t}(s,a)\frac{C^{\mathsf{P}}+\log(\iota)}{\widehat{m}_{i(t)}(s,a)}\right)$$

$$\leq \mathcal{O}\left(LC_t^{\mathsf{P}}+L\sum_{k=0}^{L-1}\sum_{(u,v,w)\in W_k}q^{P,\pi_t}(u,v)\sqrt{\frac{P(w|u,v)\log(\iota)}{\widehat{m}_{i(t)}(s,a)}}\right)$$

$$+\mathcal{O}\left(|S|^3\sum_{s,a}q^{P,\pi_t}(s,a)\frac{C^{\mathsf{P}}+\log(\iota)}{\widehat{m}_{i(t)}(s,a)}\right),$$

where the first step uses the triangle inequality; the second step uses Corollary D.3.6; the third step uses Lemma D.3.8; the last step follows from the fact that $\sum_{s\in S}q^{P,\pi_t}(s|w)\leq L$.

Taking the summation over all episodes yields the following:

$$\sum_{t=1}^{T}\sum_{s\in S}\left|u_t(s)-q^{P_t,\pi_t}(s)\right|$$

$$\leq \mathcal{O}\left(\sum_{t=1}^{T}LC_t^{\mathsf{P}}+L\sum_{t=1}^{T}\sum_{k=0}^{L-1}\sum_{(u,v,w)\in W_k}q^{P,\pi_t}(u,v)\sqrt{\frac{P(w|u,v)\log(\iota)}{\widehat{m}_{i(t)}(u,v)}}\right)$$

$$+\mathcal{O}\left(|S|^3\sum_{t=1}^{T}\sum_{s,a}q^{P,\pi_t}(s,a)\frac{C^{\mathsf{P}}+\log(\iota)}{\widehat{m}_{i(t)}(s,a)}\right)$$

$$\leq \mathcal{O}\left(LC^{\mathsf{P}}+|S|^3\sum_{k=0}^{L-1}\left(C^{\mathsf{P}}+\log(\iota)\right)|S_k||A|\log(\iota)\right)$$

$$+\mathcal{O}\left(L\sum_{t=1}^{T}\sum_{k=0}^{L-1}\sum_{(u,v)\in S_k\times A}q^{P,\pi_t}(u,v)\sqrt{\frac{|S_{k+1}|\log(\iota)}{\widehat{m}_{i(t)}(u,v)}}\right)$$

$$\leq \mathcal{O}\left(|S|^4|A|\left(C^{\mathsf{P}}+\log(\iota)\right)\log(\iota)+L\sum_{t=1}^{T}\sum_{k=0}^{L-1}\sum_{(u,v)\in S_k\times A}q^{P_t,\pi_t}(u,v)\sqrt{\frac{|S_{k+1}|\log(\iota)}{\widehat{m}_{i(t)}(u,v)}}\right)$$

$$+\mathcal{O}\left(L\sum_{t=1}^{T}\sum_{k=0}^{L-1}\sum_{(u,v)\in S_k\times A}\left|q^{P,\pi_t}(u,v)-q^{P_t,\pi_t}(u,v)\right|\sqrt{\frac{|S|\log(\iota)}{\widehat{m}_{i(t)}(u,v)}}\right)$$

$$\leq \mathcal{O}\left(|S|^4|A|\left(C^{\mathsf{P}} + \log\left(\iota\right)\right)\log\left(\iota\right) + L\sum_{t=1}^{T}\sum_{k=0}^{L-1}\sum_{(u,v)\in S_k\times A}q^{P_t,\pi_t}(u,v)\sqrt{\frac{|S_{k+1}|\log\left(\iota\right)}{\widehat{m}_{i(t)}(u,v)}}\right)$$

$$+ \mathcal{O}\left(L\sqrt{|S|}\sum_{t=1}^{T}\sum_{k=0}^{L-1}\sum_{(u,v)\in S_k\times A}\left|q^{P,\pi_t}(u,v) - q^{P_t,\pi_t}(u,v)\right|\right)$$

$$\leq \mathcal{O}\left(|S|^4|A|\left(C^{\mathsf{P}} + \log\left(\iota\right)\right)\log\left(\iota\right) + L\sum_{k=0}^{L-1}\sqrt{|S_{k+1}|\log\left(\iota\right)}\left(\sqrt{|S_k||A|T} + |S_k||A|\log\left(\iota\right)\right)\right)$$

$$= \mathcal{O}\left(|S|^4|A|\left(C^{\mathsf{P}} + \log\left(\iota\right)\right)\log\left(\iota\right) + L|S|\sqrt{|A|T\log\left(\iota\right)}\right),$$

where the second step uses the Cauchy-Schwartz inequality: $\sum_{w\in S_{k+1}}\sqrt{P(w|u,v)} \leq \sqrt{|S_{k+1}|}$ and also applies Lemma D.5.3; the fourth step bounds $\widehat{m}_{i(t)}(u,v) \geq \log(\iota)$ for any $(u,v)$ and $|S_{k+1}| \leq |S|$; the fifth applies Proposition D.5.2, and Corollary D.3.7 to bound $\sum_{t=1}^{T}\sum_{k=0}^{L-1}\sum_{(u,v)\in S_k\times A}\left|q^{P,\pi_t}(u,v) - q^{P_t,\pi_t}(u,v)\right|$ by $\mathcal{O}\left(LC^{\mathsf{P}}\right)$; the last step uses the fact that $\sqrt{xy} \leq x + y$ for any $x, y \geq 0$, and thus we have $\sum_{k=0}^{L-1}\sqrt{|S_k||S_{k+1}|} \leq 2\sum_{k=0}^{L-1}|S_k| = 2|S|$.

Finally, using a similar argument in the proof of Lemma A.2.1 to bound the case that $\mathcal{E}_{\text{CON}} \wedge \mathcal{E}_{\text{EST}}$ does not hold, we complete the proof. $\qquad\square$

## A.4 Bounding REG

**Lemma A.4.1** (Bound of REG). *If learning rate $\eta$ satisfies $\eta \in (0, \frac{1}{8L}]$, then, Algorithm 1 ensures*

$$\text{REG} = \mathcal{O}\left(\frac{|S|^2|A|\log\left(\iota\right)}{\eta} + \eta\left(L|S|C^{\mathsf{P}}\log T + LT\right) + \delta LT\right).$$

*Proof.* We consider a specific transition $P_0$ defined in Lee et al. [2020, Lemma C.4] and occupancy measure $u$ such that

$$u = \left(1 - \frac{1}{T}\right)q^{P,\mathring{\pi}} + \frac{1}{T|A|}\sum_{a\in A}q^{P_0,\pi_a},$$

where $\pi_a$ is a policy such that action $a$ is selected at every state.

By direct calculation, we have

$$D_\phi(u,\widehat{q}_1) = \sum_{k=0}^{L-1}\sum_{(s,a,s')\in W_k}\left(\log\left(\frac{\widehat{q}_1(s,a,s')}{u(s,a,s')}\right) + \frac{u(s,a,s')}{\widehat{q}_1(s,a,s')} - 1\right)$$

$$= \sum_{k=0}^{L-1}\sum_{(s,a,s')\in W_k}\log\left(\frac{\widehat{q}_1(s,a,s')}{u(s,a,s')}\right) + \sum_{k=0}^{L-1}\sum_{(s,a,s')\in W_k}\left(|S_k||A||S_{k+1}|u(s,a,s') - 1\right)$$

$$= \sum_{k=0}^{L-1}\sum_{(s,a,s')\in W_k}\log\left(\frac{\widehat{q}_1(s,a,s')}{u(s,a,s')}\right)$$

$$\leq 3|S|^2|A|\log\left(\iota\right),$$

where the second step uses the definition $\widehat{q}_1(s,a,s') = \frac{1}{|S_k||A||S_{k+1}|}$ for $k = 0,\cdots,L-1$ and the fourth step lower-bounds $u(s,a,s') \geq \frac{1}{T^3|S|^2|A|}$ from [Lee et al., 2020, Lemma C.10], thereby $u(s,a,s') \geq \frac{1}{\iota^3}$, and upper-bounds $\widehat{q}_1(s,a,s') \leq 1$.

According to [Lee et al., 2020, Lemma C.4], we have $q^{P_0,\pi_a} \in \cap_i \Omega\left(\mathcal{P}_i\right)$. Therefore, $u$ is a convex combination of points in that convex set, and we can use Lemma A.4.2 (included after this proof) to show

$$\left\langle q^{\widehat{P}_t,\pi_t} - u, \widehat{\ell}_t - b_t\right\rangle$$

$$\leq \frac{3|S|^2|A|\log(\iota)}{\eta} + 2\eta\left(\sum_{t=1}^{T}\sum_{s,a} q^{\widehat{P}_t,\pi_t}(s,a)^2\widehat{\ell}_t(s,a)^2 + \sum_{t=1}^{T}\sum_{s,a} q^{\widehat{P}_t,\pi_t}(s,a)^2 b_t(s)^2\right). \quad (17)$$

On the one hand, we bound the first summation in Eq. (17) as

$$\sum_{t=1}^{T}\sum_{s,a} q^{\widehat{P}_t,\pi_t}(s,a)^2\widehat{\ell}_t(s,a)^2 = \sum_{t=1}^{T}\sum_{s,a} q^{\widehat{P}_t,\pi_t}(s,a)^2 \cdot \frac{\ell_t(s,a)^2\mathbb{I}_t\{s,a\}}{u_t(s,a)^2} \leq LT,$$

since $q^{\widehat{P}_t,\pi_t}(s,a) \leq u_t(s,a)$ by definition of the upper occupancy bound.

On the other hand, we bound the second summation in Eq. (17) as

$$\sum_{t=1}^{T}\sum_{s,a} q^{\widehat{P}_t,\pi_t}(s,a)^2 b_t(s)^2 \leq 4L\sum_{t=1}^{T}\sum_{s} q^{\widehat{P}_t,\pi_t}(s)b_t(s) = \mathcal{O}\left(L|S|C^{\mathsf{P}}\log T\right), \quad (18)$$

where the first step uses the facts that $b_t(s)^2 \leq 4Lb_t(s)$ and $\sum_a q^{\widehat{P}_t,\pi_t}(s,a)^2 \leq q^{\widehat{P}_t,\pi_t}(s)$, and the last step applies Lemma 3.1.

Putting these inequalities together concludes the proof. $\square$

**Lemma A.4.2.** *With $\eta \in (0, \frac{1}{8L}]$, Algorithm 1 ensures*

$$\sum_{t=1}^{T}\left\langle \widehat{q}_t - u, \widehat{\ell}_t - b_t\right\rangle \leq \frac{D_\phi(u,\widehat{q}_1)}{\eta} + 2\eta\sum_{t=1}^{T}\sum_{s,a} \widehat{q}_t(s,a)^2\left(\widehat{\ell}_t(s,a)^2 + b_t(s)^2\right), \quad (19)$$

*for any $u \in \cap_i \Omega(\mathcal{P}_i)$.*

*Proof.* To use the standard analysis of OMD with log-barrier (e.g.,see [Agarwal et al., 2017, Lemma 12]), we need to ensure that $\eta\widehat{q}_t(s,a,s')\left(\widehat{\ell}_t(s,a) - b_t(s)\right) \geq -\frac{1}{2}$, since $\log(1+x) \geq x - x^2$ holds for any $x \geq -\frac{1}{2}$. Clearly, by choosing $\eta \in (0, \frac{1}{8L}]$, we have

$$\eta\widehat{q}_t(s,a,s')\left(\widehat{\ell}_t(s,a) - b_t(s)\right)$$
$$\geq -\eta\widehat{q}_t(s,a,s')b_t(s)$$
$$\geq -4L\eta\frac{\widehat{q}_t(s,a,s')}{u_t(s)}$$
$$\geq -4L\eta$$
$$\geq -\frac{1}{2},$$

where the first step follows from the fact that $\widehat{\ell}_t(s,a) \geq 0$ for all $t, s, a$; the second step follows from the definition of amortized bonus in Eq. (7); the third step bounds $\widehat{q}_t(s,a,s') \leq u_t(s)$.

Now, we are ready to apply the standard analysis to show that

$$\sum_{t=1}^{T}\left\langle \widehat{q}_t - u, \widehat{\ell}_t - b_t\right\rangle$$
$$\leq \frac{\sum_{t=1}^{T}\left(D_\phi(u,\widehat{q}_t) - D_\phi(u,\widehat{q}_{t+1})\right)}{\eta} + \eta\sum_{t=1}^{T}\sum_{s,a,s'}\widehat{q}_t(s,a,s')^2\left(\widehat{\ell}_t(s,a) - b_t(s)\right)^2$$
$$= \frac{D_\phi(u,\widehat{q}_1) - D_\phi(u,\widehat{q}_{T+1})}{\eta} + \eta\sum_{t=1}^{T}\sum_{s,a,s'}\left(\widehat{q}_t(s,a)\widehat{P}_t(s'|s,a)\right)^2\left(\widehat{\ell}_t(s,a) - b_t(s)\right)^2$$
$$\leq \frac{D_\phi(u,\widehat{q}_1)}{\eta} + \eta\sum_{t=1}^{T}\sum_{s,a}\widehat{q}_t(s,a)^2\left(\widehat{\ell}_t(s,a) - b_t(s)\right)^2 \cdot \left(\sum_{s'}\widehat{P}_t(s'|s,a)^2\right)$$

$$\leq \frac{D_\phi(u, \widehat{q}_1)}{\eta} + \eta \sum_{t=1}^{T} \sum_{s,a} \widehat{q}_t(s,a)^2 \left(\widehat{\ell}_t(s,a) - b_t(s)\right)^2$$

$$\leq \frac{D_\phi(u, \widehat{q}_1)}{\eta} + 2\eta \sum_{t=1}^{T} \sum_{s,a} \widehat{q}_t(s,a)^2 \left(\widehat{\ell}_t(s,a)^2 + b_t(s)^2\right),$$

where the second step uses the fact that $\widehat{q}_t(s,a,s') = \widehat{q}_t(s,a) \cdot \widehat{P}_t(s'|s,a)$ (see [Jin et al., 2020, Lemma 1] for more details); the third step follows from the fact that the Bregman divergence is non-negative; the fourth step follows from the fact $\sum_{s'} \widehat{P}_t(s'|s,a) = 1$; the last step uses the fact that $(x - y)^2 \leq 2\left(x^2 + y^2\right)$ for any $x, y \in \mathbb{R}$. $\qquad\square$

### A.5  Bounding $\text{BIAS}_2$

**Lemma A.5.1** (Bound of $\text{BIAS}_2$). *Algorithm 1 ensures*

$$\text{BIAS}_2 = \mathcal{O}\left(|S|C^P \log T + \delta LT\right).$$

*Proof.* We first rewrite $\text{BIAS}_2$ as

$$\text{BIAS}_2 = \underbrace{\mathbb{E}\left[\sum_{t=1}^{T} \mathbb{E}_t\left[\left\langle q^{P,\mathring{\pi}}, \widehat{\ell}_t - \ell_t\right\rangle\right] - \sum_{t=1}^{T} \left\langle q^{P,\mathring{\pi}}, b_t\right\rangle\right]}_{=:(I)} + \underbrace{\mathbb{E}\left[\sum_{t=1}^{T} \left\langle q^{\widehat{P}_t, \pi_t}, b_t\right\rangle\right]}_{=:(II)},$$

where (II) is bounded by $\mathcal{O}\left(|S|C^P \log T\right)$ by Lemma 3.1 (whose proof is included after this proof). Now, we show that (I) is bounded by $\mathcal{O}\left(\delta LT\right)$. Suppose that $\mathcal{E}_{\text{CON}}$ holds. We then have

$$\mathbb{E}_t\left[\left\langle q^{P,\mathring{\pi}}, \widehat{\ell}_t - \ell_t\right\rangle\right]$$

$$= \sum_{s,a} q^{P,\mathring{\pi}}(s,a) \left(\frac{q^{P_t, \pi_t}(s,a) - u_t(s,a)}{u_t(s,a)}\right)$$

$$\leq \sum_{s,a} q^{P,\mathring{\pi}}(s,a) \left(\frac{\left|q^{P_t, \pi_t}(s,a) - q^{P, \pi_t}(s,a)\right| + q^{P, \pi_t}(s,a) - u_t(s,a)}{u_t(s,a)}\right)$$

$$= \sum_{s,a} q^{P,\mathring{\pi}}(s,a) \cdot \frac{\left|q^{P_t, \pi_t}(s) - q^{P, \pi_t}(s)\right|}{u_t(s)} + \sum_{s,a} q^{P,\mathring{\pi}}(s,a) \left(\frac{q^{P, \pi_t}(s,a) - u_t(s,a)}{u_t(s,a)}\right)$$

$$\leq \sum_{s,a} q^{P,\mathring{\pi}}(s,a) \cdot \frac{C_t^P}{u_t(s)} + \sum_{s,a} q^{P,\mathring{\pi}}(s,a) \left(\frac{q^{P, \pi_t}(s,a) - u_t(s,a)}{u_t(s,a)}\right)$$

$$\leq \sum_{s,a} q^{P,\mathring{\pi}}(s,a) \cdot \frac{C_t^P}{u_t(s)},$$

where the fourth step applies Corollary D.3.6 to bound $\left|q^{P_t, \pi_t}(s) - q^{P, \pi_t}(s)\right| \leq C_t^P$, and in the last step, the second term (in the fifth step) is bounded by $0$ under $\mathcal{E}_{\text{CON}}$.

Since $q^{P,\mathring{\pi}}(s,a)$ is fixed over all episodes, we apply Lemma 3.1 to show

$$\sum_{t=1}^{T} \mathbb{E}_t\left[\left\langle q^{P,\mathring{\pi}}, \widehat{\ell}_t - \ell_t\right\rangle\right] - \sum_{t=1}^{T} \left\langle q^{P,\mathring{\pi}}, b_t\right\rangle \leq \sum_{s,a} q^{P,\mathring{\pi}}(s,a) \sum_{t=1}^{T} \left(\frac{C_t^P}{u_t(s)} - b_t(s)\right) \leq 0,$$

For the case that $\mathcal{E}_{\text{CON}}$ does not occur, we bound the expected regret by $\mathcal{O}\left(\delta LT\right)$. Combining two cases via Lemma D.1.1, we conclude the proof. $\qquad\square$

**Lemma A.5.2** (Restatement of Lemma 3.1). *The amortized bonus defined in Eq. (7) satisfies* $\sum_{t=1}^{T} \frac{C_t^P}{u_t(s)} \leq \sum_{t=1}^{T} b_t(s)$ *and* $\sum_{t=1}^{T} \widehat{q}_t(s)b_t(s) = \mathcal{O}(C^P \log T)$ *for any* $s$.

*Proof.* In the following proof, we use the equivalent definition of $b_t(s)$ given in Eq. (14). We first show $\sum_{t=1}^{T} \frac{C_t^{\mathsf{P}}}{u_t(s)} \leq \sum_{t=1}^{T} b_t(s)$. On the one hand, we have

$$
\begin{aligned}
\sum_{t=1}^{T} \left( \frac{C_t^{\mathsf{P}}}{u_t(s)} \right) &= \sum_{t=1}^{T} \sum_{j=0}^{\lceil \log(|S|T) \rceil} \mathbb{I}\{y_t(s) = j\} \frac{C_t^{\mathsf{P}}}{u_t(s)} \\
&\leq \sum_{t=1}^{T} \sum_{j=0}^{\lceil \log(|S|T) \rceil} \mathbb{I}\{y_t(s) = j\} \frac{C_t^{\mathsf{P}}}{2^{-j-1}} \\
&= \sum_{j=0}^{\lceil \log(|S|T) \rceil} \frac{\sum_{t=1}^{T} \mathbb{I}\{y_t(s) = j\} C_t^{\mathsf{P}}}{2^{-j-1}} \\
&\leq \sum_{j=0}^{\lceil \log(|S|T) \rceil} \frac{\min\{2L \sum_{t=1}^{T} \mathbb{I}\{y_t(s) = j\}, C^{\mathsf{P}}\}}{2^{-j-1}},
\end{aligned}
\tag{20}
$$

where the second step uses the construction of the bin, i.e., if $u_t(s)$ falls into a bin $(2^{-j-1}, 2^{-j}]$, then, it is lower-bounded by $2^{-j-1}$; the fourth step follows the facts that $C_t^{\mathsf{P}} \leq 2L$ and $\sum_{t=1}^{T} C_t^{\mathsf{P}} \leq C^{\mathsf{P}}$.

On the other hand, one can show

$$
\begin{aligned}
\sum_{t=1}^{T} b_t(s) &= 4L \sum_{t=1}^{T} \sum_{j=0}^{\lceil \log(|S|T) \rceil} \frac{\mathbb{I}\{y_t(s) = j\} \mathbb{I}\left\{ z_t^j(s) \leq \frac{C^{\mathsf{P}}}{2L} \right\}}{u_t(s)} \\
&\geq 4L \sum_{j=0}^{\lceil \log(|S|T) \rceil} \frac{\sum_{t=1}^{T} \mathbb{I}\{y_t(s) = j\} \mathbb{I}\left\{ z_t^j(s) \leq \frac{C^{\mathsf{P}}}{2L} \right\}}{2^{-j}} \\
&\geq 2L \sum_{j=0}^{\lceil \log(|S|T) \rceil} \frac{\min\{\sum_{t=1}^{T} \mathbb{I}\{y_t(s) = j\}, \frac{C^{\mathsf{P}}}{2L}\}}{2^{-j-1}} \\
&= \sum_{j=0}^{\lceil \log(|S|T) \rceil} \frac{\min\{2L \sum_{t=1}^{T} \mathbb{I}\{y_t(s) = j\}, C^{\mathsf{P}}\}}{2^{-j-1}},
\end{aligned}
\tag{21}
$$

where the second step uses the construction of the bin, i.e., if $u_t(s)$ falls into a bin $(2^{-j-1}, 2^{-j}]$, then, it is upper-bounded by $2^{-j}$, and the third step follows the definitions of $y_t(s)$ and $z_t^j(s)$.

Combining Eq. (20) and Eq. (21), we complete the proof of $\sum_{t=1}^{T} \frac{C_t^{\mathsf{P}}}{u_t(s)} \leq \sum_{t=1}^{T} b_t(s)$.

For the proof of $\sum_{t=1}^{T} \widehat{q}_t(s) b_t(s) = \mathcal{O}(|S| C^P \log T)$, one can show

$$
\begin{aligned}
\sum_{t=1}^{T} \widehat{q}_t(s) b_t(s) &= 4L \sum_{t=1}^{T} \sum_{i=0}^{\lceil \log(|S|T) \rceil} \frac{q^{\widehat{P}_t, \pi_t}(s) \mathbb{I}\{y_t(s) = i\} \mathbb{I}\left\{ z_t^i(s) \leq \frac{C^{\mathsf{P}}}{2L} \right\}}{u_t(s)} \\
&\leq 4L \sum_{t=1}^{T} \sum_{i=0}^{\lceil \log(|S|T) \rceil} \mathbb{I}\{y_t(s) = i\} \mathbb{I}\left\{ z_t^i(s) \leq \frac{C^{\mathsf{P}}}{2L} \right\} \\
&= 4L \sum_{i=0}^{\lceil \log(|S|T) \rceil} \sum_{t=1}^{T} \mathbb{I}\{y_t(s) = i\} \mathbb{I}\left\{ z_t^i(s) \leq \frac{C^{\mathsf{P}}}{2L} \right\} \\
&\leq 4L \sum_{i=0}^{\lceil \log(|S|T) \rceil} \frac{C^{\mathsf{P}}}{2L} \\
&= \mathcal{O}\left( C^{\mathsf{P}} \log(|S|T) \right) = \mathcal{O}\left( C^{\mathsf{P}} \log(T) \right),
\end{aligned}
$$

where the first inequality uses the fact $q^{\widehat{P}_t, \pi_t}(s) \leq u_t(s)$, and the last equality uses $|S| \leq T$. $\qquad \square$

## A.6 Proof of Theorem 3.2

For REG, we choose $\eta = \min\left\{\sqrt{\frac{|S|^2|A|\log(\iota)}{LT}}, \frac{1}{8L}\right\}$. First consider the case $\eta \neq \frac{1}{8L}$:

$$
\begin{aligned}
\text{REG} &= \mathcal{O}\left(\frac{|S|^2|A|\log(\iota)}{\eta} + \eta\left(L|S|C^{\mathsf{P}}\log T + LT\right) + LT\delta\right) \\
&\leq \mathcal{O}\left(\frac{|S|^2|A|\log(\iota)}{\eta} + \eta LT + |S|C^{\mathsf{P}}\log T + LT\delta\right) \\
&= \mathcal{O}\left(|S|\sqrt{|A|\log(\iota)\,LT} + |S|C^{\mathsf{P}}\log T + LT\delta\right),
\end{aligned}
$$

where the second step uses $\eta \leq \frac{1}{8L}$, and the third step applies choice of $\eta$ in the case of $\eta \neq \frac{1}{8L}$.

For the case of $\eta = \frac{1}{8L}$, we have $T \leq 64L|S|^2|A|\ln(\iota)$ and show

$$
\text{REG} = \mathcal{O}\left(L|S|^2|A|\log(\iota) + |S|C^{\mathsf{P}}\log T + LT\delta\right).
$$

Finally, choosing $\delta = \frac{1}{T}$ and putting the bound above together with ERROR, BIAS$_1$, and BIAS$_2$, we complete the proof of Theorem 3.2.

# B Omitted Details for Section 4

## B.1 Bottom Layer Reduction: STABILISE

*Proof of Theorem 4.3.* Define indicators

$$g_{t,j} = \mathbb{I}\{w_t \in (2^{-j-1}, 2^{-j}]\}$$
$$h_{t,j} = \mathbb{I}\{\mathsf{ALG}_j \text{ receives the feedback for episode } t\}.$$

Now we consider the regret of $\mathsf{ALG}_j$. Notice that $\mathsf{ALG}_j$ makes an update only when $g_{t,j}h_{t,j} = 1$. By the guarantee of the base algorithm (Definition 4.1), we have

$$\mathbb{E}\left[\sum_{t=1}^{T}(\ell_t(\pi_t) - \ell_t(\pi))g_{t,j}h_{t,j}\right]$$

$$\leq \mathbb{E}\left[\sqrt{\beta_1 \sum_{t=1}^{T} g_{t,j}h_{t,j} + (\beta_2 + \beta_3\theta_j)\max_{t\leq T} g_{t,j}}\right] + \Pr\left[\sum_{t=1}^{T} C_t^{\mathsf{P}} g_{t,j}h_{t,j} > \theta_j\right] LT. \tag{22}$$

We first bound the last term: Notice that $\mathbb{E}[h_{t,j}|g_{t,j}] = 2^{-j-1}g_{t,j}$ by Algorithm 2. Therefore,

$$\sum_{t=1}^{T} C_t^{\mathsf{P}} g_{t,j}\mathbb{E}[h_{t,j}|g_{t,j}] = 2^{-j-1}\sum_{t=1}^{T} C_t^{\mathsf{P}} g_{t,j} \leq 2^{-j-1}C^{\mathsf{P}} \tag{23}$$

By Freedman's inequality, with probability at least $1 - \frac{1}{T^2}$,

$$\sum_{t=1}^{T} C_t^{\mathsf{P}} g_{t,j}h_{t,j} - \sum_{t=1}^{T} C_t^{\mathsf{P}} g_{t,j}\mathbb{E}[h_{t,j}|g_{t,j}]$$

$$\leq 2\sqrt{2\sum_{t=1}^{T}(C_t^{\mathsf{P}})^2 g_{t,j}\mathbb{E}[h_{t,j}|g_{t,j}]\log(T) + 4L\log(T)}$$

$$\leq 4\sqrt{L\sum_{t=1}^{T} C_t^{\mathsf{P}} g_{t,j}\mathbb{E}[h_{t,j}|g_{t,j}]\log(T) + 4L\log(T)} \qquad (C_t^{\mathsf{P}} \leq 2L)$$

$$\leq \sum_{t=1}^{T} C_t^{\mathsf{P}} g_{t,j}\mathbb{E}[h_{t,j}|g_{t,j}] + 8L\log(T) \qquad \text{(AM-GM inequality)}$$

which gives

$$\sum_{t=1}^{T} C_t^{\mathsf{P}} g_{t,j}h_{t,j} \leq 2\sum_{t=1}^{T} C_t^{\mathsf{P}} g_{t,j}\mathbb{E}[h_{t,j}|g_{t,j}] + 8L\log(T)$$

$$\leq 2^{-j}C^{\mathsf{P}} + 8L\log(T) \leq \theta_j$$

with probability at least $1 - \frac{1}{T^2}$ using Eq. (23). Therefore, the last term in Eq. (22) is bounded by $\frac{1}{T^2}LT \leq \frac{L}{T}$.

Next, we deal with other terms in Eq. (22). Again, by $\mathbb{E}[h_{t,j}|g_{t,j}] = 2^{-j-1}g_{t,j}$, Eq. (22) implies

$$2^{-j-1}\mathbb{E}\left[\sum_{t=1}^{T}(\ell_t(\pi_t) - \ell_t(\pi))g_{t,j}\right] \leq \mathbb{E}\left[\sqrt{2^{-j-1}\beta_1 \sum_{t=1}^{T} g_{t,j} + (\beta_2 + \beta_3\theta_j)\max_{t\leq T} g_{t,j}}\right] + \frac{L}{T}.$$

which implies after rearranging:

$$\mathbb{E}\left[\sum_{t=1}^{T}(\ell_t(\pi_t) - \ell_t(\pi))g_{t,j}\right]$$

$$\leq \mathbb{E}\left[\sqrt{\frac{1}{2^{-j-1}}\beta_1 \sum_{t=1}^{T} g_{t,j}} + \left(\frac{\beta_2}{2^{-j-1}} + \frac{\beta_3 \theta_j}{2^{-j-1}}\right) \max_{t \leq T} g_{t,j}\right] + \frac{L}{T2^{-j-1}}$$

$$\leq \mathbb{E}\left[\sqrt{\beta_1 \sum_{t=1}^{T} \frac{2g_{t,j}}{w_t}} + \left(\frac{2\beta_2 + 16\beta_3 L \log(T)}{2^{-j}} + 4\beta_3 C^{\mathsf{P}}\right) \max_{t \leq T} g_{t,j}\right] + \frac{L}{T2^{-j-1}}.$$

$$\text{(using that when } g_{t,j} = 1, \frac{1}{2^{-j-1}} \leq \frac{2}{w_t}, \text{ and the definition of } \theta_j\text{)}$$

Now, summing this inequality over all $j \in \{0, 1, \ldots, \lceil \log_2 T \rceil\}$, we get

$$\mathbb{E}\left[\sum_{t=1}^{T}(\ell_t(\pi_t) - \ell_t(\pi))\mathbb{I}\left\{w_t \geq \frac{1}{T}\right\}\right]$$

$$\leq \mathcal{O}\left(\mathbb{E}\left[\sqrt{N\beta_1 \sum_{t=1}^{T}\frac{1}{w_t} + (\beta_2 + \beta_3 L \log(T))\frac{1}{\min_{t \leq T} w_t} + N\beta_3 C^{\mathsf{P}}}\right] + L\right)$$

$$\leq \mathcal{O}\left(\mathbb{E}\left[\sqrt{\beta_1 T \log(T)\rho_T} + (\beta_2 + \beta_3 L \log(T))\rho_T\right] + \beta_3 C^{\mathsf{P}}\log T + L\right)$$

where $N \leq \mathcal{O}(\log T)$ is the number of $\mathsf{ALG}_j$'s that has been executed at least once.

On the other hand,

$$\mathbb{E}\left[\sum_{t=1}^{T}(\ell_t(\pi_t) - \ell_t(\pi))\mathbb{I}\left\{w_t \leq \frac{1}{T}\right\}\right] \leq LT\mathbb{E}\left[\mathbb{I}\left\{\rho_T \geq T\right\}\right] \leq L\mathbb{E}\left[\rho_T\right].$$

Combining the two parts and using the assumption $\beta_2 \geq L$ finishes the proof. $\qquad\square$

## B.2 Top Layer Reduction: Corral

In this subsection, we use a base algorithm that satisfies Definition 4.2 to construct an algorithm with $\sqrt{T} + C^{\mathsf{P}}$ regret under unknown $C^{\mathsf{P}}$. The idea is to run multiple base algorithms, each with a different hypothesis on $C^{\mathsf{P}}$; on top of them, run another multi-armed bandit algorithm to adaptively choose among them. The goal is to let the top-level bandit algorithm perform almost as well as the best base algorithm. This is the Corral idea outlined in Agarwal et al. [2017], Foster et al. [2020], Luo et al. [2022], and the algorithm is presented in Algorithm 3.

The top-level bandit algorithm is an FTRL with log-barrier regularizer. We first state the standard regret bound of FTRL under log-barrier regularizer, whose proof can be found in, e.g., Theorem 7 of Wei and Luo [2018].

**Lemma B.2.1.** *The FTRL algorithm over a convex subset $\Omega$ of the $(M-1)$-dimensional simplex $\Delta(M)$:*

$$w_t = \operatorname*{argmin}_{w \in \Omega}\left\{\left\langle w, \sum_{\tau=1}^{t-1}\ell_\tau\right\rangle + \frac{1}{\eta}\sum_{i=1}^{M}\log\frac{1}{w_i}\right\}$$

*ensures for all $u \in \Omega$,*

$$\sum_{t=1}^{T}\langle w - u, \ell_t\rangle \leq \frac{M\log T}{\eta} + \eta \sum_{t=1}^{T}\sum_{i=1}^{M}w_{t,i}^2 \ell_{t,i}^2$$

*as long as $\eta w_{t,i}|\ell_{t,i}| \leq \frac{1}{2}$ for all $t, i$.*

*Proof of Theorem 4.4.* The Corral algorithm is essential an FTRL with log-barrier regularizer. To apply Lemma B.2.1, we first verify the condition $\eta w_{t,i}|\ell_{t,i}| \leq \frac{1}{2}$ where $\ell_{t,i} = \hat{c}_{t,i} - r_{t,i}$. By our choice of $\eta$,

$$\eta w_{t,i}|\hat{c}_{t,i}| \leq \eta c_{t,i} \leq \frac{1}{4}, \qquad\qquad\qquad \text{(because } \beta_2 \geq L \text{ by Definition 4.2)}$$

---

**Algorithm 3** (A Variant of) Corral

---

**Initialize**: a log-barrier algorithm with each arm being an instance of any base algorithm satisfying Definition 4.2. The hypothesis on $C^{\mathsf{P}}$ is set to $2^i$ for arm $i$ ($i = 1, 2, \ldots, M \triangleq \lceil \log_2 T \rceil$).
**Initialize**: $\rho_{0,i} = M, \ \forall i$

**for** $t = 1, 2, \ldots, T$ **do**
    Let

$$w_t = \operatorname*{argmin}_{w \in \Delta(M), w_i \geq \frac{1}{T}, \forall i} \left\{ \left\langle w, \sum_{\tau=1}^{t-1} (\hat{c}_\tau - r_\tau) \right\rangle + \frac{1}{\eta} \sum_{i=1}^{M} \log \frac{1}{w_i} \right\}$$

    where $\eta = \frac{1}{4(\sqrt{\beta_1 T} + \beta_2)}$.
    For all $i$, send $w_{t,i}$ to instance $i$.
    Draw $i_t \sim w_t$.
    Execute the $\pi_t$ output by instance $i_t$
    Receive the loss $c_{t,i_t}$ for policy $\pi_t$ (whose expectation is $\ell_t(\pi_t)$) and send it to instance $i_t$.
    Define for all $i$:

$$\hat{c}_{t,i} = \frac{c_{t,i} \mathbb{I}[i_t = i]}{w_{t,i}},$$

$$\rho_{t,i} = \min_{\tau \leq t} \frac{1}{w_{\tau,i}},$$

$$r_{t,i} = \sqrt{\beta_1 T} \left( \sqrt{\rho_{t,i}} - \sqrt{\rho_{t-1,i}} \right) + \beta_2 \left( \rho_{t,i} - \rho_{t-1,i} \right).$$

---

$$\eta w_{t,i} r_{t,i} = \eta \sqrt{\beta_1 T} w_{t,i} \left( \sqrt{\rho_{t,i}} - \sqrt{\rho_{t-1,i}} \right) + \eta \beta_2 w_{t,i} \left( \rho_{t,i} - \rho_{t-1,i} \right).$$

The right-hand side of the last equality is non-zero only when $\rho_{t,i} > \rho_{t-1,i}$, implying that $\rho_{t,i} = \frac{1}{w_{t,i}}$. Therefore, we further bound it by

$$
\begin{aligned}
& \eta w_{t,i} r_{t,i} \\
\leq\ & \eta \sqrt{\beta_1 T} \frac{1}{\rho_{t,i}} \left( \sqrt{\rho_{t,i}} - \sqrt{\rho_{t-1,i}} \right) + \eta \beta_2 \frac{1}{\rho_{t,i}} \left( \rho_{t,i} - \rho_{t-1,i} \right) \\
=\ & \eta \sqrt{\beta_1 T} \left( \frac{1}{\sqrt{\rho_{t,i}}} - \frac{\sqrt{\rho_{t-1,i}}}{\rho_{t,i}} \right) + \eta \beta_2 \left( 1 - \frac{\rho_{t-1,i}}{\rho_{t,i}} \right) \\
\leq\ & \eta \sqrt{\beta_1 T} \left( \frac{1}{\sqrt{\rho_{t-1,i}}} - \frac{1}{\sqrt{\rho_{t,i}}} \right) + \eta \beta_2 \left( 1 - \frac{\rho_{t-1,i}}{\rho_{t,i}} \right) \qquad (\tfrac{1}{\sqrt{a}} - \tfrac{\sqrt{b}}{a} \leq \tfrac{1}{\sqrt{b}} - \tfrac{1}{\sqrt{a}} \text{ for } a, b > 0) \\
& \hspace{11cm} (24) \\
\leq\ & \eta \sqrt{\beta_1 T} + \eta \beta_2 \hspace{7.5cm} (\rho_{t,i} \geq 1) \\
=\ & \frac{1}{4} \hspace{10.2cm} (\text{definition of } \eta)
\end{aligned}
$$

which can be combined to get the desired property $\eta w_{t,i} |\hat{c}_{t,i} - r_{t,i}| \leq \frac{1}{2}$.

Hence, by the regret guarantee of log-barrier FTRL (Lemma B.2.1), we have

$$
\begin{aligned}
& \mathbb{E}\left[ \sum_{t=1}^{T} (c_{t,i_t} - c_{t,i^\star}) \right] \\
\leq\ & \mathcal{O}\left( \frac{M \log T}{\eta} + \eta \mathbb{E}\left[ \underbrace{\sum_{t=1}^{T} \sum_{i=1}^{M} w_{t,i}^2 (\hat{c}_{t,i} - r_{t,i})^2}_{\text{stability-term}} \right] \right) + \mathbb{E}\left[ \underbrace{\sum_{t=1}^{T} \left( \sum_{i=1}^{M} w_{t,i} r_{t,i} - r_{t,i^\star} \right)}_{\text{bonus-term}} \right]
\end{aligned}
$$

where $i^\star$ is the smallest $i$ such that $2^i$ upper bounds the true corruption amount $C^{\mathsf{P}}$.

**Bounding stability-term**:

$$\textbf{stability term} \le 2\eta \sum_{t=1}^{T}\sum_{i=1}^{M} w_{t,i}^2 (\hat{c}_{t,i}^2 + r_{t,i}^2)$$

where

$$2\eta \sum_{t=1}^{T}\sum_{i=1}^{M} w_{t,i}^2 \hat{c}_{t,i}^2 = 2\eta \sum_{t=1}^{T}\sum_{i=1}^{M} c_{t,i}^2 \mathbb{I}\{i_t = i\} \le \mathcal{O}(\eta L^2 T)$$

and

$$2\eta \sum_{t=1}^{T}\sum_{i=1}^{M} w_{t,i}^2 r_{t,i}^2 \le 4\eta \sum_{t=1}^{T}\sum_{i=1}^{M}(\sqrt{\beta_1 T})^2 \left( \frac{1}{\sqrt{\rho_{t-1,i}}} - \frac{1}{\sqrt{\rho_{t,i}}} \right)^2 + 4\eta\beta_2 \sum_{t=1}^{T}\sum_{i=1}^{M} \left(1 - \frac{\rho_{t-1,i}}{\rho_{t,i}}\right)^2$$

(continue from Eq. (24))

$$\le 4\eta\beta_1 T \times \sum_{t=1}^{T}\sum_{i=1}^{M} \left( \frac{1}{\sqrt{\rho_{t-1,i}}} - \frac{1}{\sqrt{\rho_{t,i}}} \right) + 4\eta\beta_2 \sum_{t=1}^{T}\sum_{i=1}^{M} \ln \frac{\rho_{t,i}}{\rho_{t-1,i}}$$

$(\frac{1}{\sqrt{\rho_{t-1,i}}} - \frac{1}{\sqrt{\rho_{t,i}}} \le 1 \text{ and } 1 - a \le -\ln a)$

$$\le 4\eta\beta_1 T M^{\frac{3}{2}} + 4\eta\beta_2 M \ln T. \text{ (telescoping and using } \rho_{0,i} = M \text{ and } \rho_{T,i} \le T)$$

**Bounding bonus-term**:

$$\textbf{bonus-term} = \sum_{t=1}^{T}\sum_{i=1}^{M} w_{t,i} r_{t,i} - \sum_{t=1}^{T} r_{t,i^\star}$$

$$\le \sqrt{\beta_1 T} \sum_{t=1}^{T}\sum_{i=1}^{M} \left( \frac{1}{\sqrt{\rho_{t-1,i}}} - \frac{1}{\sqrt{\rho_{t,i}}} \right) + \beta_2 \sum_{t=1}^{T}\sum_{i=1}^{M} \log \frac{\rho_{t,i}}{\rho_{t-1,i}}$$

$$- \left( \sqrt{\rho_{T,i^\star}\beta_1 T} + \rho_{T,i^\star}\beta_2 - \sqrt{\rho_{0,i^\star}\beta_1 T} - \rho_{0,i^\star}\beta_2 \right)$$

(continue from Eq. (24) and using $1 - a \le -\ln a$)

$$\le \mathcal{O}\left( \sqrt{\beta_1 T} M^{\frac{3}{2}} + \beta_2 M \log T \right) - \left( \sqrt{\rho_{T,i^\star}\beta_1 T} + \rho_{T,i^\star}\beta_2 \right).$$

Combining the two terms and using $\eta = \Theta\left( \frac{1}{\sqrt{\beta_1 T} + \beta_2} \right)$, $M = \Theta(\log T)$, we get

$$\mathbb{E}\left[ \sum_{t=1}^{T}(\ell_t(\pi_t) - c_{t,i^\star}) \right] = \mathbb{E}\left[ \sum_{t=1}^{T}(c_{t,i_t} - c_{t,i^\star}) \right]$$

$$= \mathcal{O}\left( \sqrt{\beta_1 T \log^3 T} + \beta_2 \log^2 T \right) - \mathbb{E}\left[ \sqrt{\rho_{T,i^\star}\beta_1 T} + \rho_{T,i^\star}\beta_2 \right] \quad (25)$$

On the other hand, by Definition 4.2 and that $C^{\mathsf{P}} \in [2^{i^\star - 1}, 2^{i^\star}]$, we have

$$\mathbb{E}\left[ \sum_{t=1}^{T}(c_{t,i^\star} - \ell_t(\mathring{\pi})) \right] \le \mathbb{E}\left[ \sqrt{\rho_{T,i^\star}\beta_1 T} + \rho_{T,i^\star}\beta_2 \right] + 2\beta_3 C^{\mathsf{P}}. \quad (26)$$

Combining Eq. (25) and Eq. (26), we get

$$\mathbb{E}\left[ \sum_{t=1}^{T}(\ell_t(\pi_t) - \ell_t(\mathring{\pi})) \right] \le \mathcal{O}\left( \sqrt{\beta_1 T \log^3 T} + \beta_2 \log^2 T \right) + 2\beta_3 C^{\mathsf{P}},$$

which finishes the proof. $\qquad \square$

**Algorithm 4** Algorithm with Optimistic Transition Achieving Gap-Dependent Bounds (Known $C^{\mathsf{P}}$)

**Input:** confidence parameter $\delta \in (0,1)$.
**Initialize:** $\forall (s,a)$, learning rate $\gamma_1(s,a) = 256L^2|S|$; epoch index $i = 1$ and epoch starting time $t_i = 1$; $\forall (s,a,s')$, set counters $m_1(s,a) = m_1(s,a,s') = m_0(s,a) = m_0(s,a,s') = 0$; empirical transition $\bar{P}_1$ and confidence width $B_1$ based on Eq. (2); optimistic transition $\widetilde{P}_i$ by Definition C.1.1.
**for** $t = 1, \ldots, T$ **do**

> Let $\phi_t$ be defined in Eq. (28) and compute
>
> $$\widehat{q}_t = \underset{q \in \Omega(\widetilde{P}_i)}{\operatorname{argmin}} \left\langle q, \sum_{\tau = t_i}^{t-1} \left( \widehat{\ell}_\tau - b_\tau \right) \right\rangle + \phi_t(q). \tag{27}$$
>
> Compute policy $\pi_t$ from $\widehat{q}_t$ such that $\pi_t(a|s) \propto \widehat{q}_t(s,a)$.
> Execute policy $\pi_t$ and obtain trajectory $(s_{t,k}, a_{t,k})$ for $k = 0, \ldots, L-1$.
> Construct loss estimator $\widehat{\ell}_t$ as defined in Eq. (5).
> Compute amortized bonus $b_t$ based on Eq. (29).
> Compute learning rate $\gamma_{t+1}$ according to Definition 5.2.
> Increment counters: for each $k < L$, $m_i(s_{t,k}, a_{t,k}, s_{t,k+1}) \overset{+}{\leftarrow} 1$, $m_i(s_{t,k}, a_{t,k}) \overset{+}{\leftarrow} 1$.
>
> **if** $\exists k, \ m_i(s_{t,k}, a_{t,k}) \geq \max\{1, 2m_{i-1}(s_{t,k}, a_{t,k})\}$ **then**         ▷ entering a new epoch
> > Increment epoch index $i \overset{+}{\leftarrow} 1$ and set new epoch starting time $t_i = t + 1$.
> > Initialize new counters: $\forall (s,a,s')$, $m_i(s,a,s') = m_{i-1}(s,a,s'), m_i(s,a) = m_{i-1}(s,a)$.
> > Update empirical transition $\bar{P}_i$ and confidence width $B_i$ based on Eq. (2).
> > Update optimistic transition $\widetilde{P}_i$ based on Definition C.1.1

## C   Omitted Details for Section 5

In this section, we consider a variant of the algorithm proposed in Jin et al. [2021], which instead ensures exploration via *optimistic transitions* and also switch the regularizer from Tsallis entropy to log-barrier. We present the pseudocode in Algorithm 4, and show that it automatically adapts to easier environments with improved gap-dependent regret bounds.

**Remark C.1.** *Throughout the analysis, we denote by $Q^{P,\pi}(s,a;r)$ the state-action value of state-action pair $(s,a)$ with respect to transition function $P$ ($P$ could be an optimistic transition function), policy $\pi$ and loss function $r$; similarly, we denote by $V^{P,\pi}(s;r)$ the corresponding state-value function.*

*Specifically, the state value function $V^{P,\pi}(s;r)$ is computed in a backward manner from layer $L$ to layer $0$ as following:*

$$V^{P,\pi}(s;r) = \begin{cases} 0, & s = s_L, \\ \sum_{a \in A} \pi(a|s) \cdot \left( r(s,a) + \sum_{s' \in S_{k(s)+1}} P(s'|s,a) V^{P,\pi}(s';r) \right), & s \neq s_L. \end{cases}$$

*Similarly, the state-action value function $Q^{P,\pi}(s,a;r)$ is calculated in the same manner:*

$$Q^{P,\pi}(s,a;r) = \begin{cases} 0, & s = s_L, \\ r(s,a) + \sum_{s' \in S_{k(s)+1}} P(s'|s,a) \sum_{a' \in A} \pi(a'|s') Q^{P,\pi}(s',a';r), & s \neq s_L. \end{cases}$$

*Clearly, $V^{P,\pi}(s_0;r)$, the state value of $s_0$, is equal to*

$$V^{P,\pi}(s_0;r) = \sum_{s \neq s_L} \sum_{a \in A} q^{P,\pi}(s,a) r(s,a) \triangleq \left\langle q^{P,\pi}, r \right\rangle.$$

*This equality can be further extended to the other state $u$ as:*

$$V^{P,\pi}(u;r) = \sum_{s \neq s_L} \sum_{a \in A} q^{P,\pi}(s,a|u) r(s,a),$$

*where $q^{P,\pi}(s,a|u)$ denotes the probability of arriving at $(s,a)$ starting at state $u$ under policy $\pi$ and transition function $P$.*

## C.1 Description of the Algorithm

The construction of this algorithm follows similar ideas to the work of Jin et al. [2021], while also including several key differences. One of them is that we rely on the log-barrier regularizer, defined with a positive learning rate $\gamma_t(s, a)$ as

$$\phi_t(q) = -\sum_{s,a} \gamma_t(s, a) \log(q(s, a)). \tag{28}$$

The choice of log-barrier is important for adversarial transitions as discussed in Section 3, while the adaptive choice of learning rate is important for adapting to easy environments. The formal definition of the learning rate is given in Definition 5.2, and further details and properties of the learning rate are provided in Section C.7.

As the algorithm runs a new instance of FTRL for each epoch $i$, we modify the definition of the amortized bonus $b_t$ accordingly,

$$b_t(s) = b_t(s, a) = \begin{cases} \frac{4L}{u_t(s)} & \text{if } \sum_{\tau=t_{i(t)}}^{t} \mathbb{I}\{\lceil \log_2 u_\tau(s) \rceil = \lceil \log_2 u_t(s) \rceil\} \leq \frac{C^{\mathsf{P}}}{2L}, \\ 0 & \text{else.} \end{cases} \tag{29}$$

The bonus $b_t(s)$ in Eq. (29) is defined based on each epoch $i$, in the sense that $\sum_{\tau=1}^{t} \mathbb{I}\{\lceil \log_2 u_\tau(s) \rceil = \lceil \log_2 u_t(s) \rceil\}$ counts, among all previous rounds $\tau = t_i, \ldots, t$ in epoch $i$, the number of times that the value of $u_\tau(s)$ falls into the same bin as $u_t(s)$.

Again, for the ease of analysis, in the analysis we use the equivalent definition of the bonus $b_t(s)$ defined in Eq. (14), except that the counter $z_t^j(s)$ now will be reset to zero at the beginning of each epoch $i$.

Next, we formally define the optimistic transition as follows.

**Definition C.1.1** (Optimistic Transition). *For epoch $i$, the optimistic transition $\widetilde{P}_i : S \times A \times S \to [0, 1]$ is defined as:*

$$\widetilde{P}_i(s'|s, a) = \begin{cases} \max\left\{0, \bar{P}_i(s'|s, a) - B_i(s, a, s')\right\}, & (s, a, s') \in W_k, \\ \sum_{s' \in S_{k+1}} \min\left\{\bar{P}_i(s'|s, a), B_i(s, a, s')\right\}, & (s, a) \in S_k \times A \text{ and } s' = s_L, \end{cases}$$

*where $\bar{P}_i$ is the empirical transition defined in Eq. (2).*

Note that the optimistic transition $\widetilde{P}_i(\cdot|s, a)$ is a valid distribution as we have

$$\sum_{s' \in S_{k+1}} \min\left\{\bar{P}_i(s'|s, a), B_i(s, a, s')\right\} = 1 - \sum_{s' \in S_{k+1}} \max\left\{0, \bar{P}_i(s'|s, a) - B_i(s, a, s')\right\}.$$

We summarize the properties of the optimistic transition functions in Appendix C.8.

## C.2 Self-Bounding Properties of the Regret

In order to achieve best-of-both-worlds guarantees, our goal is to bound $\text{Reg}_T(\pi^\star)$ in terms of two self-bounding quantities for some $x > 0$ (plus other minor terms):

$$\mathbb{S}_1(x) = \sqrt{x \cdot \mathbb{E}\left[\sum_{t=1}^{T} \sum_{s \neq s_L} \sum_{a \neq \pi^\star(s)} q^{P_t, \pi_t}(s, a)\right]},$$

$$\mathbb{S}_2(x) = \sum_{s \neq s_L} \sum_{a \neq \pi^\star(s)} \sqrt{x \cdot \mathbb{E}\left[\sum_{t=1}^{T} q^{P_t, \pi_t}(s, a)\right]}. \tag{30}$$

These two quantities enjoy a certain self-bounding property which is critical to achieve the gap-dependent bound under Condition (9), as they can be related back to the regret against policy $\pi^\star$ itself. To see this, we first show the following implication of Condition (9).

**Lemma C.2.1.** *Under Condition* (9)*, the following holds.*

$$\mathrm{Reg}_T(\pi^\star) \geq \mathbb{E}\left[\sum_{t=1}^{T}\sum_{s\neq s_L}\sum_{a\neq\pi^\star(s)} q^{P_t,\pi_t}(s,a)\Delta(s,a)\right] - C^{\mathsf{L}} - 2LC^{\mathsf{P}} - L^2C^{\mathsf{P}}. \qquad (31)$$

*Proof.* From the definition of $\mathrm{Reg}_T(\pi^\star)$, we first note that:

$$\mathrm{Reg}_T(\pi^\star)$$

$$= \mathbb{E}\left[\sum_{t=1}^{T}\left\langle q^{P_t,\pi_t} - q^{P_t,\pi^\star}, \ell_t\right\rangle\right]$$

$$= \mathbb{E}\left[\sum_{t=1}^{T}\left\langle q^{P,\pi_t} - q^{P,\pi^\star}, \ell_t\right\rangle\right] + \mathbb{E}\left[\sum_{t=1}^{T}\left\langle q^{P_t,\pi_t} - q^{P,\pi_t}, \ell_t\right\rangle\right] + \mathbb{E}\left[\sum_{t=1}^{T}\left\langle q^{P,\pi^\star} - q^{P_t,\pi^\star}, \ell_t\right\rangle\right]$$

$$\geq \mathbb{E}\left[\sum_{t=1}^{T}\left\langle q^{P,\pi_t} - q^{P,\pi^\star}, \ell_t\right\rangle\right] - 2LC^{\mathsf{P}}$$

$$\geq \mathbb{E}\left[\sum_{t=1}^{T}\sum_{s\neq s_L}\sum_{a\neq\pi^\star(s)} q^{P,\pi_t}(s,a)\Delta(s,a)\right] - C^{\mathsf{L}} - 2LC^{\mathsf{P}}$$

where the third step applies Corollary D.3.7 and the last step uses the Condition (9). We continue to bound the first term above as:

$$\mathbb{E}\left[\sum_{t=1}^{T}\sum_{s\neq s_L}\sum_{a\neq\pi^\star(s)} q^{P,\pi_t}(s,a)\Delta(s,a)\right]$$

$$\geq \mathbb{E}\left[\sum_{t=1}^{T}\sum_{s\neq s_L}\sum_{a\neq\pi^\star(s)} q^{P_t,\pi_t}(s,a)\Delta(s,a)\right]$$

$$- \mathbb{E}\left[\sum_{t=1}^{T}\sum_{s\neq s_L}\sum_{a\neq\pi^\star(s)} \left|q^{P_t,\pi_t}(s,a) - q^{P,\pi_t}(s,a)\right|\Delta(s,a)\right]$$

$$\geq \mathbb{E}\left[\sum_{t=1}^{T}\sum_{s\neq s_L}\sum_{a\neq\pi^\star(s)} q^{P_t,\pi_t}(s,a)\Delta(s,a)\right] - L^2C^{\mathsf{P}},$$

where the last step uses $\Delta(s,a) \in (0, L]$ and Corollary D.3.6. $\qquad\square$

We are now ready to show the following important self-bounding properties (recall $\Delta_{\mathrm{MIN}} = \min_{s\neq s_L, a\neq\pi^\star(s)}\Delta(s,a)$).

**Lemma C.2.2** (Self-Bounding Quantities). *Under Condition* (31)*, we have for any $z > 0$:*

$$\mathbb{S}_1(x) \leq z\left(\mathrm{Reg}_T(\pi^\star) + C^{\mathsf{L}} + 2LC^{\mathsf{P}} + L^2C^{\mathsf{P}}\right) + \frac{1}{z}\left(\frac{x}{4\Delta_{\mathrm{MIN}}}\right),$$

$$\mathbb{S}_2(x) \leq z\left(\mathrm{Reg}_T(\pi^\star) + C^{\mathsf{L}} + 2LC^{\mathsf{P}} + L^2C^{\mathsf{P}}\right) + \frac{1}{z}\left(\sum_{s\neq s_L}\sum_{a\neq\pi^\star(s)}\frac{x}{4\Delta(s,a)}\right).$$

*Besides, it always holds that $\mathbb{S}_1(x) \leq \sqrt{x\cdot LT}$ and $\mathbb{S}_2(x) \leq \sqrt{x\cdot L|S||A|T}$.*

*Proof.* For any $z > 0$, we have

$$\mathbb{S}_1(x) = \sqrt{\frac{x}{2z\Delta_{\mathrm{MIN}}}\cdot\mathbb{E}\left[\left(2z\Delta_{\mathrm{MIN}}\sum_{t=1}^{T}\sum_{s\neq s_L}\sum_{a\neq\pi^\star(s)} q^{P_t,\pi_t}(s,a)\right)\right]}$$

$$\leq \mathbb{E}\left[z\Delta_{\text{MIN}}\sum_{t=1}^{T}\sum_{s\neq s_L}\sum_{a\neq\pi^\star(s)}q^{P_t,\pi_t}(s,a)\right] + \frac{x}{4z\Delta_{\text{MIN}}}$$

$$\leq z\left(\text{Reg}_T(\pi^\star) + C^{\mathsf{L}} + 2LC^{\mathsf{P}} + L^2C^{\mathsf{P}}\right) + \frac{x}{4z\Delta_{\text{MIN}}},$$

where the second step follows from the AM-GM inequality: $2\sqrt{xy} \leq x + y$ for any $x, y \geq 0$, and the last step follows from Condition (31).

By similar arguments, we have $\mathbb{S}_2(x)$ bounded for any $z > 0$ as:

$$\sum_{s\neq s_L}\sum_{a\neq\pi^\star(s)}\sqrt{\frac{x}{2z\Delta(s,a)}\cdot\mathbb{E}\left[2z\Delta(s,a)\sum_{t=1}^{T}q^{P_t,\pi_t}(s,a)\right]}$$

$$\leq \sum_{s\neq s_L}\sum_{a\neq\pi^\star(s)}\frac{x}{4z\Delta(s,a)} + z\mathbb{E}\left[\sum_{s\neq s_L}\sum_{a\neq\pi^\star(s)}\sum_{t=1}^{T}q^{P_t,\pi_t}(s,a)\Delta(s,a)\right]$$

$$= z\left(\text{Reg}_T(\pi^\star) + C^{\mathsf{L}} + 2LC^{\mathsf{P}} + L^2C^{\mathsf{P}}\right) + \sum_{s\neq s_L}\sum_{a\neq\pi^\star(s)}\frac{x}{4z\Delta(s,a)}.$$

Finally, by direct calculation, we can show that

$$\mathbb{S}_1(x) = \sqrt{x\cdot\mathbb{E}\left[\sum_{t=1}^{T}\sum_{s\neq s_L}\sum_{a\neq\pi^\star(s)}q^{P_t,\pi_t}(s,a)\right]} \leq \sqrt{x\cdot LT},$$

according to the fact that $\sum_{s\neq s_L}\sum_{a\in A}q^{P_t,\pi_t}(s,a) \leq L$. On the other hand, $\mathbb{S}_2(x)$ is bounded as

$$\mathbb{S}_2(x) \leq \sqrt{x\cdot|S||A|\mathbb{E}\left[\sum_{s\neq s_L}\sum_{a\neq\pi^\star(s)}\sum_{t=1}^{T}q^{P_t,\pi_t}(s,a)\right]} \leq \sqrt{x\cdot L|S||A|T},$$

with the help of the Cauchy-Schwarz inequality in the first step. $\qquad\square$

Finally, we show that Algorithm 4 achieves the following adaptive regret bound, which directly leads to the best-of-both-worlds guarantee in Theorem 5.1.

**Lemma C.2.3.** *Algorithm 4 with $\delta = 1/T^2$ and learning rate defined in Definition 5.2 ensures that, for any mapping $\pi^\star : S \to A$, the regret $\text{Reg}_T(\pi^\star)$ is bounded by $\mathcal{O}\left(\left(C^{\mathsf{P}} + 1\right)L^2|S|^4|A|^2\log^2(\iota)\right)$ plus*

$$\mathcal{O}\left(\mathbb{S}_1\left(L^3|S|^2|A|\log^2(\iota)\right) + \mathbb{S}_2\left(L^2|S||A|\log^2(\iota)\right)\right).$$

The proof of this result is detailed in Section C.3. We emphasize that this bound holds for any mapping $\pi^\star$, and is not limited to that policy in Eq. (9). This is important for proving the robustness result when losses are arbitrary, as shown in the following proof of Theorem 5.1.

*Proof of Theorem 5.1.* When losses are arbitrary, we simply select $\pi^\star = \mathring{\pi}$ where $\mathring{\pi}$ is one of the optimal deterministic policies in hindsight and obtain the following bound of $\text{Reg}_T(\mathring{\pi})$:

$$\mathcal{O}\left(\left(C^{\mathsf{P}} + 1\right)L^2|S|^4|A|^2\log^2(\iota) + \mathbb{S}_1\left(L^3|S|^2|A|\log^2(\iota)\right) + \mathbb{S}_2\left(L^2|S||A|\log^2(\iota)\right)\right)$$

$$= \mathcal{O}\left(\left(C^{\mathsf{P}} + 1\right)L^2|S|^4|A|^2\log^2(\iota) + \sqrt{L^4|S|^2|A|T\log^2(\iota)} + \sqrt{L^3|S|^2|A|^2T\log^2(\iota)}\right),$$

where the first step follows from Lemma C.2.3 and the second step follows from Lemma C.2.2.

Next, suppose that Condition (9) holds. We set $\pi^\star$ as defined in the condition and use Lemma C.2.3 to write the regret against $\pi^\star$ as:

$$\text{Reg}_T(\pi^\star) \leq \mu\left(\mathbb{S}_1\left(L^3|S|^2|A|\log^2(\iota)\right) + \mathbb{S}_2\left(L^2|S||A|\log^2(\iota)\right)\right) + \xi\left(C^{\mathsf{P}} + 1\right),$$

where $\mu > 0$ is an absolute constant and $\xi = \mathcal{O}\left(L^2|S|^4|A|^2\log^2(\iota)\right)$.

For any $z > 0$, according to Lemma C.2.2 (where we set the $z$ there as $z/2\mu$), we have:

$$\text{Reg}_T(\pi^\star) \leq z\left(\text{Reg}_T(\pi^\star) + C^{\mathsf{L}} + 2LC^{\mathsf{P}} + L^2C^{\mathsf{P}}\right)$$
$$+ \frac{\mu^2}{z}\left(\frac{L^3|S|^2|A|\log^2(\iota)}{\Delta_{\text{MIN}}} + \sum_{s\neq s_L}\sum_{a\neq\pi^\star(s)}\frac{L^2|S||A|\log^2(\iota)}{\Delta(s,a)}\right) + \xi\left(C^{\mathsf{P}}+1\right).$$

Let $x = \frac{1-z}{z}$ and $U = \frac{L^3|S|^2|A|\log^2(\iota)}{\Delta_{\text{MIN}}} + \sum_{s\neq s_L}\sum_{a\neq\pi^\star(s)}\frac{L^2|S||A|\log^2(\iota)}{\Delta(s,a)}$. Rearranging the above inequality leads to

$$\text{Reg}_T(\pi^\star) \leq \frac{z\left(C^{\mathsf{L}} + 2LC^{\mathsf{P}} + L^2C^{\mathsf{P}}\right)}{1-z} + \frac{\mu^2 U}{z\left(1-z\right)} + \frac{\xi\left(C^{\mathsf{P}}+1\right)}{1-z}$$
$$= \frac{\left(C^{\mathsf{L}} + 2LC^{\mathsf{P}} + L^2C^{\mathsf{P}}\right)}{x} + \left(x + 2 + \frac{1}{x}\right)\mu^2 U + \left(1 + \frac{1}{x}\right)\xi\left(C^{\mathsf{P}}+1\right)$$
$$= \frac{1}{x}\left(C^{\mathsf{L}} + 2LC^{\mathsf{P}} + L^2C^{\mathsf{P}} + \mu^2 U + \xi\left(C^{\mathsf{P}}+1\right)\right) + x\cdot\mu^2 U + 2\mu^2 U + \xi\left(C^{\mathsf{P}}+1\right).$$

Picking the optimal $x$ to minimize the upper bound of $\text{Reg}_T(\pi^\star)$ yields

$$\text{Reg}_T(\pi^\star) = 2\sqrt{\left(C^{\mathsf{L}} + 2LC^{\mathsf{P}} + L^2C^{\mathsf{P}} + \mu^2 U + \xi\left(C^{\mathsf{P}}+1\right)\right)\mu^2 U} + 2\mu^2 U + \xi\left(C^{\mathsf{P}}+1\right)$$
$$\leq 2\sqrt{\left(C^{\mathsf{L}} + 2LC^{\mathsf{P}} + L^2C^{\mathsf{P}} + \xi\left(C^{\mathsf{P}}+1\right)\right)\mu^2 U} + 4\mu^2 U + \xi\left(C^{\mathsf{P}}+1\right)$$
$$= \mathcal{O}\left(\sqrt{\left(C^{\mathsf{L}} + L^2|S|^4|A|^2\log^2(\iota)C^{\mathsf{P}}\right)U} + U + L^2|S|^4|A|\log^2(\iota)\left(C^{\mathsf{P}}+1\right)\right)$$
$$\leq \mathcal{O}\left(\sqrt{UC^{\mathsf{L}}} + U + L^2|S|^4|A|^2\log^2(\iota)\left(C^{\mathsf{P}}+1\right)\right),$$

where the second step uses $\sqrt{x+y} \leq \sqrt{x} + \sqrt{y}$ for any $x, y \in \mathbb{R}_{\geq 0}$ and the last step uses $2\sqrt{xy} \leq x + y$ for any $x, y \geq 0$. $\qquad\square$

## C.3 Regret Decomposition of $\text{Reg}_T(\pi^\star)$ and Proof of Lemma C.2.3

In the following sections, we will first decompose $\text{Reg}_T(\pi^\star)$ for any mapping $\pi^\star : S \to A$ into several parts, and then bound each part separately from Section C.4 to Section C.7, in order to prove Lemma C.2.3.

For any mapping $\pi^\star : S \to A$, we start from the following decomposition of $\text{Reg}_T(\pi^\star)$ as

$$\underbrace{\mathbb{E}\left[\sum_{t=1}^{T}\left\langle q^{P_t,\pi_t}, \ell_t\right\rangle - \left\langle q^{\widehat{P}_t,\pi_t}, \widehat{\ell}_t - b_t\right\rangle\right]}_{\text{ERROR}_1} + \underbrace{\mathbb{E}\left[\sum_{t=1}^{T}\left\langle q^{\widehat{P}_t,\pi_t}, \widehat{\ell}_t - b_t\right\rangle - \left\langle q^{\widehat{P}_t,\pi^\star}, \widehat{\ell}_t - b_t\right\rangle\right]}_{\text{ESTREG}}$$
$$+ \underbrace{\mathbb{E}\left[\sum_{t=1}^{T}\left\langle q^{\widehat{P}_t,\pi^\star}, \widehat{\ell}_t - b_t\right\rangle - \left\langle q^{P_t,\pi^\star}, \ell_t\right\rangle\right]}_{\text{ERROR}_2},$$
(32)

where $\widehat{P}_t = \widetilde{P}_{i(t)}$ denotes the optimistic transition for episode $t$ for simplicity (which is consistent with earlier definition such that $\widehat{q}_t = q^{\widehat{P}_t,\pi_t}$). Here, ESTREG is the estimated regret controlled by FTRL, while $\text{ERROR}_1$ and $\text{ERROR}_2$ are estimation errors incurred on the selected policies $\{\pi_t\}_{t=1}^{T}$, and that on the comparator policy $\pi^\star$ respectively.

In order to achieve the gap-dependent bound under Condition (9), we consider these two estimation error terms $\text{ERROR}_1$ and $\text{ERROR}_2$ together as:

$$\mathbb{E}\left[\sum_{t=1}^{T}\left\langle q^{P_t,\pi_t},\ell_t\right\rangle - \left\langle q^{\widehat{P}_t,\pi_t},\widehat{\ell}_t - b_t\right\rangle + \left\langle q^{\widehat{P}_t,\pi^\star},\widehat{\ell}_t - b_t\right\rangle - \left\langle q^{P_t,\pi^\star},\ell_t\right\rangle\right]$$

$$= \mathbb{E}\left[\sum_{t=1}^{T}\left\langle q^{P_t,\pi_t},\ell_t\right\rangle - \left\langle q^{\widehat{P}_t,\pi_t},\mathbb{E}_t\left[\widehat{\ell}_t\right] - b_t\right\rangle + \left\langle q^{\widehat{P}_t,\pi^\star},\mathbb{E}_t\left[\widehat{\ell}_t\right] - b_t\right\rangle - \left\langle q^{P_t,\pi^\star},\ell_t\right\rangle\right],$$

(33)

where $\mathbb{E}_t\left[\cdot\right]$ denotes the conditional expectation given the history prior to episode $t$.

To better analyze the conditional expectation of the loss estimators, we define $\alpha_t,\beta_t : S \times A \to \mathbb{R}$ as:

$$\alpha_t(s,a) \triangleq \frac{q^{P,\pi_t}(s,a)\ell_t(s,a)}{u_t(s,a)}, \quad \beta_t(s,a) \triangleq \frac{\left(q^{P_t,\pi_t}(s,a) - q^{P,\pi_t}(s,a)\right)\ell_t(s,a)}{u_t(s,a)},$$

which ensures that $\mathbb{E}_t\left[\widehat{\ell}_t(s,a)\right] = \alpha_t(s,a) + \beta_t(s,a)$ for any state-action pair $(s,a)$.

With the help of $\alpha_t,\beta_t$, we have

$$\left\langle q^{\widehat{P}_t,\pi_t},\mathbb{E}_t\left[\widehat{\ell}_t\right] - b_t\right\rangle = \left\langle q^{\widehat{P}_t,\pi_t},\alpha_t\right\rangle + \left\langle q^{\widehat{P}_t,\pi_t},\beta_t - b_t\right\rangle,$$

$$\left\langle q^{\widehat{P}_t,\pi^\star},\mathbb{E}_t\left[\widehat{\ell}_t\right] - b_t\right\rangle = \left\langle q^{\widehat{P}_t,\pi^\star},\alpha_t\right\rangle + \left\langle q^{\widehat{P}_t,\pi^\star},\beta_t - b_t\right\rangle,$$

which helps us further rewrite Eq. (33) as

$$\mathbb{E}\left[\sum_{t=1}^{T}\left\langle q^{P_t,\pi_t} - q^{P,\pi_t},\ell_t\right\rangle + \sum_{t=1}^{T}\left\langle q^{P,\pi^\star} - q^{P_t,\pi^\star},\ell_t\right\rangle\right]$$

$$+ \mathbb{E}\left[\sum_{t=1}^{T}\left\langle q^{P,\pi_t},\ell_t\right\rangle - \left\langle q^{\widehat{P}_t,\pi_t},\alpha_t\right\rangle + \left\langle q^{\widehat{P}_t,\pi^\star},\alpha_t\right\rangle - \left\langle q^{P,\pi^\star},\ell_t\right\rangle\right]$$

(34)

$$+ \mathbb{E}\left[\sum_{t=1}^{T}\left\langle q^{\widehat{P}_t,\pi_t},b_t - \beta_t\right\rangle + \sum_{t=1}^{T}\left\langle q^{\widehat{P}_t,\pi^\star},\beta_t - b_t\right\rangle\right].$$

Based on this decomposition of $\text{ERROR}_1 + \text{ERROR}_2$, we then bound each parts respectively in the following lemmas.

**Lemma C.3.1.** *For any $\delta \in (0,1)$ and any policy sequence $\{\pi_t\}_{t=1}^{T}$, Algorithm 4 ensures that*

$$\mathbb{E}\left[\sum_{t=1}^{T}\left\langle q^{P_t,\pi_t} - q^{P,\pi_t},\ell_t\right\rangle + \sum_{t=1}^{T}\left\langle q^{P,\pi^\star} - q^{P_t,\pi^\star},\ell_t\right\rangle\right] = \mathcal{O}\left(C^P L\right).$$

**Lemma C.3.2.** *For any $\delta \in (0,1)$ and any mapping $\pi^\star : S \to A$, Algorithm 4 ensures that*

$$\mathbb{E}\left[\sum_{t=1}^{T}\left\langle q^{\widehat{P}_t,\pi_t},b_t - \beta_t\right\rangle + \left\langle q^{\widehat{P}_t,\pi^\star},\beta_t - b_t\right\rangle\right] = \mathcal{O}\left(C^P L|S|^2|A|\log^2(T)\right).$$

**Lemma C.3.3.** *For any $\delta \in (0,1)$ and any mapping $\pi^\star : S \to A$, Algorithm 4 ensures that*

$$\mathbb{E}\left[\sum_{t=1}^{T}\left\langle q^{P,\pi_t},\ell_t\right\rangle - \left\langle q^{\widehat{P}_t,\pi_t},\alpha_t\right\rangle + \left\langle q^{\widehat{P}_t,\pi^\star},\alpha_t\right\rangle - \left\langle q^{P,\pi^\star},\ell_t\right\rangle\right]$$

$$= \mathcal{O}\left(\mathbb{S}_1\left(L^3|S|^2|A|\log^2(\iota)\right) + \left(C^P + 1\right)L^2|S|^4|A|\log^2(\iota) + \delta L|S|^2|A|T^2\right).$$

**Lemma C.3.4.** *With the learning rates $\{\gamma_t\}_{t=1}^{T}$ defined in Definition 5.2, Algorithm 4 ensures that for any $\delta \in (0,1)$ and any mapping $\pi^\star$ that,*

$\text{ESTREG}(\pi^\star)$

$$= \mathbb{E}\left[\sum_{t=1}^{T}\left\langle q^{\widehat{P}_t,\pi_t},\widehat{\ell}_t - b_t\right\rangle - \left\langle q^{\widehat{P}_t,\pi^\star},\widehat{\ell}_t - b_t\right\rangle\right]$$

$$= \mathcal{O}\left(\mathbb{S}_2\left(L^2|S||A|\log^2(\iota)\right) + \left(C^P + 1\right)L|S|^2|A|^2\log^2(\iota) + \delta TL^2|S|^2|A|\log(\iota)\right).$$

*Proof of Lemma C.2.3.* According to the previous discussion, we have the regret against any mapping $\pi^\star : S \to A$ decomposed as:

$$\text{Reg}_T(\pi^\star) = \mathbb{E}\left[\sum_{t=1}^{T}\left\langle q^{P_t,\pi_t} - q^{P,\pi_t}, \ell_t\right\rangle + \sum_{t=1}^{T}\left\langle q^{P,\pi^\star} - q^{P_t,\pi^\star}, \ell_t\right\rangle\right]$$

$$+ \mathbb{E}\left[\sum_{t=1}^{T}\left\langle q^{\widehat{P}_t,\pi_t}, b_t - \beta_t\right\rangle + \sum_{t=1}^{T}\left\langle q^{\widehat{P}_t,\pi^\star}, \beta_t - b_t\right\rangle\right]$$

$$+ \mathbb{E}\left[\sum_{t=1}^{T}\left\langle q^{P,\pi_t}, \ell_t\right\rangle - \left\langle q^{\widehat{P}_t,\pi_t}, \alpha_t\right\rangle + \left\langle q^{\widehat{P}_t,\pi^\star}, \alpha_t\right\rangle - \left\langle q^{P,\pi^\star}, \ell_t\right\rangle\right]$$

$$+ \mathbb{E}\left[\sum_{t=1}^{T}\left\langle q^{\widehat{P}_t,\pi_t}, \widehat{\ell}_t - b_t\right\rangle - \left\langle q^{\widehat{P}_t,\pi^\star}, \widehat{\ell}_t - b_t\right\rangle\right],$$

where the first term is mainly caused by the difference between $\{P_t\}_{t=1}^{T}$ and $P$, which is unavoidable and can be bounded by $\mathcal{O}\left(C^{\mathsf{P}}L\right)$ as shown in Lemma C.3.1; the second term is the extra cost of using the amortized losses $b_t$ to handle the biases of loss estimators, which is controlled by $\widetilde{\mathcal{O}}(C^{\mathsf{P}})$ as shown in Lemma C.3.2; the third term measures the estimation error related to the optimistic transitions $\{\widehat{P}_t\}_{t=1}^{T}$, which can be bounded by some self-bounding quantities in Lemma C.3.2; the final term is the estimated regret calculated with respect to the optimistic transitions $\{\widehat{P}_t\}_{t=1}^{T}$, which is controlled by FTRL as shown in Lemma C.3.4. Putting all these bounds together finishes the proof. $\qquad\square$

## C.4 Proof of Lemma C.3.1

The result is immediate by directly applying Corollary D.3.7.

## C.5 Proof of Lemma C.3.2

For this proof, we bound $\mathbb{E}\left[\sum_{t=1}^{T}\left\langle q^{\widehat{P}_t,\pi_t}, b_t - \beta_t\right\rangle\right]$ and $\mathbb{E}\left[\sum_{t=1}^{T}\left\langle q^{\widehat{P}_t,\pi^\star}, \beta_t - b_t\right\rangle\right]$ separately.

**Bounding** $\mathbb{E}\left[\sum_{t=1}^{T}\left\langle q^{\widehat{P}_t,\pi_t}, b_t - \beta_t\right\rangle\right]$. We have

$$\sum_{t=1}^{T}\left\langle q^{\widehat{P}_t,\pi_t}, b_t - \beta_t\right\rangle$$

$$= \sum_{t=1}^{T}\sum_{s,a} q^{\widehat{P}_t,\pi_t}(s,a)\left(b_t(s) - \frac{\left(q^{P_t,\pi_t}(s,a) - q^{P,\pi_t}(s,a)\right)\ell_t(s,a)}{u_t(s,a)}\right)$$

$$\leq \sum_{t=1}^{T}\sum_{s,a} q^{\widehat{P}_t,\pi_t}(s,a)\left(b_t(s) + \frac{\left|q^{P_t,\pi_t}(s,a) - q^{P,\pi_t}(s,a)\right|}{u_t(s,a)}\right)$$

$$\leq \sum_{t=1}^{T}\sum_{s,a} q^{\widehat{P}_t,\pi_t}(s,a)b_t(s) + \sum_{t=1}^{T}\sum_{s,a}\left|q^{P_t,\pi_t}(s,a) - q^{P,\pi_t}(s,a)\right|$$

$$\leq \sum_{t=1}^{T}\sum_{s,a} q^{\widehat{P}_t,\pi_t}(s,a)b_t(s) + LC^{\mathsf{P}},$$

where the second step bounds $\ell_t(s,a) \leq 1$; the third step follows the fact that $q^{\widehat{P}_t,\pi_t}(s,a) \leq u_t(s,a)$ for all $(s,a)$. Let $E_i$ be a set of episodes that belong to epoch $i$ and let $N$ be the total number of epochs through $T$ episodes. Then, we turn to bound

$$\sum_{t=1}^{T}\sum_{s,a} q^{\widehat{P}_t,\pi_t}(s,a)b_t(s) = \sum_{i=1}^{N}\sum_{t\in E_i}\sum_{s,a} q^{\widehat{P}_t,\pi_t}(s,a)b_t(s) \leq \mathcal{O}\left(L|S|^2|A|\log^2\left(T\right)C^{\mathsf{P}}\right),$$

where the last step repeats the same argument of Lemma 3.1 for every epoch and the number of epochs is at most $O(|S||A|\log T)$ according to [Jin et al., 2021, Lemma D.3.12].

**Bounding** $\mathbb{E}\left[\sum_{t=1}^{T}\left\langle q^{\widehat{P}_t,\pi^{\star}},\beta_t - b_t\right\rangle\right]$. For this term, we show that for any given epoch $i$, $\sum_{t\in E_i}\left\langle q^{\widehat{P}_t,\pi^{\star}},\beta_t - b_t\right\rangle \leq 0$ which yields $\sum_{t=1}^{T}\left\langle q^{\widehat{P}_t,\pi^{\star}},\beta_t - b_t\right\rangle \leq 0$. To this end, we first consider a fixed epoch $i$ and upper-bound

$$\left\langle q^{\widehat{P}_t,\pi^{\star}},\beta_t\right\rangle = \sum_{s,a} q^{\widehat{P}_t,\pi^{\star}}(s,a)\left(\frac{\left(q^{P_t,\pi_t}(s) - q^{P,\pi_t}(s)\right)\ell_t(s,a)}{u_t(s)}\right) \leq \sum_{s,a} q^{\widehat{P}_t,\pi^{\star}}(s,a)\left(\frac{C_t^{\mathsf{P}}}{u_t(s)}\right),$$

where the second step uses Corollary D.3.6 to bound $\left(q^{P_t,\pi_t}(s) - q^{P,\pi_t}(s)\right) \leq C_t^{\mathsf{P}}$.

For any epoch $i$ and episode $t \in E_i$, we have $\widehat{P}_t = \widetilde{P}_{i(t)}$, which gives that

$$\sum_{t=1}^{T}\left\langle q^{\widehat{P}_t,\pi^{\star}},\beta_t - b_t\right\rangle = \sum_{i=1}^{N}\sum_{t\in E_i}\left\langle q^{\widetilde{P}_i,\pi^{\star}},\beta_t - b_t\right\rangle$$

$$\leq \sum_{i=1}^{N}\sum_{s,a} q^{\widetilde{P}_i,\pi^{\star}}(s,a)\sum_{t\in E_i}\left(\frac{C_t^{\mathsf{P}}}{u_t(s)} - b_t(s)\right)$$

$$\leq 0,$$

where the last step applies Lemma 3.1 for every epoch $i$. Thus, $\mathbb{E}\left[\sum_{t=1}^{T}\left\langle q^{\widehat{P}_t,\pi^{\star}},\beta_t - b_t\right\rangle\right] \leq 0$ holds.

### C.6 Proof of Lemma C.3.3

We introduce the following lemma to evaluate the estimated performance via the true occupancy measure, which helps us analyze the the estimation error.

**Lemma C.6.1.** *For any transition function pair* $(P,\widehat{P})$ *($P$ and $\widehat{P}$ can be optimistic transition), policy* $\pi$, *and loss function* $\ell$, *it holds that* $\left\langle q^{P,\pi},\ell\right\rangle = \left\langle q^{\widehat{P},\pi}, \mathcal{Z}_{\ell}^{P,\widehat{P},\pi}\right\rangle$, *where* $\mathcal{Z}_{\ell}^{P,\widehat{P},\pi}$ *is defined as*

$$\mathcal{Z}_{\ell}^{P,\widehat{P},\pi}(s,a) = Q^{P,\pi}(s,a;\ell) - \sum_{s'\in S_{k(s)+1}}\widehat{P}(s'|s,a)V^{P,\pi}(s';\ell)$$

$$= \ell(s,a) + \sum_{s'\in S_{k(s)+1}}\left(P(s'|s,a) - \widehat{P}(s'|s,a)\right)V^{P,\pi}(s';\ell)$$

*for all state-action pairs* $(s,a)$.

*Proof.* By direct calculation, we have:

$$V^{P,\pi}(s;\ell) = \sum_a \pi(a|s)Q^{P,\pi}(s,a;\ell)$$

$$= \sum_a \pi(a|s)\left(Q^{P,\pi}(s,a;\ell) - \sum_{s'\in S_{k(s)+1}}\widehat{P}(s'|s,a)V^{P,\pi}(s;\ell)\right)$$

$$+ \sum_a \pi(a|s)\sum_{s'\in S_{k(s)+1}}\widehat{P}(s'|s,a)V^{P,\pi}(s';\ell)$$

$$= \sum_{s'\in S_{k(s)+1}} q^{\widehat{P},\pi}(s'|s)V^{P,\pi}(s';\ell)$$

$$+ \sum_a \pi(a|s)\left(Q^{P,\pi}(s,a;\ell) - \sum_{s'\in S_{k(s)+1}}\widehat{P}(s'|s,a)V^{P,\pi}(s;\ell)\right)$$

$$= \sum_{k=k(s)}^{L-1} \sum_{s' \in S_k} \sum_{a \in A} q^{\widehat{P}, \pi}(s', a | s) \left( Q^{P, \pi}(s', a; \ell) - \sum_{s'' \in S_{k(s')+1}} \widehat{P}(s'' | s', a) V^{P, \pi}(s''; \ell) \right)$$

$$= \sum_{k=k(s)}^{L-1} \sum_{s' \in S_k} \sum_{a \in A} q^{\widehat{P}, \pi}(s', a | s) \mathcal{Z}_\ell^{P, \widehat{P}, \pi}(s', a)$$

where the second to last step follows from recursively repeating the first three steps. The proof is completed by noticing $\langle q^{P, \pi}, \ell \rangle = V^{P, \pi}(s_0, \ell)$. □

According to Lemma C.6.1, we rewrite $\left\langle q^{\widehat{P}_t, \pi_t}, \alpha_t \right\rangle$ and $\left\langle q^{\widehat{P}_t, \pi^\star}, \alpha_t \right\rangle$ with $q^{P, \pi_t}$ and $q^{P, \pi^\star}$ for any policy $\pi^\star$ as

$$\left\langle q^{\widehat{P}_t, \pi_t}, \alpha_t \right\rangle = \left\langle q^{P, \pi_t}, \mathcal{Z}_{\alpha_t}^{\widehat{P}_t, P, \pi_t} \right\rangle, \quad \left\langle q^{\widehat{P}_t, \pi^\star}, \alpha_t \right\rangle = \left\langle q^{P, \pi^\star}, \mathcal{Z}_{\alpha_t}^{\widehat{P}_t, P, \pi^\star} \right\rangle.$$

Therefore, we can further decompose as

$$\mathbb{E} \left[ \left\langle q^{P, \pi_t}, \ell_t \right\rangle - \left\langle q^{\widehat{P}_t, \pi_t}, \alpha_t \right\rangle + \left\langle q^{\widehat{P}_t, \pi^\star}, \alpha_t \right\rangle - \left\langle q^{P, \pi^\star}, \ell_t \right\rangle \right]$$

$$= \mathbb{E} \left[ \sum_{t=1}^T \left\langle q^{P, \pi_t}, \ell_t - \mathcal{Z}_{\alpha_t}^{\widehat{P}_t, P, \pi_t} \right\rangle - \left\langle q^{P, \pi^\star}, \ell_t - \mathcal{Z}_{\alpha_t}^{\widehat{P}_t, P, \pi^\star} \right\rangle \right]$$

$$= \mathbb{E} \left[ \sum_{t=1}^T \left\langle q^{P, \pi_t} - q^{P, \pi^\star}, \ell_t - \mathcal{Z}_{\alpha_t}^{\widehat{P}_t, P, \pi^\star} \right\rangle + \left\langle q^{P, \pi_t}, \mathcal{Z}_{\alpha_t}^{\widehat{P}_t, P, \pi^\star} - \mathcal{Z}_{\alpha_t}^{\widehat{P}_t, P, \pi_t} \right\rangle \right]$$

$$= \mathbb{E} \left[ \underbrace{ \sum_{t=1}^T \sum_{s \neq s_L} \sum_{a \in A} \left( q^{P, \pi_t}(s, a) - q^{P, \pi^\star}(s, a) \right) (\ell_t(s, a) - \alpha_t(s, a)) }_{\text{TERM } 1(\pi^\star)} \right]$$

$$+ \mathbb{E} \left[ \underbrace{ \sum_{t=1}^T \sum_{s \neq s_L} \sum_{a \in A} \left( q^{P, \pi_t}(s, a) - q^{P, \pi^\star}(s, a) \right) \sum_{s' \in S_{k(s)+1}} \left( P(s' | s, a) - \widehat{P}_t(s' | s, a) \right) V^{\widehat{P}_t, \pi^\star}(s'; \alpha_t) }_{\text{TERM } 2(\pi^\star)} \right]$$

$$+ \mathbb{E} \left[ \underbrace{ \sum_{t=1}^T \left\langle q^{P, \pi_t}, \mathcal{Z}_{\alpha_t}^{\widehat{P}_t, P, \pi^\star} - \mathcal{Z}_{\alpha_t}^{\widehat{P}_t, P, \pi_t} \right\rangle }_{\text{TERM } 3(\pi^\star)} \right].$$

We will bound these terms with some self-bounding quantities in Lemma C.6.2, Lemma C.6.3 and Lemma C.6.4 respectively. In these proofs, we will follow the idea of Lemma D.1.1 to first bound these terms conditioning on the events $\mathcal{E}_{\text{EST}}$ and $\mathcal{E}_{\text{CON}}$ defined in Proposition D.5.2 and Eq. (4), while ensuring that these terms are always bounded by $\mathcal{O}\left( |S|^2 |A| T^2 \right)$ in the worst case.

### C.6.1 Bounding Term 1

**Lemma C.6.2.** *For any $\delta \in (0, 1)$ and any mapping $\pi^\star : S \to A$, Algorithm 4 ensures that* $\mathbb{E}\left[ \text{TERM } 1(\pi^\star) \right]$ *is bounded by*

$$\mathcal{O}\left( \mathbb{S}_1 \left( L^2 |S|^2 |A| \log^2(\iota) \right) + \delta |S|^2 |A| T^2 + \left( C^P + 1 \right) L^2 |S|^4 |A| \log^2(\iota) \right).$$

*Proof.* Clearly, we have $\alpha_t(s, a) \leq |S| T$ for every state-action pair $(s, a)$, due to the fact that $u_t(s) \geq 1/|S|T$ (Lemma D.2.8). Therefore, we have $\text{TERM } 1(\pi^\star) \leq |S|^2 |A| T^2$ holds always. In the remaining, our main goal is to bound $\text{TERM } 1(\pi^\star)$ conditioning on $\mathcal{E}_{\text{CON}} \wedge \mathcal{E}_{\text{EST}}$.

For any state-action pair $(s, a) \in S \times A$ and episode $t$, we have

$$\ell_t(s, a) - \alpha_t(s, a) = \frac{\left( u_t(s, a) - q^{P, \pi_t}(s, a) \right) \ell_t(s, a)}{u_t(s, a)} = \frac{\left( u_t(s) - q^{P, \pi_t}(s) \right) \ell_t(s, a)}{u_t(s)} \geq 0,$$

conditioning on the event $\mathcal{E}_{\text{CON}}$.

Therefore, under event $\mathcal{E}_{\text{CON}} \wedge \mathcal{E}_{\text{EST}}$, TERM $1(\pi^\star)$ is bounded by

$$\sum_{t=1}^{T} \sum_{s \neq s_L} \sum_{a \in A} \left( q^{P,\pi_t}(s,a) - q^{P,\pi^\star}(s,a) \right) \left( \ell_t(s,a) - \alpha_t(s,a) \right)$$

$$\leq \sum_{t=1}^{T} \sum_{s \neq s_L} \sum_{a \in A} \left[ q^{P,\pi_t}(s,a) - q^{P,\pi^\star}(s,a) \right]_+ \frac{\left( u_t(s) - q^{P,\pi_t}(s) \right) \ell_t(s,a)}{u_t(s)}$$

$$\leq \sum_{t=1}^{T} \sum_{s \neq s_L} \sum_{a \in A} \frac{\left[ q^{P,\pi_t}(s,a) - q^{P,\pi^\star}(s,a) \right]_+}{u_t(s)} \cdot \left| u_t(s) - q^{P,\pi_t}(s) \right|$$

$$= \mathcal{O}\left( \sqrt{ L|S|^2|A| \log^2(\iota) \sum_{t=1}^{T} \sum_{s \neq s_L} q^{P,\pi_t}(s) \cdot \left( \sum_a \frac{\left[ q^{P,\pi_t}(s,a) - q^{P,\pi^\star}(s,a) \right]_+}{u_t(s)} \right)^2 } \right)$$

$$+ \mathcal{O}\left( \left( C^{\mathsf{P}} + \log(\iota) \right) L^2 |S|^4 |A| \log(\iota) \right)$$

$$\leq \mathcal{O}\left( \sqrt{ L|S|^2|A| \log^2(\iota) \sum_{t=1}^{T} \sum_{s \neq s_L} \sum_{a \in A} \left[ q^{P,\pi_t}(s,a) - q^{P,\pi^\star}(s,a) \right]_+ } \right)$$

$$+ \mathcal{O}\left( \left( C^{\mathsf{P}} + \log(\iota) \right) L^2 |S|^4 |A| \log(\iota) \right)$$

$$\leq \mathcal{O}\left( \sqrt{ L^2|S|^2|A| \log^2(\iota) \sum_{t=1}^{T} \sum_{s \neq s_L} \sum_{a \neq \pi^\star(s)} q^{P,\pi_t}(s,a) + \left( C^{\mathsf{P}} + \log(\iota) \right) L^2 |S|^4 |A| \log(\iota) } \right)$$

$$\leq \mathcal{O}\left( \sqrt{ L^2|S|^2|A| \log^2(\iota) \sum_{t=1}^{T} \sum_{s \neq s_L} \sum_{a \neq \pi^\star(s)} q^{P_t,\pi_t}(s,a) + \left( C^{\mathsf{P}} + 1 \right) L^2 |S|^4 |A| \log^2(\iota) } \right),$$

where the first step follows from the non-negativity of $\ell_t(s,a) - \alpha_t(s,a)$; the third step applies Lemma D.5.4 with $G = 1$ as $\sum_a \left[ q^{P,\pi_t}(s,a) - q^{P,\pi^\star}(s,a) \right]_+ \leq q^{P,\pi_t}(s) \leq u_t(s)$; the fifth step follows from the fact that $\sum_{s \neq s_L} \sum_{a \in A} \left[ q^{P,\pi_t}(s,a) - q^{P,\pi^\star}(s,a) \right]_+ \leq 2L \sum_{s \neq s_L} \sum_{a \neq \pi^\star(s)} q^{P,\pi_t}(s,a)$ according to Corollary D.3.5; the last step applies Corollary D.3.6.

Applying Lemma D.1.1 with event $\mathcal{E}_{\text{CON}} \wedge \mathcal{E}_{\text{EST}}$ yields that

$$\mathbb{E}\left[ \text{TERM } 1(\pi^\star) \right] = \mathcal{O}\left( \mathbb{E}\left[ \sqrt{ L^2|S|^2|A| \log^2(\iota) \sum_{t=1}^{T} \sum_{s \neq s_L} \sum_{a \neq \pi^\star(s)} q^{P_t,\pi_t}(s,a) } \right] \right)$$

$$+ \mathcal{O}\left( \left( C^{\mathsf{P}} + \log(\iota) \right) L^2 |S|^4 |A| \log(\iota) + \delta |S|^2 |A| T^2 \right)$$

$$= \mathcal{O}\left( \mathbb{S}_1 \left( L^2|S|^2|A| \log^2(\iota) \right) + \delta |S|^2 |A| T^2 + \left( C^{\mathsf{P}} + \log(\iota) \right) L^2 |S|^4 |A| \log(\iota) \right).$$

$\square$

### C.6.2 Bounding Term 2

**Lemma C.6.3.** *For any $\delta \in (0,1)$ and any mapping $\pi^\star : S \to A$, Algorithm 4 ensures that*

$$\mathbb{E}\left[ \text{TERM } 2(\pi^\star) \right] = \mathcal{O}\left( \mathbb{S}_1 \left( L|S|^2|A| \log^2(\iota) \right) + \delta L|S|^2|A|T + L|S|^2|A| \left( C^{\mathsf{P}} + \log(\iota) \right) \log(T) \right).$$

*Proof.* Suppose that $\mathcal{E}_{\text{CON}} \wedge \mathcal{E}_{\text{EST}}$ occurs. We have

$$P(s'|s,a) \geq \widehat{P}_t(s'|s,a) = \widetilde{P}_{i(t)}(s'|s,a), \forall (s,a,s') \in W_k, k = 0, \dots, L-1.$$

Therefore, we can show that

$$0 \leq \sum_{s' \in S_{k(s)+1}} \left( P(s'|s,a) - \widehat{P}_t(s'|s,a) \right) V^{\widehat{P}_t, \pi^\star}(s'; \alpha_t)$$

$$= \mathcal{O}\left(L\left(\sqrt{\frac{|S|\log(\iota)}{\widehat{m}_{i(t)}(s,a)}} + \frac{|S|\left(C^{\mathsf{P}}+\log(\iota)\right)}{\widehat{m}_{i(t)}(s,a)}\right)\right),$$

where the last step follows from the definition of optimistic transition and the fact that $\alpha_t(s,a) \leq 1$.

By direct calculation, we have $\text{TERM }2(\pi^\star)$ bounded by

$$\mathcal{O}\left(L\sum_{s\neq s_L}\sum_{a\in A}\sum_{t=1}^{T}\left[q^{P,\pi_t}(s,a) - q^{P,\pi^\star}(s,a)\right]_+\left(\sqrt{\frac{|S|\log(\iota)}{\widehat{m}_{i(t)}(s,a)}} + \frac{|S|\left(C^{\mathsf{P}}+\log(\iota)\right)}{\widehat{m}_{i(t)}(s,a)}\right)\right)$$

$$\leq \mathcal{O}\left(L\sum_{s\neq s_L}\sum_{a\in A}\sum_{t=1}^{T}\left[q^{P,\pi_t}(s,a) - q^{P,\pi^\star}(s,a)\right]_+\sqrt{\frac{|S|\log(\iota)}{\widehat{m}_{i(t)}(s,a)}}\right)$$

$$+ \mathcal{O}\left(L|S|\sum_{s\neq s_L}\sum_{a\in A}\sum_{t=1}^{T}q^{P,\pi_t}(s,a)\left(\frac{\left(C^{\mathsf{P}}+\log(\iota)\right)}{\widehat{m}_{i(t)}(s,a)}\right)\right)$$

$$= \mathcal{O}\left(L\sum_{s\neq s_L}\sum_{a\in A}\sum_{t=1}^{T}\left[q^{P,\pi_t}(s,a) - q^{P,\pi^\star}(s,a)\right]_+\sqrt{\frac{|S|\log(\iota)}{\widehat{m}_{i(t)}(s,a)}} + L|S|^2|A|\left(C^{\mathsf{P}}+\log(\iota)\right)\log(T)\right),$$

where the last step follows from Lemma D.5.3.

Moreover, we have

$$\sum_{s\neq s_L}\sum_{a\in A}\sum_{t=1}^{T}\left[q^{P,\pi_t}(s,a) - q^{P,\pi^\star}(s,a)\right]_+\sqrt{\frac{|S|\log(\iota)}{\widehat{m}_{i(t)}(s,a)}}$$

$$\leq \sum_{s\neq s_L}\sum_{a\in A}\sum_{t=1}^{T}\sqrt{\left[q^{P,\pi_t}(s,a) - q^{P,\pi^\star}(s,a)\right]_+}\cdot\sqrt{\frac{q^{P,\pi_t}(s,a)|S|\log(\iota)}{\widehat{m}_{i(t)}(s,a)}}$$

$$\leq \sqrt{\sum_{s\neq s_L}\sum_{a\in A}\sum_{t=1}^{T}\left[q^{P,\pi_t}(s,a) - q^{P,\pi^\star}(s,a)\right]_+}\cdot\sqrt{\sum_{s\neq s_L}\sum_{a\in A}\sum_{t=1}^{T}\frac{q^{P,\pi_t}(s,a)|S|\log(\iota)}{\widehat{m}_{i(t)}(s,a)}}$$

$$= \mathcal{O}\left(\sqrt{|S|^2|A|\log^2(\iota)\sum_{s\neq s_L}\sum_{a\in A}\sum_{t=1}^{T}\left[q^{P,\pi_t}(s,a) - q^{P,\pi^\star}(s,a)\right]_+}\right)$$

$$\leq \mathcal{O}\left(\sqrt{L|S|^2|A|\log^2(\iota)\sum_{t=1}^{T}\sum_{s\neq s_L}\sum_{a\neq\pi^\star(s)}q^{P,\pi_t}(s,a)}\right)$$

$$\leq \mathcal{O}\left(\sqrt{L|S|^2|A|\log^2(\iota)\sum_{t=1}^{T}\sum_{s\neq s_L}\sum_{a\neq\pi^\star(s)}q^{P_t,\pi_t}(s,a)} + \sqrt{L^2|S|^2|A|\log^2 C^{\mathsf{P}}}\right),$$

where the second step uses the Cauchy-Schwarz inequality; the third step applies Lemma D.5.3; the fifth step follows from Corollary D.3.5; the last step uses Corollary D.3.6 and the fact that $\sqrt{x+y} \leq \sqrt{x} + \sqrt{y}$ for any $x, y \geq 0$.

Finally, applying Lemma D.1.1 finishes the proof. $\qquad\square$

### C.6.3 Bounding Term 3

**Lemma C.6.4.** *For any $\delta \in (0,1)$ and the policy $\pi^\star$, Algorithm 4 ensures that*

$$\mathbb{E}\left[\text{TERM }3(\pi^\star)\right] = \mathcal{O}\left(\mathbb{S}_1\left(L^3|S|^2|A|\log^2(\iota)\right) + \delta L|S|^2|A|T + \left(C^{\mathsf{P}}+\log(\iota)\right)L^2|S|^4|A|\log(\iota)\right).$$

*Proof.* Suppose that $\mathcal{E}_{\text{CON}} \wedge \mathcal{E}_{\text{EST}}$ occurs. We first have:

$$\left\langle q^{P,\pi_t}, \mathcal{Z}_{\alpha_t}^{\widehat{P}_t,P,\pi^\star} - \mathcal{Z}_{\alpha_t}^{\widehat{P}_t,P,\pi_t}\right\rangle$$

$$= \sum_{s \neq s_L} \sum_{a \in A} q^{P, \pi_t}(s, a) \sum_{s' \in S_{k(s)+1}} \left( \widehat{P}_t(s'|s, a) - P(s'|s, a) \right) \left( V^{\widehat{P}_t, \pi^\star}(s'; \alpha_t) - V^{\widehat{P}_t, \pi_t}(s'; \alpha_t) \right)$$

$$\leq 2 \sum_{s \neq s_L} \sum_{a \in A} \sum_{s' \in S_{k(s)+1}} q^{P, \pi_t}(s, a) B_{i(t)}(s, a, s') \left| V^{\widehat{P}_t, \pi^\star}(s'; \alpha_t) - V^{\widehat{P}_t, \pi_t}(s'; \alpha_t) \right|$$

$$\leq \mathcal{O} \left( \sum_{k=0}^{L-1} \sum_{(s, a, s') \in W_k} q^{P, \pi_t}(s, a) \sqrt{\frac{P(s'|s, a) \log(\iota)}{\widehat{m}_{i(t)}(s, a)}} \cdot \left| V^{\widehat{P}_t, \pi^\star}(s'; \alpha_t) - V^{\widehat{P}_t, \pi_t}(s'; \alpha_t) \right| \right)$$

$$+ \mathcal{O} \left( L|S| \sum_{s \neq s_L} \sum_{a \in A} q^{P, \pi_t}(s, a) \left( \frac{C^{\mathsf{P}} + \log(\iota)}{\widehat{m}_{i(t)}(s, a)} \right) \right),$$

where the last step follows from Lemma D.2.7 and the fact that $V^{\widehat{P}_t, \pi}(s'; \alpha_t) \leq L$ for any $\pi$.

By applying Lemma D.5.3, we have the second term bounded by $\mathcal{O} \left( (C^{\mathsf{P}} + \log(\iota)) L^2 |S|^4 |A| \log(\iota) \right)$. On the other hand, for the first term, we can bound $\left| V^{\widehat{P}_t, \pi^\star}(s'; \alpha_t) - V^{\widehat{P}_t, \pi_t}(s'; \alpha_t) \right|$ as

$$\left| V^{\widehat{P}_t, \pi^\star}(s'; \alpha_t) - V^{\widehat{P}_t, \pi_t}(s'; \alpha_t) \right|$$

$$\leq \sum_{u \neq s_L} \sum_{v \in A} q^{\widehat{P}_t, \pi_t}(u|s') \left| \pi_t(v|u) - \pi^\star(v|u) \right| Q^{\widehat{P}_t, \pi^\star}(u, v; \alpha_t)$$

$$\leq L \sum_{u \in S} \sum_{v \in A} q^{\widehat{P}_t, \pi_t}(u|s') \left| \pi_t(v|u) - \pi^\star(v|u) \right|$$

$$\leq \mathcal{O} \left( L \sum_{k=k(s')}^{L-1} \sum_{u \in S_k} \sum_{v \neq \pi^\star(u)} q^{\widehat{P}_t, \pi_t}(u, v|s') \right)$$

$$\leq \mathcal{O} \left( L \sum_{k=k(s')}^{L-1} \sum_{u \in S_k} \sum_{v \neq \pi^\star(u)} q^{P, \pi_t}(u, v|s') \right)$$

where the first step follows from Lemma D.3.2; the second step follows from the fact that $Q^{\widehat{P}_t, \pi^\star}(u, v; \alpha_t) \in [0, L]$; the third step uses the same reasoning as the proof of Corollary D.3.5; and the last step uses Corollary C.8.2.

Finally, we consider the following term

$$\sum_{t=1}^{T} \sum_{h=0}^{L-1} \sum_{(s, a, s') \in W_h} q^{P, \pi_t}(s, a) \sqrt{\frac{P(s'|s, a) \log(\iota)}{\widehat{m}_{i(t)}(s, a)}} \sum_{k=h+1}^{L-1} \sum_{u \in S_k} \sum_{v \neq \pi^\star(u)} q^{P, \pi_t}(u, v|s')$$

$$= \sum_{t=1}^{T} \sum_{h=0}^{L-1} \sum_{(s, a, s') \in W_h} \sum_{k=h+1}^{L-1} \sum_{u \in S_k} \sum_{v \neq \pi^\star(u)} \sqrt{\frac{q^{P, \pi_t}(s, a) q^{P, \pi_t}(u, v|s') \log(\iota)}{\widehat{m}_{i(t)}(s, a)}}$$

$$\cdot \sqrt{q^{P, \pi_t}(s, a) P(s'|s, a) q^{P, \pi_t}(u, v|s')}$$

$$\leq \sum_{h=0}^{L-1} \sqrt{\sum_{t=1}^{T} \sum_{(s, a, s') \in W_h} \sum_{k=h+1}^{L-1} \sum_{u \in S_k} \sum_{v \neq \pi^\star(u)} \frac{q^{P, \pi_t}(s, a) q^{P, \pi_t}(u, v|s') \log(\iota)}{\widehat{m}_{i(t)}(s, a)}}$$

$$\cdot \sqrt{\sum_{t=1}^{T} \sum_{(s, a, s') \in W_h} \sum_{k=h+1}^{L-1} \sum_{u \in S_k} \sum_{v \neq \pi^\star(u)} q^{P, \pi_t}(s, a) P(s'|s, a) q^{P, \pi_t}(u, v|s')}$$

$$\leq \sum_{h=0}^{L-1} \sqrt{|S_{h+1}| L \sum_{t=1}^{T} \sum_{s \in S_h} \sum_{a \in A} \frac{q^{P, \pi_t}(s, a) \log(\iota)}{\widehat{m}_{i(t)}(s, a)}} \cdot \sqrt{\sum_{t=1}^{T} \sum_{u \in S} \sum_{v \neq \pi^\star(u)} q^{P, \pi_t}(u, v)}$$

$$\leq \left( \sum_{h=0}^{L-1} \sqrt{L|S_{h+1}||S_h||A|\log^2(\iota)} \right) \cdot \sqrt{\sum_{t=1}^{T} \sum_{u \in S} \sum_{v \neq \pi^\star(u)} q^{P,\pi_t}(u,v)}$$

$$= \mathcal{O}\left( \sqrt{L|S|^2|A|\log^2(\iota) \sum_{t=1}^{T} \sum_{s \neq s_L} \sum_{a \neq \pi^\star(a)} q^{P,\pi_t}(s,a)} \right)$$

$$= \mathcal{O}\left( \sqrt{L|S|^2|A|\log^2(\iota) \sum_{t=1}^{T} \sum_{s \neq s_L} \sum_{a \neq \pi^\star(a)} q^{P_t,\pi_t}(s,a)} + \sqrt{C^{\mathsf{P}} \cdot L^2|S|^2|A|\log^2(\iota)} \right),$$

where the second step applies Cauchy-Schwarz inequality; the third step follows from the fact that $\sum_{s \in S_k} \sum_{a \in A} \sum_{s' \in S_{k(s)+1}} q^{P,\pi_t}(s,a)P(s'|s,a)q^{P,\pi_t}(u,v|s') = q^{P,\pi_t}(u,v)$; the fourth step follows from Lemma D.5.3 conditioning on the event $\mathcal{E}_{\mathrm{EST}}$; the fifth step uses the fact that $\sqrt{x+y} \leq \sqrt{x} + \sqrt{y}$ for $x, y \geq 0$; the last step follows from Corollary D.3.6. $\qquad\square$

## C.7 Proof of Lemma C.3.4

In this section we bound ESTREG using a learning rate that depends on $t$ and $(s,a)$, which is crucial to obtain a self-bounding quantity. Another key observation is that the estimated transition function is constant within each epoch, so we first bound ESTREG within one epoch before summing them.

Recall that $E_i$ is a set of episodes that belong to epoch $i$ and $N$ is the total number of epochs through $T$ episodes. By using the fact that $\widehat{P}_t = \widetilde{P}_i$ for episode $t$ belonging to epoch $i$, we make the following decomposition ESTREG($\pi^\star$)

$$\mathrm{ESTREG}(\pi^\star) = \mathbb{E}\left[ \sum_{i=1}^{N} \sum_{t \in E_i} \left\langle q^{\widetilde{P}_i,\pi_t} - q^{\widehat{P}_t,\pi^\star}, \widehat{\ell}_t - b_t \right\rangle \right]$$

$$\leq \mathbb{E}\left[ \sum_{i=1}^{N} \mathbb{E}_{t_i}\left[ \sum_{t \in E_i} \left\langle q^{\widetilde{P}_i,\pi_t} - q^{\widehat{P}_t,\pi^\star}, \widehat{\ell}_t - b_t \right\rangle \right] \right] = \mathbb{E}\left[ \sum_{i=1}^{N} \mathrm{ESTREG}_i(\pi^\star) \right].$$

This learning rate is defined in Definition 5.2 and restated below.

**Definition C.7.1.** *For any $t$, if it is the starting episode of an epoch, we set $\gamma_t(s,a) = 256L^2|S|$; otherwise, we set*

$$\gamma_{t+1}(s,a) = \gamma_t(s,a) + \frac{D\nu_t(s,a)}{2\gamma_t(s,a)},$$

*where $D = \frac{1}{\log(\iota)}$ and*

$$\nu_t(s,a) = q^{\widetilde{P}_{i(t)},\pi_t}(s,a)^2 \left( Q^{\widetilde{P}_{i(t)},\pi_t}(s,a;\widehat{\ell}_t) - V^{\widetilde{P}_{i(t)},\pi_t}(s;\widehat{\ell}_t) \right)^2. \tag{35}$$

Importantly, both $Q^{\widetilde{P}_{i(t)},\pi_t}(s,a;\widehat{\ell}_t)$ and $V^{\widetilde{P}_{i(t)},\pi_t}(s;\widehat{\ell}_t)$ can be computed, which ensures that the learning rate is properly defined.

### C.7.1 Properties of the Learning Rate

In this section, we prove key properties of $\nu_t(s,a)$ and of $\gamma_t(s,a)$. We first present some results in Lemma C.7.2 that are useful to bound $\nu_t(s,a)$, and then use these results to bound $\gamma_t(s,a)$.

**Lemma C.7.2.** *For any state-action pair $(s,a)$ and any episode $t$, it holds that*

$$q^{\widehat{P}_t,\pi_t}(s,a)Q^{\widehat{P}_t,\pi_t}(s,a;\widehat{\ell}_t) \leq L, \text{ and } q^{\widehat{P}_t,\pi_t}(s)V^{\widehat{P}_t,\pi_t}(s;\widehat{\ell}_t) \leq L, \ \forall (s,a) \in S \times A,$$

*which ensures $q^{\widehat{P}_t,\pi_t}(s,a)\left( Q^{\widehat{P}_t,\pi_t}(s,a;\widehat{\ell}_t) - V^{\widehat{P}_t,\pi_t}(s;\widehat{\ell}_t) \right) \in [-L, L]$. Suppose that the high-probability event $\mathcal{E}_{\mathrm{CON}}$ holds. Then, it further holds for all state-action pair $(s,a)$ that*

$$q^{\widehat{P}_t,\pi_t}(s,a)^2 \cdot \mathbb{E}_t\left[ \left( Q^{\widehat{P}_t,\pi_t}(s,a;\widehat{\ell}_t) - V^{\widehat{P}_t,\pi_t}(s;\widehat{\ell}_t) \right)^2 \right] \leq \mathcal{O}\left( L^2 q^{\widehat{P}_t,\pi_t}(s,a)(1 - \pi_t(a|s)) + L|S|C_t^{\mathsf{P}} \right),$$

*where $\widehat{P}_t$ here is the optimistic transition $\widetilde{P}_{i(t)}$ defined in Definition C.1.1.*

*Proof.* We first verify $q^{\widehat{P}_t,\pi_t}(s,a)Q^{\widehat{P}_t,\pi_t}(s,a;\widehat{\ell}_t) \le L$:

$$q^{\widehat{P}_t,\pi_t}(s,a)Q^{\widehat{P}_t,\pi_t}(s,a;\widehat{\ell}_t)$$

$$= q^{\widehat{P}_t,\pi_t}(s,a) \sum_{h=k(s)} \sum_{u \in S} \sum_{v \in A} q^{\widehat{P}_t,\pi_t}(u,v|s,a)\widehat{\ell}_t(u,v)$$

$$= q^{\widehat{P}_t,\pi_t}(s,a) \sum_{h=k(s)} \sum_{u \in S} \sum_{v \in A} q^{\widehat{P}_t,\pi_t}(u,v|s,a) \cdot \frac{\mathbb{I}_t(u,v)\ell_t(u,v)}{u_t(u,v)}$$

$$\le \sum_{h=k(s)} \sum_{u \in S_h} \sum_{v \in A} \mathbb{I}_t(u,v) \cdot \frac{q^{\widehat{P}_t,\pi_t}(s,a)q^{\widehat{P}_t,\pi_t}(u,v|s,a)}{u_t(u,v)}$$

$$\le \sum_{h=k(s)} \sum_{u \in S_h} \sum_{v \in A} \mathbb{I}_t(u,v) \le L,$$

where the second step follows from the definition of loss estimator, the fourth step follows from $q^{\widehat{P}_t,\pi_t}(s,a)q^{\widehat{P}_t,\pi_t}(u,v|s,a) \le q^{\widehat{P}_t,\pi_t}(u,v) \le u_t(u,v)$, and the last step uses the fact that $\sum_{u \in S_h} \sum_{v \in A} \mathbb{I}_t(u,v) = 1$. Following the same idea, we can show that $q^{\widehat{P}_t,\pi_t}(s)V^{\widehat{P}_t,\pi_t}(s;\widehat{\ell}_t) \le L$ as well.

Next, we have $\mathbb{E}_t\left[\left(Q^{\widehat{P}_t,\pi_t}(s,a;\widehat{\ell}_t) - V^{\widehat{P}_t,\pi_t}(s;\widehat{\ell}_t)\right)^2\right]$ bounded as

$$\mathbb{E}_t\left[\left(Q^{\widehat{P}_t,\pi_t}(s,a;\widehat{\ell}_t) - V^{\widehat{P}_t,\pi_t}(s;\widehat{\ell}_t)\right)^2\right]$$

$$= \mathbb{E}_t\left[\left(Q^{\widehat{P}_t,\pi_t}(s,a;\widehat{\ell}_t) - \pi_t(a|s)Q^{\widehat{P}_t,\pi_t}(s,a;\widehat{\ell}_t) - \sum_{a' \ne a} \pi_t(a'|s)Q^{\widehat{P}_t,\pi_t}(s,a';\widehat{\ell}_t)\right)^2\right]$$

$$\le 2 \cdot \mathbb{E}_t\left[\left(Q^{\widehat{P}_t,\pi_t}(s,a;\widehat{\ell}_t) - \pi_t(a|s)Q^{\widehat{P}_t,\pi_t}(s,a;\widehat{\ell}_t)\right)^2\right]$$

$$+ 2 \cdot \mathbb{E}_t\left[\left(\sum_{a' \ne a} \pi_t(a'|s)Q^{\widehat{P}_t,\pi_t}(s,a';\widehat{\ell}_t)\right)^2\right]$$

$$= 2(1 - \pi_t(a|s))^2 \mathbb{E}_t\left[\left(Q^{\widehat{P}_t,\pi_t}(s,a;\widehat{\ell}_t)\right)^2\right] + 2\mathbb{E}_t\left[\left(\sum_{a' \ne a} \pi_t(a'|s)Q^{\widehat{P}_t,\pi_t}(s,a';\widehat{\ell}_t)\right)^2\right], \quad (36)$$

where the second step follows from the fact that $(x + y)^2 \le 2\left(x^2 + y^2\right)$ for any $x, y \in \mathbb{R}$.

By direct calculation, we have

$$\mathbb{E}_t\left[\left(Q^{\widehat{P}_t,\pi_t}(s,a;\widehat{\ell}_t)\right)^2\right]$$

$$= \mathbb{E}_t\left[\left(\sum_{k=k(s)}^{L-1} \sum_{x \in S_k} \sum_{y \in A} q^{\widehat{P}_t,\pi_t}(x,y|s,a)\widehat{\ell}_t(x,y)\right)^2\right]$$

$$\le L \cdot \mathbb{E}_t\left[\sum_{k=k(s)}^{L-1} \left(\sum_{x \in S_k} \sum_{y \in A} q^{\widehat{P}_t,\pi_t}(x,y|s,a)\widehat{\ell}_t(x,y)\right)^2\right]$$

$$= L \cdot \mathbb{E}_t\left[\sum_{k=k(s)}^{L-1} \sum_{x \in S_k} \sum_{y \in A} q^{\widehat{P}_t,\pi_t}(x,y|s,a)^2\widehat{\ell}_t(x,y)^2\right]$$

$$\leq L \cdot \sum_{k=k(s)}^{L-1} \sum_{x \in S_k} \sum_{y \in A} q^{\widehat{P}_t, \pi_t}(x, y|s, a)^2 \cdot \left( \frac{q^{P_t, \pi_t}(x, y)}{u_t(x, y)^2} \right)$$

$$= \frac{L}{q^{\widehat{P}_t, \pi_t}(s, a)} \cdot \sum_{k=k(s)}^{L-1} \sum_{x \in S_k} \sum_{y \in A} q^{\widehat{P}_t, \pi_t}(x, y|s, a) \left( \frac{q^{\widehat{P}_t, \pi_t}(x, y|s, a) q^{\widehat{P}_t, \pi_t}(s, a)}{u_t(s, a)} \right) \cdot \left( \frac{q^{P_t, \pi_t}(x, y)}{u_t(s, a)} \right)$$

$$\leq \frac{L}{q^{\widehat{P}_t, \pi_t}(s, a)} \cdot \sum_{k=k(s)}^{L-1} \sum_{x \in S_k} \sum_{y \in A} q^{\widehat{P}_t, \pi_t}(x, y|s, a) \left( \frac{q^{P_t, \pi_t}(x, y)}{u_t(s, a)} \right)$$

$$\leq \frac{L}{q^{\widehat{P}_t, \pi_t}(s, a)} \cdot \sum_{k=k(s)}^{L-1} \sum_{x \in S_k} \sum_{y \in A} q^{\widehat{P}_t, \pi_t}(x, y|s, a) \left( \frac{q^{P, \pi_t}(x, y)}{u_t(x, y)} + \frac{C_t^{\mathsf{P}}}{u_t(x, y)} \right), \tag{37}$$

where the second step uses Cauchy-Schwarz inequality; the third step uses the fact that $\widehat{\ell}_t(s, a) \cdot \widehat{\ell}_t(s', a') = 0$ for any $(s, a) \neq (s', a')$; the fourth step takes the conditional expectation of $\widehat{\ell}_t(x, y)^2$; the sixth step follows from the fact that $q^{\widehat{P}_t, \pi_t}(x, y|s, a) q^{\widehat{P}_t, \pi_t}(s, a) \leq q^{\widehat{P}_t, \pi_t}(x, y) \leq u_t(x, y)$ according to the definition of upper occupancy bound; the last step follows from Corollary D.3.6.

Similarly, for the second term in Eq. (36), we have

$$\mathbb{E}_t \left[ \left( \sum_{b \neq a} \pi_t(b|s) Q^{\widehat{P}_t, \pi_t}(s, b; \widehat{\ell}_t) \right)^2 \right]$$

$$\leq L \cdot \mathbb{E}_t \left[ \sum_{k=k(s)}^{L-1} \left( \sum_{x \in S_k} \sum_{y \in A} \left( \sum_{b \neq a} \pi_t(b|s) q^{\widehat{P}_t, \pi_t}(x, y|s, b) \right) \widehat{\ell}_t(x, y) \right)^2 \right]$$

$$= L \cdot \mathbb{E}_t \left[ \sum_{k=k(s)}^{L-1} \sum_{x \in S_k} \sum_{y \in A} \left( \sum_{b \neq a} \pi_t(b|s) q^{\widehat{P}_t, \pi_t}(x, y|s, b) \right)^2 \widehat{\ell}_t(x, y)^2 \right]$$

$$\leq L \cdot \sum_{k=k(s)}^{L-1} \sum_{x \in S_k} \sum_{y \in A} \left( \sum_{b \neq a} \pi_t(b|s) q^{\widehat{P}_t, \pi_t}(x, y|s, b) \right)^2 \left( \frac{q^{P_t, \pi_t}(x, y)}{u_t(x, y)^2} \right)$$

$$\leq \frac{L}{q^{\widehat{P}_t, \pi_t}(s)} \sum_{k=k(s)}^{L-1} \sum_{x \in S_k} \sum_{y \in A} \left( \sum_{b \neq a} \pi_t(b|s) q^{\widehat{P}_t, \pi_t}(x, y|s, b) \right) \left( \frac{q^{P_t, \pi_t}(x, y)}{u_t(x, y)} \right)$$

$$\leq \frac{L}{q^{\widehat{P}_t, \pi_t}(s)} \sum_{b \neq a} \pi_t(b|s) \left( \sum_{k=k(s)}^{L-1} \sum_{x \in S_k} \sum_{y \in A} q^{\widehat{P}_t, \pi_t}(x, y|s, b) \left( \frac{q^{P, \pi_t}(x, y)}{u_t(x, y)} + \frac{C_t^{\mathsf{P}}}{u_t(x, y)} \right) \right), \tag{38}$$

where the first three steps are following the same idea of previous analysis; the fourth step uses the fact that $\sum_{b \neq a} q^{\widehat{P}_t, \pi_t}(s) \pi_t(b|s) q^{\widehat{P}_t, \pi_t}(x, y|s, b) \leq q^{\widehat{P}_t, \pi_t}(x, y)$; the last step follows from Corollary D.3.6 as well.

Conditioning on the event $\mathcal{E}_{\mathrm{CON}}$, we have the term in Eq. (37) further bounded as

$$\frac{L}{q^{\widehat{P}_t, \pi_t}(s, a)} \cdot \sum_{k=k(s)}^{L-1} \sum_{x \in S_k} \sum_{y \in A} q^{\widehat{P}_t, \pi_t}(x, y|s, a) \left( \frac{q^{P, \pi_t}(x, y)}{u_t(x, y)} + \frac{C_t^{\mathsf{P}}}{u_t(x, y)} \right)$$

$$\leq \frac{L}{q^{\widehat{P}_t, \pi_t}(s, a)} \cdot \sum_{k=k(s)}^{L-1} \sum_{x \in S_k} \sum_{y \in A} q^{\widehat{P}_t, \pi_t}(x, y|s, a) \left( 1 + \frac{C_t^{\mathsf{P}}}{u_t(x, y)} \right)$$

$$= \frac{L}{q^{\widehat{P}_t, \pi_t}(s, a)} \cdot \sum_{k=k(s)}^{L-1} \sum_{x \in S_k} \sum_{y \in A} q^{\widehat{P}_t, \pi_t}(x, y|s, a)$$

$$+ \frac{L}{q^{\widehat{P}_t,\pi_t}(s,a)} \cdot \sum_{k=k(s)}^{L-1} \sum_{x \in S_k} \sum_{y \in A} q^{\widehat{P}_t,\pi_t}(x,y|s,a) \left( \frac{C_t^{\mathsf{P}}}{u_t(x,y)} \right)$$

$$\leq \frac{L^2}{q^{\widehat{P}_t,\pi_t}(s,a)} + \frac{LC_t^{\mathsf{P}}}{q^{\widehat{P}_t,\pi_t}(s,a)} \cdot \sum_{k=k(s)}^{L-1} \sum_{x \in S_k} \sum_{y \in A} \left( \frac{q^{\widehat{P}_t,\pi_t}(x,y|s,a)}{u_t(x,y)} \right), \tag{39}$$

where the second step follows from the fact that $q^{P,\pi_t}(x,y) \leq u_t(x,y)$ as $P \in \mathcal{P}_{i(t)}$ according to the event $\mathcal{E}_{\mathrm{CON}}$; the last step follows from the fact that $\sum_{x \in S_k} \sum_{y \in A} q^{\widehat{P}_t,\pi_t}(x,y|s,a) \leq 1$.

Following the same argument, for the term in Eq. (38), we have

$$\frac{L}{q^{\widehat{P}_t,\pi_t}(s)} \sum_{b \neq a} \pi_t(b|s) \left( \sum_{k=k(s)}^{L-1} \sum_{x \in S_k} \sum_{y \in A} q^{\widehat{P}_t,\pi_t}(x,y|s,b) \left( \frac{q^{P,\pi_t}(x,y)}{u_t(x,y)} + \frac{C_t^{\mathsf{P}}}{u_t(x,y)} \right) \right)$$

$$\leq \frac{L}{q^{\widehat{P}_t,\pi_t}(s)} \sum_{b \neq a} \pi_t(b|s) \left( L + \sum_{k=k(s)}^{L-1} \sum_{x \in S_k} \sum_{y \in A} \left( \frac{q^{\widehat{P}_t,\pi_t}(x,y|s,b) C_t^{\mathsf{P}}}{u_t(x,y)} \right) \right)$$

$$\leq \frac{L^2 (1 - \pi_t(a|s))}{q^{\widehat{P}_t,\pi_t}(s)} + \frac{LC_t^{\mathsf{P}}}{q^{\widehat{P}_t,\pi_t}(s)} \sum_{b \neq a} \pi_t(b|s) \sum_{k=k(s)}^{L-1} \sum_{x \in S_k} \sum_{y \in A} \left( \frac{q^{\widehat{P}_t,\pi_t}(x,y|s,b)}{u_t(x,y)} \right). \tag{40}$$

Plugging Eq. (39) and Eq. (40) into Eq. (36), we have

$$q^{\widehat{P}_t,\pi_t}(s,a)^2 \cdot \mathbb{E}_t \left[ \left( Q^{\widehat{P}_t,\pi_t}(s,a;\widehat{\ell}_t) - V^{\widehat{P}_t,\pi_t}(s;\widehat{\ell}_t) \right)^2 \right]$$

$$\leq 2q^{\widehat{P}_t,\pi_t}(s,a)^2 (1 - \pi_t(a|s))^2 \left( \frac{L^2}{q^{\widehat{P}_t,\pi_t}(s,a)} + \frac{LC_t^{\mathsf{P}}}{q^{\widehat{P}_t,\pi_t}(s,a)} \cdot \sum_{k=k(s)}^{L-1} \sum_{x \in S_k} \sum_{y \in A} \left( \frac{q^{\widehat{P}_t,\pi_t}(x,y|s,a)}{u_t(x,y)} \right) \right)$$

$$+ 2q^{\widehat{P}_t,\pi_t}(s,a)^2 \left( \frac{L^2 (1 - \pi_t(a|s))}{q^{\widehat{P}_t,\pi_t}(s)} + \frac{LC_t^{\mathsf{P}}}{q^{\widehat{P}_t,\pi_t}(s)} \sum_{b \neq a} \pi_t(b|s) \sum_{k=k(s)}^{L-1} \sum_{x \in S_k} \sum_{y \in A} \left( \frac{q^{\widehat{P}_t,\pi_t}(x,y|s,b)}{u_t(x,y)} \right) \right)$$

$$\leq \mathcal{O} \left( L^2 q^{\widehat{P}_t,\pi_t}(s,a) (1 - \pi_t(a|s)) \right)$$

$$+ \mathcal{O} \left( LC_t^{\mathsf{P}} \cdot q^{\widehat{P}_t,\pi_t}(s,a) \sum_{k=k(s)}^{L-1} \sum_{x \in S_k} \sum_{y \in A} \left( \frac{q^{\widehat{P}_t,\pi_t}(x,y|s,a)}{u_t(x,y)} \right) \right)$$

$$+ \mathcal{O} \left( LC_t^{\mathsf{P}} \cdot q^{\widehat{P}_t,\pi_t}(s) \sum_{b \neq a} \pi_t(b|s) \sum_{k=k(s)}^{L-1} \sum_{x \in S_k} \sum_{y \in A} \left( \frac{q^{\widehat{P}_t,\pi_t}(x,y|s,b)}{u_t(x,y)} \right) \right)$$

$$\leq \mathcal{O} \left( L^2 q^{\widehat{P}_t,\pi_t}(s,a) (1 - \pi_t(a|s)) \right)$$

$$+ \mathcal{O} \left( LC_t^{\mathsf{P}} \cdot \sum_{k=k(s)}^{L-1} \sum_{x \in S_k} \sum_{y \in A} \left( \frac{q^{\widehat{P}_t,\pi_t}(x,y|s,a) q^{\widehat{P}_t,\pi_t}(s,a)}{u_t(x,y)} \right) \right)$$

$$+ \mathcal{O} \left( LC_t^{\mathsf{P}} \cdot \sum_{k=k(s)}^{L-1} \sum_{x \in S_k} \sum_{y \in A} \left( \frac{\sum_{b \neq a} q^{\widehat{P}_t,\pi_t}(s) \pi_t(b|s) q^{\widehat{P}_t,\pi_t}(x,y|s,b)}{u_t(x,y)} \right) \right)$$

$$\leq \mathcal{O} \left( L^2 q^{\widehat{P}_t,\pi_t}(s,a) (1 - \pi_t(a|s)) + L|S|C_t^{\mathsf{P}} \right),$$

where the third step follows from the facts that $q^{\widehat{P}_t,\pi_t}(x,y|s,a) q^{\widehat{P}_t,\pi_t}(s,a) \leq q^{\widehat{P}_t,\pi_t}(x,y) \leq u_t(x,y)$ for any $(s,a), (x,y) \in S \times A$, and $\sum_{b \neq a} q^{\widehat{P}_t,\pi_t}(s) \pi_t(b|s) q^{\widehat{P}_t,\pi_t}(x,y|s,b) \leq q^{\widehat{P}_t,\pi_t}(x,y)$ similarly.

$\square$

The first part of Lemma C.7.2 ensures that $\nu_t(s,a) \in [0, L^2]$, which can be used to bound the growth of the learning rate.

**Proposition C.7.3.** *The learning rate $\gamma_t$ defined in Definition 5.2 satisfies for any state-action pair $(s, a)$:*

$$\gamma_t(s,a) \leq \sqrt{D \sum_{j=t_{i(t)}}^{t-1} \nu_j(s,a) + \gamma_{t_{i(t)}}(s,a)}$$

*where $i(t)$ is the epoch to which episode $t$ belongs and $t_i$ is the first episode of epoch $i$.*

*Proof.* We prove this statement by induction on $t$. The equation trivially holds for $t = t_{i(t)}$ which is the first round of epoch $i(t)$. For the induction step, we first note that $D\nu_t(s,a) \in [0, L^2]$. We introduce some notations to simplify the proof: we use $c$ to denote $D\nu_t(s,a)$, $x$ to denote $\gamma_t(s,a)$, and the induction hypothesis is $x \leq \sqrt{S} + \gamma$ where $S = D \sum_{j=t_{i(t)}}^{t-1} \nu_j(s,a)$ and $\gamma = \gamma_{t_{i(t)}}(s,a)$. Proving the induction step is the same as proving:

$$x + \frac{c}{2x} \leq \sqrt{S+c} + \gamma. \tag{41}$$

First, we can verify that $f(x) = x + \frac{c}{2x}$ is an increasing function of $x$ for $x \geq \sqrt{c/2}$. As $c \in [0, L^2]$ and $x \geq \gamma \geq L$, we can use the induction hypothesis $x \leq \sqrt{S} + \gamma$ to upper bound $x$ in the left-hand side of Eq. (41), and we get:

$$
\begin{aligned}
x + \frac{c}{2x} &= \frac{x^2 + c/2}{x} \\
&\leq \frac{S + 2\gamma\sqrt{S} + \gamma^2 + c/2}{\sqrt{S} + \gamma} \\
&= \frac{S + \gamma\sqrt{S} + c/2}{\sqrt{S} + \gamma} + \frac{\gamma\sqrt{S} + \gamma^2}{\sqrt{S} + \gamma} \\
&= \frac{\sqrt{S} + \gamma}{\sqrt{S} + \gamma}\left(\sqrt{S} + \frac{c}{2\sqrt{S} + 2\gamma}\right) + \frac{\gamma\sqrt{S} + \gamma^2}{\sqrt{S} + \gamma} \\
&= \left(\sqrt{S} + \frac{c}{2\sqrt{S} + 2\gamma}\right) + \gamma \\
&\leq \left(\sqrt{S} + \frac{c}{2\sqrt{S+c}}\right) + \gamma \\
&\leq \sqrt{S+c} + \gamma,
\end{aligned}
$$

where the second to last step follows from $c \leq L^2$ and $\gamma \geq L$, which ensures that $2\sqrt{S} + 2\sqrt{\gamma} \geq 2\sqrt{S + \gamma^2} \geq 2\sqrt{S+c}$. The last step follows from the concavity of the square-root function which ensures that $\sqrt{S} \leq \sqrt{S+c} - \frac{c}{2\sqrt{S+c}}$. $\qquad\square$

Then, we can use the second part of Lemma C.7.2 to continue the bound of Proposition C.7.3 and derive a bound that only depends on the suboptimal actions.

**Proposition C.7.4.** *If $\nu_t$ is defined as in Definition 5.2, we have for any deterministic policy $\pi : S \to A$ and any state $s \neq s_L$:*

$$
\mathbb{E}_{t_i}\left[\sum_{a\in A}\sqrt{\sum_{t=t_i}^{t_{i+1}-1}\nu_t(s,a)}\right] \leq \mathcal{O}\left(\sum_{a\neq\pi(s)}\sqrt{\mathbb{E}_{t_i}\left[L^2\sum_{t=t_i}^{t_{i+1}-1}q^{P_t,\pi_t}(s,a)\right]}\right)
$$
$$
+ \mathcal{O}\left(\delta L^2 |S||A|(t_{i+1}-t_i) + \sqrt{L|S||A|^2\sum_{t=t_i}^{t_{i+1}-1}C_t^P}\right).
$$

*Proof.* For each epoch $i$ and state-action pair $(s, a)$, we have:

$$\mathbb{E}_{t_i}\left[\sqrt{\sum_{t=t_i}^{t_{i+1}-1} \nu_t(s,a)}\right] \leq \sqrt{\mathbb{E}_{t_i}\left[\sum_{t=t_i}^{t_{i+1}-1} \nu_t(s,a)\right]}$$

$$= \mathcal{O}\left(\sqrt{\mathbb{E}_{t_i}\left[\sum_{t=t_i}^{t_{i+1}-1} L^2\, q^{\widehat{P}_t,\pi_t}(s,a)(1-\pi_t(a|s)) + L|S|C_t^{\mathsf{P}}\right]}\right), \tag{42}$$

where the first step follows from Jensen's inequality, and the second step uses Lemma C.7.2.

Note that, for $a = \pi(s)$, we have:

$$q^{\widehat{P}_t,\pi_t}(s,\pi(s))(1-\pi_t(\pi(s)|s)) \leq q^{\widehat{P}_t,\pi_t}(s)\sum_{b\neq\pi(s)}\pi_t(b|s) \leq \sum_{b\neq\pi(s)} q^{\widehat{P}_t,\pi_t}(s,b).$$

Therefore, we have

$$\mathbb{E}_{t_i}\left[\sum_{a\in A}\sqrt{\sum_{t=t_i}^{t_{i+1}-1} \nu_t(s,a)}\right]$$

$$= \mathcal{O}\left(\sum_{a\in A}\sqrt{\mathbb{E}_{t_i}\left[\sum_{t=t_i}^{t_{i+1}-1} L^2\, q^{\widehat{P}_t,\pi_t}(s,a)(1-\pi_t(a|s)) + L|S|C_t^{\mathsf{P}}\right]}\right)$$

$$= \mathcal{O}\left(\sum_{a\in A}\sqrt{\mathbb{E}_{t_i}\left[L^2 \sum_{t=t_i}^{t_{i+1}-1} q^{\widehat{P}_t,\pi_t}(s,a)(1-\pi_t(a|s))\right]} + \sqrt{L|S|\sum_{t=t_i}^{t_{i+1}-1} C_t^{\mathsf{P}}}\right)$$

$$= \mathcal{O}\left(\sum_{a\neq\pi(s)}\sqrt{\mathbb{E}_{t_i}\left[L^2 \sum_{t=t_i}^{t_{i+1}-1} q^{\widehat{P}_t,\pi_t}(s,a)\right]} + \sqrt{L|S||A|^2 \sum_{t=t_i}^{t_{i+1}-1} C_t^{\mathsf{P}}}\right)$$

$$\leq \mathcal{O}\left(\delta L^2 |S||A|\,(t_{i+1}-1-t_i) + \sum_{a\neq\pi(s)}\sqrt{\mathbb{E}_{t_i}\left[L^2 \sum_{t=t_i}^{t_{i+1}-1} q^{P,\pi_t}(s,a)\right]} + \sqrt{L|S||A|^2 \sum_{t=t_i}^{t_{i+1}-1} C_t^{\mathsf{P}}}\right)$$

$$\leq \mathcal{O}\left(\delta L^2 |S||A|\,(t_{i+1}-1-t_i) + \sum_{a\neq\pi(s)}\sqrt{\mathbb{E}_{t_i}\left[L^2 \sum_{t=t_i}^{t_{i+1}-1} q^{P_t,\pi_t}(s,a)\right]} + \sqrt{L|S||A|^2 \sum_{t=t_i}^{t_{i+1}-1} C_t^{\mathsf{P}}}\right),$$

where the first step follows from Eq. (42); the second step uses the fact that $\sqrt{x+y} \leq \sqrt{x} + \sqrt{y}$ for $x, y \geq 0$; the fourth step follows from Lemma D.1.1 with the event $\mathcal{E}_{\mathrm{CON}}$ and Corollary C.8.2; the last step applies Corollary D.3.6. $\qquad\square$

### C.7.2 Bounding $\mathrm{ESTREG}_i(\pi^\star)$ for Varying Learning Rate

We now focus on bounding $\mathrm{ESTREG}_i(\pi^\star)$ for an individual epoch $i$.

**Notations for FTRL Analysis.** For any fixed epoch $i$ and any integer $t \in [t_i, t_{i+1}-1]$, we introduce the following notations.

$$F_t(q) = \left\langle q, \sum_{\tau=t_i}^{t-1}\left(\widehat{\ell}_\tau - b_\tau\right)\right\rangle + \phi_t(q)\,, \quad G_t(q) = \left\langle q, \sum_{\tau=t_i}^{t}\left(\widehat{\ell}_\tau - b_\tau\right)\right\rangle + \phi_t(q),$$

$$q_t = \operatorname*{argmin}_{q\in\Omega(\widetilde{P}_i)} F_t(q)\,, \quad \widetilde{q}_t = \operatorname*{argmin}_{q\in\Omega(\widetilde{P}_i)} G_t(q). \tag{43}$$

With these notations, we have $q_t = \widehat{q}_t = q^{\widehat{P}_t,\pi_t} = q^{\widetilde{P}_i,\pi_t}$. Also, according to the loss shifting technique (specifically Corollary D.4.2), $q_t$ and $\widetilde{q}_t$ can be equivalently written as

$$q_t = \operatorname*{argmin}_{q\in\Omega(\widetilde{P}_i)}\left\langle q, \sum_{\tau=t_i}^{t-1}(g_\tau - b_\tau)\right\rangle + \phi_t(q), \quad \widetilde{q}_t = \operatorname*{argmin}_{q\in\Omega(\widetilde{P}_i)}\left\langle q, \sum_{\tau=t_i}^{t}(g_\tau - b_\tau)\right\rangle + \phi_t(q), \tag{44}$$

where $g_t : S \times A \to \mathbb{R}$ is defined as

$$g_t(s,a) = Q^{\widehat{P}_t,\pi_t}(s,a;\widehat{\ell}_t) - V^{\widehat{P}_t,\pi_t}(s;\widehat{\ell}_t), \forall(s,a) \in S \times A. \tag{45}$$

Finally, We also define $u = q^{\widetilde{P}_i,\pi^\star}$ and

$$v = \left(1 - \frac{1}{T^2}\right)u + \frac{1}{T^2|A||S|}\sum_{s,a} q_{s,a}^{\max}, \tag{46}$$

where $q_{s,a}^{\max}$ is the occupancy measure associated with a policy that maximizes the probability of visiting $(s,a)$ given the optimistic transition $\widetilde{P}_i$. Then, we have $v \in \Omega(\widetilde{P}_i)$ because $u, q_{s,a}^{\max} \in \Omega(\widetilde{P}_i)$ for all $(s,a)$, and $\Omega(\widetilde{P}_i)$ is convex [Jin and Luo, 2020, Lemma 10].

Therefore, $\text{ESTREG}_i(\pi^\star)$ can be rewritten as:

$$
\begin{aligned}
\text{ESTREG}_i(\pi^\star) &= \mathbb{E}_{t_i}\left[\sum_{t=t_i}^{t_{i+1}-1}\left\langle q_t - u, \widehat{\ell}_t - b_t\right\rangle\right] \\
&= \underbrace{\mathbb{E}_{t_i}\left[\sum_{t=t_i}^{t_{i+1}-1}\left\langle q_t - v, \widehat{\ell}_t - b_t\right\rangle\right]}_{=:(I)} + \underbrace{\mathbb{E}_{t_i}\left[\sum_{t=t_i}^{t_{i+1}-1}\left\langle v - u, \widehat{\ell}_t - b_t\right\rangle\right]}_{=:(II)},
\end{aligned}
\tag{47}
$$

where term (I) is equivalent to

$$\mathbb{E}_{t_i}\left[\sum_{t=t_i}^{t_{i+1}-1}\langle q_t - v, g_t - b_t\rangle\right], \tag{48}$$

because $\left\langle q_t - v, g_t - \widehat{\ell}_t\right\rangle = 0$ according to Lemma D.4.1.

We then present the following lemma to provide a bound for $\text{ESTREG}_i(\pi^\star)$.

**Lemma C.7.5.** *For $g_t$ define in Eq. (45), $v$ defined in Eq. (46), and any non-decreasing learning rate such that $\gamma_1(s,a) \geq 256L^2|S|$ for all $(s,a)$, Algorithm 4 ensures that $\text{ESTREG}_i(\pi^\star)$ is bounded by*

$$
\begin{aligned}
&\mathcal{O}\left(\frac{L + L|S|C^P}{T} + \mathbb{E}_{t_i}\left[\sum_{s\neq s_L}\sum_{a\in A}\gamma_{t_{i+1}-1}(s,a)\log(\iota)\right]\right) \\
&+ \quad \mathcal{O}\left(\min\left\{\mathbb{E}_{t_i}\left[\sum_{t=t_i}^{t_{i+1}-1}\left\|\widehat{\ell}_t - b_t\right\|_{\nabla^{-2}\phi_t(q_t)}^2\right], \mathbb{E}_{t_i}\left[\sum_{t=t_i}^{t_{i+1}-1}\|g_t - b_t\|_{\nabla^{-2}\phi_t(q_t)}^2\right]\right\}\right),
\end{aligned}
\tag{49}
$$

*Proof.* We start from the decomposition in Eq. (47).

**Bounding (I).** By adding and subtracting $F_t(q_t) - G_t(\tilde{q}_t)$, we decompose (I) into a stability term and a penalty term:

$$\underbrace{\mathbb{E}_{t_i}\left[\sum_{t=t_i}^{t_{i+1}-1}\left(\left\langle q_t, \widehat{\ell}_t - b_t\right\rangle + F_t(q_t) - G_t(\tilde{q}_t)\right)\right]}_{\text{Stability term}} + \underbrace{\mathbb{E}_{t_i}\left[\sum_{t=t_i}^{t_{i+1}-1}\left(G_t(\tilde{q}_t) - F_t(q_t) - \left\langle v, \widehat{\ell}_t - b_t\right\rangle\right)\right]}_{\text{Penalty term}}.$$

As $\gamma_1(s,a) \geq 256L^2|S|$, Lemma C.7.6 ensures $\frac{1}{2}q_t(s,a) \leq \widetilde{q}_t(s,a) \leq 2q_t(s,a)$ for all $(s,a)$, which allows us to apply [Jin and Luo, 2020, Lemma 13] to bound the stability term by

$$\text{Stability term} = \mathcal{O}\left(\mathbb{E}_{t_i}\left[\sum_{t=t_i}^{t_{i+1}-1}\left\|\widehat{\ell}_t - b_t\right\|_{\nabla^{-2}\phi_t(q_t)}^2\right]\right). \tag{50}$$

The penalty term without expectation is bounded as:

$$\sum_{t=t_i}^{t_{i+1}-1} \left( G_t(\tilde{q}_t) - F_t(q_t) - \left\langle v, \widehat{\ell}_t - b_t \right\rangle \right)$$

$$= -F_{t_i}(q_{t_i}) + \sum_{t=t_i+1}^{t_{i+1}-1} (G_{t-1}(\tilde{q}_{t-1}) - F_t(q_t)) + G_{t_{i+1}-1}(\tilde{q}_{t_{i+1}-1}) - \left\langle v, \sum_{t=t_i}^{t_{i+1}-1} \left( \widehat{\ell}_t - b_t \right) \right\rangle$$

$$\leq -F_{t_i}(q_{t_i}) + \sum_{t=t_i+1}^{t_{i+1}-1} (G_{t-1}(q_t) - F_t(q_t)) + G_{t_{i+1}-1}(v) - \left\langle v, \sum_{t=t_i}^{t_{i+1}-1} \left( \widehat{\ell}_t - b_t \right) \right\rangle$$

$$= -\phi_{t_i}(q_{t_i}) + \sum_{t=t_i+1}^{t_{i+1}-1} (\phi_{t-1}(q_t) - \phi_t(q_t)) + \phi_{t_{i+1}-1}(v)$$

$$= -\phi_{t_i}(q_{t_i}) + \sum_{t=t_i+1}^{t_{i+1}-1} \sum_{s \neq s_L} \sum_{a \in A} (\gamma_{t-1}(s,a) - \gamma_t(s,a)) \log \left( \frac{1}{q_t(s,a)} \right) + \phi_{t_{i+1}-1}(v)$$

$$= -\phi_{t_i}(q_{t_i}) + \phi_{t_i}(v) + \sum_{t=t_i+1}^{t_{i+1}-1} \sum_{s \neq s_L} \sum_{a \in A} (\gamma_t(s,a) - \gamma_{t-1}(s,a)) \log \left( \frac{q_t(s,a)}{v(s,a)} \right)$$

$$= \sum_{s \neq s_L} \sum_{a \in A} \gamma_{t_i}(s,a) \log \left( \frac{q_{t_i}(s,a)}{v(s,a)} \right) + \sum_{t=t_i+1}^{t_{i+1}-1} \sum_{s \neq s_L} \sum_{a \in A} (\gamma_t(s,a) - \gamma_{t-1}(s,a)) \log \left( \frac{q_t(s,a)}{v(s,a)} \right)$$

$$\leq \sum_{s \neq s_L} \sum_{a \in A} \gamma_{t_i}(s,a) \log \left( \frac{\iota^2 q_{t_i}(s,a)}{q_{s,a}^{\max}(s,a)} \right) + \sum_{t=t_i+1}^{t_{i+1}-1} \sum_{s \neq s_L} \sum_{a \in A} (\gamma_t(s,a) - \gamma_{t-1}(s,a)) \log \left( \frac{\iota^2 q_t(s,a)}{q_{s,a}^{\max}(s,a)} \right)$$

$$\leq 2 \sum_{s \neq s_L} \sum_{a \in A} \gamma_{t_i}(s,a) \log (\iota) + 2 \sum_{t=t_i+1}^{t_{i+1}-1} \sum_{s \neq s_L} \sum_{a \in A} (\gamma_t(s,a) - \gamma_{t-1}(s,a)) \log (\iota)$$

$$= 2 \sum_{s \neq s_L} \sum_{a \in A} \gamma_{t_{i+1}-1}(s,a) \log (\iota), \tag{51}$$

where the second step uses the optimality of $\tilde{q}_{t-1}$ and $\tilde{q}_{t_{i+1}-1}$ so that $G_{t-1}(\tilde{q}_{t-1}) \leq G_{t-1}(q_t)$ and $G_{t_{i+1}-1}(\tilde{q}_{t_{i+1}-1}) \leq G_{t_{i+1}-1}(v)$; the third step follows the definitions in Eq. (43); the seventh step lower-bounds $v(s,a)$ by $\frac{1}{\iota^2} q_{s,a}^{\max}(s,a)$ for each $(s,a)$; the eighth step uses the definition of $q_{s,a}^{\max}$.

Now, by taking the equivalent perspective in Eq. (44) and Eq. (48) and repeating the exact same argument, the same bound holds with $\widehat{\ell}_t$ repalced by $g_t$. Thus, we have shown

$$\text{(I)} = \mathcal{O} \left( \mathbb{E}_{t_i} \left[ \sum_{s \neq s_L} \sum_{a \in A} \gamma_{t_{i+1}-1}(s,a) \log (\iota) \right] \right)$$

$$+ \mathcal{O} \left( \min \left\{ \mathbb{E}_{t_i} \left[ \sum_{t=t_i}^{t_{i+1}-1} \left\| \widehat{\ell}_t - b_t \right\|_{\nabla^{-2}\phi_t(q_t)}^2 \right], \mathbb{E}_{t_i} \left[ \sum_{t=t_i}^{t_{i+1}-1} \| g_t - b_t \|_{\nabla^{-2}\phi_t(q_t)}^2 \right] \right\} \right). \tag{52}$$

**Bounding (II).** By direct calculation, we have

$$\text{(II)} = \mathbb{E}_{t_i} \left[ \sum_{t=t_i}^{t_{i+1}-1} \left\langle v - u, \widehat{\ell}_t - b_t \right\rangle \right]$$

$$= \frac{1}{T^2} \mathbb{E}_{t_i} \left[ \sum_{t=t_i}^{t_{i+1}-1} \left\langle \frac{1}{|A||S|} \sum_{s,a} q_{s,a}^{\max} - u, \widehat{\ell}_t - b_t \right\rangle \right]$$

$$\leq \frac{1}{T^2} \mathbb{E}_{t_i} \left[ \left\| \frac{1}{|A||S|} \sum_{s,a} q_{s,a}^{\max} - u \right\|_1 \left\| \sum_{t=t_i}^{t_{i+1}-1} (\ell_t - b_t) \right\|_\infty \right],$$

where the first step uses the definition of $v$ in Eq. (46) and the third step applies the Hölder's inequality. Then, we further show:

$$
\begin{aligned}
\text{(II)} &\leq \frac{1}{T^2}\mathbb{E}_{t_i}\left[\left\|\frac{1}{|A||S|}\sum_{s,a}q^{\max}_{s,a} - u\right\|_1 \left\|\sum_{t=t_i}^{t_{i+1}-1}(\ell_t - b_t)\right\|_\infty\right] \\
&\leq \frac{1}{T^2}\mathbb{E}_{t_i}\left[\left(\frac{1}{|A||S|}\sum_{s,a}\|q^{\max}_{s,a}\|_1 + \|u\|_1\right)\left\|\sum_{t=t_i}^{t_{i+1}-1}(\ell_t - b_t)\right\|_\infty\right] \\
&\leq \frac{1}{T^2}\mathbb{E}_{t_i}\left[2L\left(\left\|\sum_{t=t_i}^{t_{i+1}-1}\ell_t\right\|_\infty + \left\|\sum_{t=t_i}^{t_{i+1}-1}b_t\right\|_\infty\right)\right] \\
&\leq \mathbb{E}_{t_i}\left[\frac{2L(t_{i+1} - t_i) + 4LC^{\mathsf{P}}|S|T}{T^2}\right] \\
&\leq \frac{2L + 4LC^{\mathsf{P}}|S|}{T},
\end{aligned}
\tag{53}
$$

where the second step repeatedly applies the triangle inequality, and those terms are bounded by $2L$ in the next step; the fourth step bounds $\ell_t(s,a) \leq 1$ for all $(s,a)$ and bounds $\sum_{t=t_i}^{t_{i+1}-1}b_t(s) \leq 2C^{\mathsf{P}}|S|T$ by using the fact $u_t(s) \geq \frac{1}{|S|T}$ for all $t, s$ (see Lemma D.2.8); the fifth step bounds $t_{i+1} - t_i$ by $T$.

Finally, we combine the bounds in Eq. (52) and Eq. (53) to complete the proof. □

**Lemma C.7.6** (Multiplicative Stability). *For $\widetilde{q}_t$ and $q_t$ defined in Eq. (44), if the learning rate fulfills $\gamma_t(s,a) \geq \gamma_1(s,a) = 256L^2|S|$ for all $t$ and $(s,a)$, then, $\frac{1}{2}q_t(s,a) \leq \widetilde{q}_t(s,a) \leq 2q_t(s,a)$ holds for all $(s,a)$.*

*Proof.* We first show that

$$
\begin{aligned}
\left\|\widehat{\ell}_t - b_t\right\|^2_{\nabla^{-2}\phi_t(q_t)} &= \sum_{s,a}\left(\frac{\mathbb{I}_t(s,a)\ell_t(s,a)}{u_t(s,a)} - b_t(s)\right)^2\frac{q_t(s,a)^2}{\gamma_t(s,a)} \\
&\leq \sum_{s,a}\left(\frac{\mathbb{I}_t(s,a)\ell_t(s,a)^2}{u_t(s,a)^2} + b_t(s)^2\right)\frac{q_t(s,a)^2}{\gamma_t(s,a)} \\
&\leq \sum_{s,a}\left(\frac{\mathbb{I}_t(s,a)}{u_t(s,a)^2} + \frac{16L^2}{u_t(s)^2}\right)\frac{q_t(s,a)^2}{\gamma_t(s,a)} \\
&\leq \sum_{s,a}\frac{\mathbb{I}_t(s,a)}{256L^2|S|} + \sum_{s,a}\frac{16L^2 \cdot q_t(s,a)^2}{256L^2|S| \cdot u_t(s)^2} \\
&\leq \frac{1}{256L|S|} + \sum_{s\neq s_L}\frac{1}{16|S|} \leq \frac{1}{8},
\end{aligned}
\tag{54}
$$

where the second step uses $(x - y)^2 \leq x^2 + y^2$ for any $x, y \geq 0$; the third step bounds $\ell_t(s,a)^2 \leq 1$ and $b_t(s)^2 \leq \frac{16L^2}{u_t(s)^2}$ according to the definition of Eq. (7); the fourth step holds as $\gamma_t(s,a) \geq \gamma_1(s,a) = 256L^2|S|$ and $q_t(s,a) \leq u_t(s,a)$ for all $(s,a)$; the fifth steps uses the fact that $\sum_a q_t(s,a)^2 \leq (\sum_a q_t(s,a))^2 \leq u_t(s)^2$.

Once Eq. (54) holds, one only needs to repeat the same argument of [Jin and Luo, 2020, Lemma 12] to obtain the claimed multiplicative stability. □

**Lemma C.7.7.** *Under the conditions of Lemma C.7.5, event $\mathcal{E}_{\mathrm{CON}}$, and the learning $\gamma_t(s,a)$ defined in Definition 5.2, the following holds for any deterministic policy $\pi^\star : S \to A$:*

$$
\mathbb{E}_{t_i}\left[\sum_{s\neq s_L}\sum_{a\in A}\gamma_{t_{i+1}-1}(s,a)\log(\iota) + \sum_{t=t_i}^{t_{i+1}-1}\|g_t - b_t\|^2_{\nabla^{-2}\phi_t(q_t)}\right]
$$

$$= \mathcal{O}\left(\mathbb{E}_{t_i}\left[L\sqrt{\log(\iota)}\sum_{s\neq s_L}\sum_{a\neq\pi^\star(s)}\sqrt{\mathbb{E}\left[\sum_{t=t_i}^{t_{i+1}-1}q^{P_t,\pi_t}(s,a)\right]}\right] + C^P\log(|S|T)\right)$$

$$+ \mathcal{O}\left(\mathbb{E}_{t_i}\left[\delta L^2|S|^2|A|\log(\iota)(t_{i+1}-t_i) + \sqrt{L|S|^3|A|^2\log(\iota)\sum_{t=t_i}^{t_{i+1}-1}C_t^P}\right]\right).$$

*Proof.* We start by bounding the second part for each $t$:

$$\|g_t - b_t\|_{\nabla^{-2}\phi_t(q_t)}^2 = \sum_{s,a}\frac{1}{\gamma_t(s,a)}q_t(s,a)^2(g_t(s,a)-b_t(s))^2$$

$$\leq 2\sum_{s,a}\frac{\nu_t(s,a)}{\gamma_t(s,a)} + 2\sum_{s,a}\frac{1}{\gamma_t(s,a)}q_t(s,a)^2 b_t(s)^2,$$

where the second step uses the definition of $\nu_t$ and $(x-y)^2 \leq 2(x^2+y^2)$ for any $x,y\in\mathbb{R}$.

We first focus on bounding $\sum_{t=t_i}^{t_{i+1}-1}\sum_{s,a}\frac{1}{\gamma_t(s,a)}q_t(s,a)^2 b_t(s)^2$:

$$\sum_{t=t_i}^{t_{i+1}-1}\sum_{s,a}\frac{1}{\gamma_t(s,a)}q_t(s,a)^2 b_t(s)^2$$

$$\leq \frac{1}{256L^2|S|}\sum_{t=t_i}^{t_{i+1}-1}\sum_{s,a}q_t(s,a)^2 b_t(s)^2$$

$$\leq \frac{4L}{256L^2|S|}\sum_{t=t_i}^{t_{i+1}-1}\sum_{s\neq s_L}q_t(s)b_t(s)$$

$$= \mathcal{O}\left(C^P\log(|S|T)\right), \tag{55}$$

where the first step uses the fact that $\gamma_t(s,a) \geq 256L^2|S|$, the second step bounds $b_t(s)^2 \leq 4Lb_t(s)$ and $\sum_a q_t(s,a)^2 \leq q_t(s)$, and the last step applies Lemma 3.1.

Then, we bound the remaining term:

$$\mathbb{E}_{t_i}\left[\sum_{s,a}\gamma_{t_{i+1}-1}(s,a)\log(\iota) + 2\sum_{t=t_i}^{t_{i+1}-1}\sum_{s,a}\frac{\nu_t(s,a)}{\gamma_t(s,a)}\right]. \tag{56}$$

Using Definition 5.2, we can bound Eq. (56) as:

$$\mathbb{E}_{t_i}\left[\sum_{s,a}\gamma_{t_{i+1}-1}(s,a)\log(\iota) + 2\sum_{t=t_i}^{t_{i+1}-1}\sum_{s,a}\frac{\nu_t(s,a)}{\gamma_t(s,a)}\right]$$

$$= \mathbb{E}_{t_i}\left[\sum_{s,a}\left(\gamma_{t_{i+1}-1}(s,a)\log(\iota) + \frac{4}{D}(\gamma_{t_{i+1}-1}(s,a)-\gamma_{t_i}(s,a))\right)\right]$$

$$\leq \mathbb{E}_{t_i}\left[\sum_{s,a}\left(\log(\iota)+\frac{4}{D}\right)\gamma_{t_{i+1}-1}(s,a)\right]$$

$$= \mathbb{E}_{t_i}\left[\sum_{s,a}5\log(\iota)\gamma_{t_{i+1}-1}(s,a)\right]$$

$$\leq \mathbb{E}_{t_i}\left[\sum_{s,a}\left(5\log(\iota)\sqrt{\frac{1}{\log(\iota)}\sum_{j=t_i}^{t_{i+1}-2}\nu_j(s,a)} + 256L^2|S|\right)\right]$$

$$\leq \mathcal{O}\left(\sqrt{\log(\iota)}\mathbb{E}_{t_i}\left[\sqrt{\sum_{s,a}\sum_{j=t_i}^{t_{i+1}-1}\nu_j(s,a)}\right] + L^2|S|^2|A|\right)$$

$$\leq \mathcal{O}\left(\mathbb{E}_{t_i}\left[\sum_{s\neq s_L}\sum_{a\neq\pi(s)}\sqrt{L^2\log(\iota)\sum_{t=t_i}^{t_{i+1}-1}q^{P_t,\pi_t}(s,a)}\right]\right)$$

$$+\mathcal{O}\left(\mathbb{E}_{t_i}\left[\delta L^2|S|^2|A|\log(\iota)(t_{i+1}-t_i)+\sqrt{L|S|^3|A|^2\log(\iota)\sum_{t=t_i}^{t_{i+1}-1}C_t^{\mathsf{P}}}\right]\right), \qquad (57)$$

where the first step applies the definition of $\gamma_t(s,a)$ which gives: $\frac{\nu_t(s,a)}{\gamma_t(s,a)}=\frac{2}{D}(\gamma_{t+1}(s,a)-\gamma_t(s,a))$; the second step simplifies the telescopic sum and uses $\gamma_{t_i}(s,a)\geq 0$; the third step uses $D=\frac{1}{\log(\iota)}$; the fourth step uses Proposition C.7.3; the sixth step applies Proposition C.7.4. $\qquad\square$

### C.7.3 Bounding EstReg

Using the equality $\textsc{EstReg}(\pi^\star)=\mathbb{E}\big[\sum_{i=1}^N\textsc{EstReg}_i(\pi^\star)\big]$ where $N$ is the number of epochs, we are now ready to complete the proof of Lemma C.3.4.

**Lemma C.7.8.** *Over the course of $T$ episodes, Algorithm 4 runs at most $\mathcal{O}(|S||A|\log(T))$ epochs.*

*Proof.* By definition, the algorithm resets each time a counter of the number of visits to a specific state-action pair doubles. Each state-action pair is visited at most once for each of the $T$ rounds, so it can trigger a new epoch at most $\log T$ times. Summing on all state-action pairs finishes the proof. $\quad\square$

*Proof of Lemma C.3.4.* Using Lemma C.7.5 and Lemma C.7.7, we have

$\textsc{EstReg}(\pi^\star)$

$$=\mathbb{E}\left[\sum_{i=1}^N\textsc{EstReg}_i(\pi^\star)\right]$$

$$=\mathcal{O}\left(\mathbb{E}\left[\sum_{i=1}^N\left(\frac{L+LC^{\mathsf{P}}|S|}{T}+\delta L^2|S|^2|A|\log(\iota)(t_{i+1}-t_i)+\sqrt{L|S|^3|A|^2\log(\iota)\sum_{t=t_i}^{t_{i+1}-1}C_t^{\mathsf{P}}}\right)\right]\right)$$

$$+\mathcal{O}\left(\mathbb{E}\left[\sum_{i=1}^N\left(C^{\mathsf{P}}\log(|S|T)+L\sqrt{\log(\iota)}\sum_{s\neq s_L}\sum_{a\neq\pi^\star(s)}\mathbb{E}_{t_i}\left[\sqrt{\sum_{t=t_i}^{t_{i+1}-1}q^{P_t,\pi_t}(s,a)}\right]\right)\right]\right)$$

$$=\mathcal{O}\left(\left(\frac{L+LC^{\mathsf{P}}|S|}{T}\right)|S||A|\log(T)+\delta L^2|S|^2|A|\log(\iota)T+|A||S|^2\log(\iota)\sqrt{L|A|C^{\mathsf{P}}}\right)$$

$$+\mathcal{O}\left(C^{\mathsf{P}}|S||A|\log(\iota)^2+\sqrt{L^2\log(\iota)}\sum_{s\neq s_L}\sum_{a\neq\pi^\star(s)}\mathbb{E}\left[\sum_{i=1}^N\sqrt{\sum_{t=t_i}^{t_{i+1}-1}q^{P_t,\pi_t}(s,a)}\right]\right)$$

$$=\mathcal{O}\left(\left(\frac{L+LC^{\mathsf{P}}|S|}{T}\right)|S||A|\log(T)+\delta L^2|S|^2|A|\log(\iota)T+|A||S|^2\log(\iota)\left(|A|+LC^{\mathsf{P}}\right)\right)$$

$$+\mathcal{O}\left(C^{\mathsf{P}}|S||A|\log(\iota)^2+\sqrt{L^2\log(\iota)}\sum_{s\neq s_L}\sum_{a\neq\pi^\star(s)}\mathbb{E}\left[\sum_{i=1}^N\sqrt{\sum_{t=t_i}^{t_{i+1}-1}q^{P_t,\pi_t}(s,a)}\right]\right)$$

$$=\mathcal{O}\left(\left(\frac{L|S||A|\log(T)}{T}\right)+\delta L^2|S|^2|A|\log(\iota)T+C^{\mathsf{P}}L|S|^2|A|\log(\iota)+|A|^2|S|^2\log(\iota)\right)$$

$$+\mathcal{O}\left(C^{\mathsf{P}}|S||A|\log(\iota)^2+\sqrt{L^2|S||A|\log^2(\iota)}\sum_{s\neq s_L}\sum_{a\neq\pi^\star(s)}\mathbb{E}\left[\sqrt{\sum_{t=1}^T q^{P_t,\pi_t}(s,a)}\right]\right),$$

where the first step follows from the definition of $\textsc{EstReg}(\pi^\star)$; the third step uses the fact that $N=\mathcal{O}(|S||A|\log(T))$ and the Cauchy-Schwarz inequality; the fourth step uses $\sqrt{xy}\leq x+y$ for any $x,y\geq 0$ with $x=LC^{\mathsf{P}}$ and $y=|A|$; the last step follows from Cauchy-Schwarz inequality again. $\qquad\square$

## C.8 Properties of Optimistic Transition

We summarize the properties guaranteed by the optimistic transition defined in Definition C.1.1.

**Lemma C.8.1.** *For any epoch $i$, any transition $P' \in \mathcal{P}_i$, any policy $\pi$, and any initial state $u \in S$, it holds that*

$$q^{\widetilde{P}_i,\pi}(s,a|u) \leq q^{P',\pi}(s,a|u), \quad \forall (s,a) \in S \times A.$$

*Proof.* We prove this result via a forward induction from layer $k(u)$ to layer $L-1$.

**Base Case:** for the initial state $u$, $q^{\widetilde{P}_i,\pi}(u,a|u) = q^{P',\pi}(u,a|u) = \pi(a|u)$ for any action $a \in A$. For the other state $s \in S_{k(u)}$, we have $q^{\widetilde{P}_i,\pi}(s,a|u) = q^{P',\pi}(s,a|u) = 0$.

**Induction step:** Suppose $q^{\widetilde{P}_i,\pi}(s,a|u) \leq q^{P',\pi}(s,a|u)$ holds for all the state-action pair $(s,a)$ with $k(s) < h$. Then, for any $(s,a) \in S_h \times A$, we have

$$q^{\widetilde{P}_i,\pi}(s,a|u) = \pi(a|s) \cdot \sum_{s' \in S_{h-1}} \sum_{a' \in A} q^{\widetilde{P}_i,\pi}(s',a'|u)\widetilde{P}_i(s|s',a')$$

$$\leq \pi(a|s) \cdot \sum_{s' \in S_{h-1}} \sum_{a' \in A} q^{P',\pi}(s',a'|u)P'(s|s',a')$$

$$= q^{P',\pi}(s,a|u),$$

where the second step follows from the induction hypothesis and the definition of optimistic transition in Definition C.1.1. $\square$

**Corollary C.8.2.** *Conditioning on the event $\mathcal{E}_{\mathrm{CON}}$, it holds for any epoch $i$ and any policy $\pi$ that*

$$q^{\widetilde{P}_i,\pi}(s,a) \leq q^{P,\pi}(s,a), \quad \forall (s,a) \in S \times A.$$

**Lemma C.8.3.** *(Optimism of Optimistic Transition) Suppose the high-probability event $\mathcal{E}_{\mathrm{CON}}$ holds. Then for any policy $\pi$, any $(s,a) \in S \times A$, and any valid loss function $\ell : S \times A \to \mathbb{R}_{\geq 0}$, it holds that*

$$Q^{\widetilde{P}_i,\pi}(s,a;\ell) \leq Q^{P,\pi}(s,a;\ell), \text{ and } V^{\widetilde{P}_i,\pi}(s;\ell) \leq V^{P,\pi}(s;\ell), \forall (s,a) \in S \times A.$$

*Proof.* According to Corollary C.8.2, we have for all epoch $i$ that

$$q^{\widetilde{P}_i,\pi}(s,a|u) \leq q^{P,\pi}(s,a|u), \quad \forall u \in S \quad \forall (s,a) \in S \times A. \tag{58}$$

Therefore, we have

$$V^{\widetilde{P}_i,\pi}(s;\ell) = \sum_{u \in S} \sum_{v \in A} q^{\widetilde{P}_i,\pi}(u,v|s)\ell(u,v)$$

$$\leq \sum_{u \in S} \sum_{v \in A} q^{P,\pi}(u,v|s)\ell(u,v)$$

$$= V^{P,\pi}(s;\ell),$$

where the second step follows from Eq. (58). The statement for the $Q$-function can be proven in the same way. $\square$

Next, we argue that our optimistic transition provides a tighter performance estimation compared to the approach of Jin et al. [2021]. Specifically, Jin et al. [2021] proposes to subtract the following exploration bonuses $\mathrm{BONUS}_i : S \times A \to \mathbb{R}$ from the loss functions

$$\mathrm{BONUS}_i(s,a) = L \cdot \min\left\{1, \sum_{s' \in S_{k(s)+1}} B_i(s,a,s')\right\},$$

where $B_i(s,a,s')$ is the confidence bound defined in Eq. (4). This makes sure $Q^{\bar{P}_i,\pi}(s,a;\ell - \mathrm{BONUS}_i)$ is no larger than the true $Q$-function $Q^{P,\pi}(s,a;\ell)$ as well, but is a looser lower bound as shown below.

**Lemma C.8.4.** *(Tighter Performance Estimation) For any policy $\pi$, any $(s, a) \in S \times A$, and any bounded loss function $\ell : S \times A \to [0, 1]$, it holds that*

$$Q^{\bar{P}_i, \pi}(s, a; \ell - \text{BONUS}_i) \leq Q^{\widetilde{P}_i, \pi}(s, a; \ell), \forall (s, a) \in S \times A.$$

*Proof.* We prove this result via a backward induction from layer $L$ to layer $0$.

**Base Case:** for the terminal state $s_L$, we have $Q^{\bar{P}_i, \pi}(s, a; \ell - \text{BONUS}_i) = Q^{\widetilde{P}_i, \pi}(s, a; \ell) = 0$.

**Induction step:** suppose the induction hypothesis holds for all the state-action pairs $(s, a) \in S \times A$ with $k(s) > h$. For any state-action pair $(s, a) \in S_h \times A$, we first have

$$Q^{\bar{P}_i, \pi}(s, a; \ell - \text{BONUS}_i) = \ell(s, a) - \text{BONUS}_i(s, a) + \sum_{u \in S_{h+1}} \bar{P}_i(u|s, a) V^{\bar{P}_i, \pi}(u; \ell - \text{BONUS}_i)$$

$$\leq \ell(s, a) - \text{BONUS}_i(s, a) + \sum_{u \in S_{h+1}} \bar{P}_i(u|s, a) V^{\widetilde{P}_i, \pi}(u; \ell).$$

Clearly, when $\sum_{u \in S_{k(s)+1}} B_i(s, a, u) \geq 1$, we have $Q^{\bar{P}_i, \pi}(s, a; \ell - \text{BONUS}_i) \leq 0$ by the definition of $\text{BONUS}_i$, which directly implies that $Q^{\bar{P}_i, \pi}(s, a; \ell - \text{BONUS}_i) \leq Q^{\widetilde{P}_i, \pi}(s, a; \ell)$. So we continue the bound under the condition $\sum_{u \in S_{k(s)+1}} B_i(s, a, u) < 1$:

$$Q^{\bar{P}_i, \pi}(s, a; \ell - \text{BONUS}_i) \leq \ell(s, a) - \text{BONUS}_i(s, a) + \sum_{u \in S_{h+1}} \bar{P}_i(u|s, a) V^{\widetilde{P}_i, \pi}(u; \ell)$$

$$\leq \ell(s, a) + \sum_{u \in S_{h+1}} \left( \bar{P}_i(u|s, a) - B_i(s, a, u) \right) V^{\widetilde{P}_i, \pi}(u; \ell)$$

$$\leq \ell(s, a) + \sum_{u \in S_{h+1}} \widetilde{P}_i(u|s, a) V^{\widetilde{P}_i, \pi}(u; \ell)$$

$$= Q^{\widetilde{P}_i, \pi}(s, a; \ell),$$

where the second step follows from the fact that $V^{\widetilde{P}_i, \pi}(u; \ell) \leq L$; the third step follows from the definition of optimistic transition $\widetilde{P}_i$.

Combining these two cases proves that $Q^{\bar{P}_i, \pi}(s, a; \ell - \text{BONUS}_i) \leq Q^{\widetilde{P}_i, \pi}(s, a; \ell)$ for any $(s, a) \in S_h \times A$, finishing the induction. $\qquad \square$

# D Supplementary Lemmas

## D.1 Expectation

**Lemma D.1.1.** *([Jin et al., 2021, Lemma D.3.6]) Suppose that a random variable $X$ satisfies the following conditions:*

- *$X < R$ where $R$ is a constant.*
- *$X < Y$ conditioning on event $A$, where $Y \geq 0$ is a random variable.*

*Then, it holds that $\mathbb{E}[X] \leq \mathbb{E}[Y] + \Pr[A^c] \cdot R$ where $A^c$ is the complementary event of $A$.*

## D.2 Confidence Bound with Known Corruption

In this subsection, we show that the empirical transition is centered around the transition $P$. Let $[T] := \{1, \cdots, T\}$. Recall that $E_i := \{t \in [T] : \text{ episode } t \text{ belongs to epoch } i\}$, $\iota = \frac{|S||A|T}{\delta}$, and we define the following quantities:

$$T_i(s,a) = \left\{ t \in \cup_{j=1}^{i-1} E_j : \exists k \text{ such that } (s_{t,k}, a_{t,k}) = (s,a) \right\} \quad \forall (s,a) \in S_k \times A, \forall i \geq 2, \forall k < L;$$

$$C_t^{\mathsf{P}}(s,a,s') = |P(s'|s,a) - P_t(s'|s,a)|, \qquad\qquad \forall (s,a,s') \in W_k, \forall i \in [T], \forall k < L;$$

$$C_i^{\mathsf{P}}(s,a,s') = \sum_{t \in T_i(s,a)} C_t^{\mathsf{P}}(s,a,s'), \qquad\qquad \forall (s,a,s') \in W_k, \forall i \in [T], \forall k < L.$$

Note that based on definition of $T_i(s,a)$, we have $m_i(s,a) = |T_i(s,a)|$. Then, we present the following lemma which shows the concentration bound between $P(s'|s,a)$ and $\bar{P}_i(s'|s,a)$.

**Lemma D.2.1.** *(Detailed restatement of Lemma 2.2) Event $\mathcal{E}_{\mathrm{CON}}$ occurs with probability at least $1 - \delta$ where,*

$$\mathcal{E}_{\mathrm{CON}} := \left\{ \forall (s,a,s') \in W_k, \forall i \in [T], \forall k < L : \left|P(s'|s,a) - \bar{P}_i(s'|s,a)\right| \leq B_i(s,a,s') \right\}, \quad (59)$$

*and $B_i(s,a,s')$ is defined in Eq. (4) as:*

$$B_i(s,a,s') = \min\left\{ 1, 16\sqrt{\frac{\bar{P}_i(s'|s,a)\log(\iota)}{m_i(s,a)}} + 64\frac{(C^{\mathsf{P}} + \log(\iota))}{m_i(s,a)} \right\}. \quad (60)$$

Proving this lemma requires several auxiliary results stated below. For any episode $t \in [T]$ and any layer $k < L$, we use $s_{t,k}^{\mathrm{IMG}}$ to denote an imaginary random state sampled from $P(\cdot|s_{t,k}, a_{t,k})$. For any $(s,a,s') \in W_k$, let

$$\bar{P}_i^{\mathrm{IMG}}(s'|s,a) = \frac{1}{m_i(s,a)} \sum_{t \in T_i(s,a)} \mathbb{I}\{s_{t,k(s)+1}^{\mathrm{IMG}} = s'\}.$$

We now proceed with a couple lemmas.

**Lemma D.2.2** (Lemma 2, [Jin et al., 2020]). *Event $\mathcal{E}^1$ occurs with probability at least $1 - 3\delta/4$ where*

$$\mathcal{E}^1 := \left\{ \forall (s,a,s') \in W_k, \forall i, k : \left|P(s'|s,a) - \bar{P}_i^{\mathrm{IMG}}(s'|s,a)\right| \leq \bar{\omega}_i(s,a,s') \right\},$$

*and $\bar{\omega}_i(s,a,s')$ for any $(s,a,s') \in W_k$ and $0 \leq k \leq L-1$ is defined as*

$$\bar{\omega}_i(s,a,s') = \min\left\{ 1, 2\sqrt{\frac{\bar{P}_i(s'|s,a)\log \iota}{m_i(s,a)}} + \frac{14\log \iota}{3m_i(s,a)} \right\}. \quad (61)$$

We note that as long as $|A|T \geq {}^{16}/_3$, event $\mathcal{E}^1$ can occur with probability at least $1 - 3\delta/4$ (unlike $1 - 4\delta$ in Lemma 2 of Jin et al. [2020] ).

**Lemma D.2.3.** *Event $\mathcal{E}^2$ occurs with probability at least $1 - \delta/4$ where*

$$\mathcal{E}^2 := \left\{ \forall (s, a, s') \in W_k, \forall i, k : \left| \bar{P}_i^{\mathsf{IMG}}(s'|s, a) - \bar{P}_i(s'|s, a) \right| \leq \omega_i(s, a, s') \right\},$$

*and $\omega_i(s, a, s')$ for any $(s, a, s') \in W_k$ and $0 \leq k \leq L - 1$ is defined as*

$$\omega_i(s, a, s') = \min \left\{ 1, \frac{4C_i^{\mathsf{P}}(s, a, s')}{m_i(s, a)} + \sqrt{\frac{24P(s'|s, a)\log \iota}{m_i(s, a)}} + \frac{6\log \iota}{m_i(s, a)} \right\}. \tag{62}$$

*Proof.* For any fixed $(s, a, s')$, if $m_i(s, a) = 0$, the claimed bound in $\mathcal{E}^2$ holds trivially, so we consider the case $m_i(s, a) \neq 0$ below. By definition we have:

$$\bar{P}_i^{\mathsf{IMG}}(s'|s, a) - \bar{P}_i(s'|s, a) = \frac{1}{m_i(s, a)} \sum_{t \in T_i(s, a)} \left( \mathbb{I}\{s_{t, k(s)+1}^{\mathsf{IMG}} = s'\} - \mathbb{I}\{s_{t, k(s)+1} = s'\} \right).$$

Then, we construct the martingale difference sequence $\{X_t(s, a, s')\}_{t=1}^{\infty}$ w.r.t. filtration $\{\mathcal{F}_{t, k(s)}\}_{t=1}^{\infty}$ (see [Lykouris et al., 2019, Definition 4.9] for the formal definition of these filtrations) where

$$X_t(s, a, s') = \mathbb{I}\{s_{t, k(s)+1}^{\mathsf{IMG}} = s'\} - \mathbb{I}\{s_{t, k(s)+1} = s'\} - \left( P(s'|s, a) - P_t(s'|s, a) \right).$$

With the definition of $X_t(s, a, s')$, one can show

$$\sum_{t \in T_i(s, a)} \mathbb{E}\left[ X_t(s, a, s')^2 | \mathcal{F}_{t, k(s)} \right]$$

$$\leq \sum_{t \in T_i(s, a)} \mathbb{E}\left[ \left( \mathbb{I}\{s_{t, k(s)+1}^{\mathsf{IMG}} = s'\} - \mathbb{I}\{s_{t, k(s)+1} = s'\} - \left( P(s'|s, a) - P_t(s'|s, a) \right) \right)^2 \bigg| \mathcal{F}_{t, k(s)} \right]$$

$$\leq \sum_{t \in T_i(s, a)} \mathbb{E}\left[ 2 \left( \mathbb{I}\{s_{t, k(s)+1}^{\mathsf{IMG}} = s'\} - \mathbb{I}\{s_{t, k(s)+1} = s'\} \right)^2 + 2 \left( P(s'|s, a) - P_t(s'|s, a) \right)^2 \bigg| \mathcal{F}_{t, k(s)} \right]$$

$$\leq \sum_{t \in T_i(s, a)} \mathbb{E}\left[ 2\mathbb{I}\{s_{t, k(s)+1}^{\mathsf{IMG}} = s'\} + 2\mathbb{I}\{s_{t, k(s)+1} = s'\} + 2C_t^{\mathsf{P}}(s, a, s') \bigg| \mathcal{F}_{t, k(s)} \right]$$

$$= 2 \sum_{t \in T_i(s, a)} \left( P(s'|s, a) + P_t(s'|s, a) + C_t^{\mathsf{P}}(s, a, s') \right), \tag{63}$$

where the second step uses $(x - y)^2 \leq 2(x^2 + y^2)$ for any $x, y \in \mathbb{R}$; the third step uses $(x - y)^2 \leq x^2 + y^2$ for $x, y \in \mathbb{R}_{\geq 0}$ and the fact that $C_t^{\mathsf{P}}(s, a, s') \in [0, 1]$, thereby $C_t^{\mathsf{P}}(s, a, s')^2 \leq C_t^{\mathsf{P}}(s, a, s')$; the last step holds based on the definitions of $T_i(s, a)$, $s_{t, k(s)+1}^{\mathsf{IMG}}$, and $s_{t, k(s)+1}$ as well as the fact that $C_t^{\mathsf{P}}(s, a, s')$ is $\mathcal{F}_{t, k(s)}$-measurable.

By using the result in Eq. (63), we bound the average second moment $\sigma^2$ as

$$\sigma^2 = \frac{\sum_{t \in T_i(s, a)} \mathbb{E}\left[ X_t^2 | \mathcal{F}_{t, k(s)} \right]}{m_i(s, a)} \leq \frac{2 \sum_{t \in T_i(s, a)} \left( P(s'|s, a) + P_t(s'|s, a) + C_t^{\mathsf{P}}(s, a, s') \right)}{m_i(s, a)}. \tag{64}$$

By applying Lemma D.2.4 with $b = 2$ and the upper bound of $\sigma^2$ shown in Eq. (64), as well as using the fact that $m_i(s, a) = |T_i(s, a)|$, for any $(s, a, s')$, we have the following with probability at least $1 - \delta/(4T|S|^2|A|)$,

$$\left| \bar{P}_i^{\mathsf{IMG}}(s'|s, a) - \bar{P}_i(s'|s, a) \right|$$

$$\leq \left| P(s'|s, a) - \frac{\sum_{t \in T_i(s, a)} P_t(s'|s, a)}{m_i(s, a)} \right| + \frac{4\log \left( \frac{8m_i(s, a)T|S|^2|A|}{\delta} \right)}{3m_i(s, a)}$$

$$+ \sqrt{\frac{4\log \left( \frac{16m_i(s, a)^2 T|S|^2|A|}{\delta} \right) \sum_{t \in T_i(s, a)} \left( P(s'|s, a) + P_t(s'|s, a) + C_t^{\mathsf{P}}(s, a, s') \right)}{m_i^2(s, a)}}$$

$$\leq \left| P(s'|s,a) - \frac{\sum_{t \in T_i(s,a)} P_t(s'|s,a)}{m_i(s,a)} \right| + \frac{8 \log \iota}{3 m_i(s,a)}$$

$$+ \sqrt{\frac{12 \log \iota \sum_{t \in T_i(s,a)} \left( P(s'|s,a) + P_t(s'|s,a) + C_t^{\mathsf{P}}(s,a,s') \right)}{m_i^2(s,a)}}$$

$$\leq \left| \frac{\sum_{t \in T_i(s,a)} P(s'|s,a)}{m_i(s,a)} - \frac{\sum_{t \in T_i(s,a)} P_t(s'|s,a)}{m_i(s,a)} \right| + \frac{8 \log \iota}{3 m_i(s,a)}$$

$$+ \sqrt{\frac{12 \log \iota \sum_{t \in T_i(s,a)} \left( P(s'|s,a) + P(s'|s,a) + C_t^{\mathsf{P}}(s,a,s') + C_t^{\mathsf{P}}(s,a,s') \right)}{m_i^2(s,a)}}$$

$$\leq \frac{C_i^{\mathsf{P}}(s,a,s')}{m_i(s,a)} + \sqrt{\frac{24 P(s'|s,a) \log \iota}{m_i(s,a)}} + \frac{8 \log \iota}{3 m_i(s,a)} + \frac{\sqrt{24 C_i^{\mathsf{P}}(s,a,s') \log \iota}}{m_i(s,a)}$$

$$\leq \frac{C_i^{\mathsf{P}}(s,a,s')}{m_i(s,a)} + \sqrt{\frac{24 P(s'|s,a) \log \iota}{m_i(s,a)}} + \frac{8 \log \iota}{3 m_i(s,a)} + \frac{\sqrt{6} \left( C_i^{\mathsf{P}}(s,a,s') + \log \iota \right)}{m_i(s,a)}$$

$$\leq \omega_i(s,a,s'),$$

where the first inequality applies Lemma D.2.4; the second inequality bounds all logarithmic terms by $\log \iota$ with an appropriate constant factor (using $m_i(s,a) \leq T$ and $|A| \geq 2$); the third step follows the fact that $P_t(s'|s,a) \leq P(s'|s,a) + C_t^{\mathsf{P}}(s,a,s')$; the fourth step uses the definition $C_i^{\mathsf{P}}(s,a,s') = \sum_{t \in T_i(s,a)} C_t^{\mathsf{P}}(s,a,s')$; the fifth step applies the inequality $\sqrt{4xy} \leq x + y, \forall x, y \geq 0$ for $x = C_i^{\mathsf{P}}(s,a,s')$ and $y = \log \iota$. Applying a union bound over all $(s,a,s')$ and epochs $i \leq T$, we complete the proof. $\qquad \square$

**Lemma D.2.4** (Anytime Version of Azuma-Bernstein). *Let $\{X_i\}_{i=1}^{\infty}$ be b-bounded martingale difference sequence with respect to $\mathcal{F}_i$. Let $\sigma^2 = \frac{1}{N} \sum_{i=1}^{N} \mathbb{E}[X_i^2 | F_{i-1}]$. Then, with probability at least $1 - \delta$, for any $N \in \mathbb{N}^+$, it holds that:*

$$\left| \frac{1}{N} \sum_{i=1}^{N} X_i \right| \leq \sqrt{\frac{2\sigma^2 \log(4N^2/\delta)}{N}} + \frac{2b \log(2N/\delta)}{3N}.$$

*Proof.* This follows the same argument as Lemma G.2 in [Lykouris et al., 2019]. $\qquad \square$

**Lemma D.2.5.** *Event $\mathcal{E}$ occurs with probability at least $1 - \delta$, where*

$$\mathcal{E} := \left\{ \forall (s,a,s') \in W_k, \forall i, k : \left| P(s'|s,a) - \bar{P}_i(s'|s,a) \right| \leq \omega_i(s,a,s') + \bar{\omega}_i(s,a,s') \right\}.$$

*Proof.* Conditioning on events $\mathcal{E}^1$ and $\mathcal{E}^2$, we have

$$\left| P(s'|s,a) - \bar{P}_i(s'|s,a) \right| \leq \left| P(s'|s,a) - \bar{P}_i^{\mathrm{IMG}}(s'|s,a) \right| + \left| \bar{P}_i^{\mathrm{IMG}}(s'|s,a) - \bar{P}_i(s'|s,a) \right|$$
$$\leq \bar{\omega}_i(s,a,s') + \omega_i(s,a,s'). \tag{65}$$

Using a union bound for $\mathcal{E}^1$ and $\mathcal{E}^2$, we complete the proof. $\qquad \square$

Armed with above results, we are now ready to prove Lemma D.2.1.

*Proof of Lemma D.2.1.* Conditioning on event $\mathcal{E}$, for any fixed $(s,a,s')$ with $m_i(s,a) \neq 0$ (otherwise the desired bound holds trivially), we have

$$\omega_i(s,a,s')$$

$$\leq \sqrt{\frac{24 P(s'|s,a) \log \iota}{m_i(s,a)}} + \frac{6 \log \iota}{m_i(s,a)} + \frac{4 C_i^{\mathsf{P}}(s,a,s')}{m_i(s,a)}$$

$$\leq \sqrt{\frac{24 \left( \bar{P}_i(s'|s,a) + \bar{\omega}_i(s,a,s') + \frac{4 C_i^{\mathsf{P}}(s,a,s')}{m_i(s,a)} + \omega_i(s,a,s') \right) \log \iota}{m_i(s,a)}} + \frac{6 \log \iota}{m_i(s,a)} + \frac{4 C_i^{\mathsf{P}}(s,a,s')}{m_i(s,a)}$$

$$\leq \sqrt{\frac{24\bar{P}_i(s'|s,a)\log\iota}{m_i(s,a)}} + \sqrt{\frac{24\bar{\omega}_i(s,a,s')\log\iota}{m_i(s,a)}}$$

$$+ \sqrt{\frac{96\frac{C_i^{\mathsf{P}}(s,a,s')}{m_i(s,a)}\log\iota}{m_i(s,a)}} + \sqrt{\frac{24\omega_i(s,a,s')\log\iota}{m_i(s,a)}} + \frac{6\log\iota}{m_i(s,a)} + \frac{4C_i^{\mathsf{P}}(s,a,s')}{m_i(s,a)}$$

$$\leq \sqrt{\frac{24\bar{P}_i(s'|s,a)\log\iota}{m_i(s,a)}} + \bar{\omega}_i(s,a,s') + \frac{\sqrt{96C_i^{\mathsf{P}}(s,a,s')\log\iota}}{m_i(s,a)} + \frac{\omega_i(s,a,s')}{2} + \frac{24\log\iota + 4C_i^{\mathsf{P}}(s,a,s')}{m_i(s,a)}$$

$$\leq \sqrt{\frac{24\bar{P}_i(s'|s,a)\log\iota}{m_i(s,a)}} + \bar{\omega}_i(s,a,s') + \frac{28C_i^{\mathsf{P}}(s,a,s')}{m_i(s,a)} + \frac{\omega_i(s,a,s')}{2} + \frac{25\log\iota}{m_i(s,a)},$$

where the second step holds under $\mathcal{E}$; the third step uses $\sqrt{\sum_{i=1}^n x_i} \leq \sum_{i=1}^n \sqrt{x_i}$ for all $x_i \in \mathbb{R}_{\geq 0}$; the fourth step and the fifth step use $2\sqrt{xy} \leq x + y$ for $x, y \geq 0$. Rearranging the above, we obtain

$$\omega_i(s,a,s') \leq \sqrt{\frac{96\bar{P}_i(s'|s,a)\log\iota}{m_i(s,a)}} + 2\bar{\omega}_i(s,a,s') + \frac{56C_i^{\mathsf{P}}(s,a,s')}{m_i(s,a)} + \frac{50\log\iota}{m_i(s,a)} \qquad (66)$$

Thus, conditioning on event $\mathcal{E}$, one can show for all $(s,a,s') \in W_k$ and $k < L-1$

$$\left|P(s'|s,a) - \bar{P}_i(s'|s,a)\right| \leq \omega_i(s,a,s') + \bar{\omega}_i(s,a,s')$$

$$\leq \sqrt{\frac{96\bar{P}_i(s'|s,a)\log\iota}{m_i(s,a)}} + 3\bar{\omega}_i(s,a,s') + \frac{56C_i^{\mathsf{P}}(s,a,s')}{m_i(s,a)} + \frac{50\log\iota}{m_i(s,a)}$$

$$\leq 16\sqrt{\frac{\bar{P}_i(s'|s,a)\log\iota}{m_i(s,a)}} + \frac{56C_i^{\mathsf{P}}(s,a,s')}{m_i(s,a)} + \frac{64\log\iota}{m_i(s,a)}$$

$$\leq 16\sqrt{\frac{\bar{P}_i(s'|s,a)\log\iota}{m_i(s,a)}} + 64\frac{C_i^{\mathsf{P}}(s,a,s') + \log\iota}{m_i(s,a)}, \qquad (67)$$

where the second step uses Eq. (66), the third step applies the definition of $\bar{\omega}_i(s,a,s')$. Finally, using the fact $C_i^{\mathsf{P}}(s,a,s') \leq C^{\mathsf{P}}$, we complete the proof. $\qquad \square$

Then, we present an immediate corollary of Lemma D.2.1.

**Corollary D.2.6.** *Consider any epoch $i$ and any transition $P' \in \mathcal{P}_i$. The following holds (recall $\widehat{m}_i$ defined at the beginning of the appendix),*

$$\|P'(\cdot|s,a) - \bar{P}_i(\cdot|s,a)\|_1 \leq 2 \cdot \min\left\{1, \frac{32C^{\mathsf{P}}}{\widehat{m}_i(s,a)} + 8\sqrt{\frac{|S_{k(s)+1}|\log\iota}{\widehat{m}_i(s,a)}} + \frac{32|S_{k(s)+1}|\log\iota}{\widehat{m}_i(s,a)}\right\}.$$

*Proof.* As $P' \in \mathcal{P}_i$, we start from Eq. (67):

$$\|P'(\cdot|s,a) - \bar{P}_i(\cdot|s,a)\|_1 \leq \sum_{s'\in k(s)+1}\left(\frac{64C_i^{\mathsf{P}}(s,a,s')}{m_i(s,a)} + 16\sqrt{\frac{\bar{P}_i(s'|s,a)\log\iota}{m_i(s,a)}} + \frac{64\log\iota}{m_i(s,a)}\right)$$

$$\leq \frac{64C^{\mathsf{P}}}{m_i(s,a)} + 16\sqrt{\frac{|S_{k(s)+1}|\log\iota}{m_i(s,a)}} + \frac{64|S_{k(s)+1}|\log\iota}{m_i(s,a)}, \qquad (68)$$

where the last step uses the Cauchy-Schwarz inequality and the fact that $\sum_{s'\in k(s)+1} C_i^{\mathsf{P}}(s,a,s') \leq C^{\mathsf{P}}$. Since $\|P(\cdot|s,a) - \bar{P}_i(\cdot|s,a)\|_1 \leq 2$, we combine this trivial bound and the bound of $\|P(\cdot|s,a) - \bar{P}_i(\cdot|s,a)\|_1$ in Eq. (68) to arrive at

$$\|P'(\cdot|s,a) - \bar{P}_i(\cdot|s,a)\|_1 \leq 2 \cdot \min\left\{1, \frac{32C^{\mathsf{P}}}{m_i(s,a)} + 8\sqrt{\frac{|S_{k(s)+1}|\log\iota}{m_i(s,a)}} + \frac{32|S_{k(s)+1}|\log\iota}{m_i(s,a)}\right\}$$

$$= 2 \cdot \min\left\{1, \frac{32C^{\mathsf{P}}}{\widehat{m}_i(s,a)} + 8\sqrt{\frac{|S_{k(s)+1}|\log \iota}{\widehat{m}_i(s,a)}} + \frac{32|S_{k(s)+1}|\log \iota}{\widehat{m}_i(s,a)}\right\},$$

finishing the proof. $\qquad\square$

We conclude this subsection with two other useful lemmas.

**Lemma D.2.7.** *Conditioning on event* $\mathcal{E}_{\mathrm{CON}}$, *it holds for all tuple* $(s,a,s')$ *and epoch* $i$ *that*

$$\left|P(s'|s,a) - \bar{P}_i(s'|s,a)\right| \leq \mathcal{O}\left(\min\left\{1, \sqrt{\frac{P(s'|s,a)\log(\iota)}{\widehat{m}_i(s,a)}} + \frac{C^{\mathsf{P}} + \log(\iota)}{\widehat{m}_i(s,a)}\right\}\right).$$

*Proof.* Fix the epoch $i$ and tuple $(s,a,s')$. According to the definitions of $\mathcal{E}_{\mathrm{CON}}$ in Eq. (59) and $\widehat{m}$, we have

$$\left|P(s'|s,a) - \bar{P}_i(s'|s,a)\right| \leq \min\left\{1, 16\sqrt{\frac{\bar{P}_i(s'|s,a)\log(\iota)}{\widehat{m}_i(s,a)}} + 64 \cdot \frac{C^{\mathsf{P}} + \log(\iota)}{\widehat{m}_i(s,a)}\right\}.$$

Therefore, by direct calculation, we have

$$\left|P(s'|s,a) - \bar{P}_i(s'|s,a)\right|$$

$$\leq 16\sqrt{\frac{\left(\left|P(s'|s,a) - \bar{P}_i(s'|s,a)\right| + P(s'|s,a)\right)\log(\iota)}{\widehat{m}_i(s,a)}} + 64 \cdot \frac{C^{\mathsf{P}} + \log(\iota)}{\widehat{m}_i(s,a)}$$

$$\leq 8\left(\sqrt{\frac{P(s'|s,a)\log(\iota)}{\widehat{m}_i(s,a)}} + \sqrt{\frac{\left|P(s'|s,a) - \bar{P}_i(s'|s,a)\right|\log(\iota)}{\widehat{m}_i(s,a)}}\right) + 64 \cdot \frac{C^{\mathsf{P}} + \log(\iota)}{\widehat{m}_i(s,a)}$$

$$= 8\sqrt{\frac{P(s'|s,a)\log(\iota)}{\widehat{m}_i(s,a)}} + 64 \cdot \frac{C^{\mathsf{P}} + \log(\iota)}{\widehat{m}_i(s,a)} + \sqrt{\left|P(s'|s,a) - \bar{P}_i(s'|s,a)\right| \cdot \frac{64\log(\iota)}{\widehat{m}_i(s,a)}}$$

$$\leq 8\sqrt{\frac{P(s'|s,a)\log(\iota)}{\widehat{m}_i(s,a)}} + 96 \cdot \frac{C^{\mathsf{P}} + \log(\iota)}{\widehat{m}_i(s,a)} + \frac{1}{2}\left|P(s'|s,a) - \bar{P}_i(s'|s,a)\right|,$$

where the second step and last step follow from the fact that $\sqrt{xy} \leq \frac{1}{2}(x+y)$ for any $x, y \geq 0$. Finally, rearranging the above inequality finishes the proof. $\qquad\square$

**Lemma D.2.8.** *(Lower Bound of Upper Occupancy Measure) For any episode $t$ and state $s \neq s_L$, it always holds that* $u_t(s) \geq 1/|S|T$.

*Proof.* Fix the episode $t$ and state $s$. We prove the lemma by constructing a specific transition $\widehat{P} \in \mathcal{P}_{i(t)}$, such that $q^{\widehat{P},\pi}(s) \geq 1/|S|T$ for any policy $\pi$, which suffices due to the definition of $u_t(s)$.

Specifically, $\widehat{P}$ is defined as, for any tuple $(s,a,s') \in W_k$ and $k = 0, \ldots, L-1$:

$$\widehat{P}(s'|s,a) = \bar{P}_{i(t)}(s'|s,a) \cdot \left(1 - \frac{1}{T}\right) + \frac{1}{|S_{k+1}|T}.$$

By direct calculation, one can verify that $\widehat{P}$ is a valid transition function. Then, we show that $\widehat{P} \in \mathcal{P}_{i(t)}$ by verifying the condition for any transition tuple $(s,a,s') \in W_k$ and $k = 0, \ldots, L-1$:

$$\left|\widehat{P}(s'|s,a) - \bar{P}_{i(t)}(s'|s,a)\right| = \frac{1}{T} \cdot \left|\bar{P}_{i(t)}(s'|s,a) - \frac{1}{|S_{k+1}|}\right| \leq \frac{1}{T} \leq B_{i(t)}(s,a,s'),$$

where the last step follows from the definition of confidence intervals in Eq. (4).

Finally, we show that $q^{\widehat{P},\pi}(s) \geq 1/|S|T$ as:

$$q^{\widehat{P},\pi}(s) = \sum_{u \in S_{k(s)-1}} \sum_{v \in A} q^{\widehat{P},\pi}(u,v)\widehat{P}(s|u,v) \geq \sum_{u \in S_{k(s)-1}} \sum_{v \in A} q^{\widehat{P},\pi}(u,v) \cdot \frac{1}{T|S|} = \frac{1}{T|S|},$$

which concludes the proof. $\qquad\square$

## D.3 Difference Lemma

**Lemma D.3.1** (Theorem 5.2.1 of [Kakade, 2003]). *(Performance Difference Lemma) For any policies $\pi_1, \pi_2$ and any loss function $\ell : S \times A \to \mathbb{R}$,*

$$
\begin{aligned}
&V^{P,\pi_1}(s_0; \ell) - V^{P,\pi_2}(s_0; \ell) \\
&= \sum_{s \neq s_L} \sum_{a \in A} q^{P,\pi_2}(s,a) \left( V^{P,\pi_1}(s; \ell) - Q^{P,\pi_1}(s,a; \ell) \right) \\
&= \sum_{s \neq s_L} \sum_{a \in A} q^{P,\pi_2}(s) \left( \pi_1(a|s) - \pi_2(a|s) \right) Q^{P,\pi_1}(s,a; \ell).
\end{aligned}
$$

In fact, the same also holds for our optimistic transition where the layer structure is violated. For completeness, we include a proof below.

**Lemma D.3.2.** *For any policies $\pi_1, \pi_2$, any loss function $\ell : S \times A \to \mathbb{R}$, and any transition $\widetilde{P}$ where $\sum_{s' \in S_{k(s)+1}} \widetilde{P}(s'|s,a) \leq 1$ for all state-action pairs $(s,a)$ (the remaining probability is assigned to $s_L$), we have*

$$
V^{\widetilde{P},\pi_1}(s_0; \ell) - V^{\widetilde{P},\pi_2}(s_0; \ell) = \sum_{s \neq s_L} \sum_{a \in A} q^{\widetilde{P},\pi_2}(s,a) \left( V^{\widetilde{P},\pi_1}(s; \ell) - Q^{\widetilde{P},\pi_1}(s,a; \ell) \right).
$$

*Proof.* By direct calculation, we have for any state $s \neq s_L$:

$$
\begin{aligned}
V^{\widetilde{P},\pi_1}(s_0; \ell) - V^{\widetilde{P},\pi_2}(s_0; \ell) &= \left( \sum_{a \in A} \pi_2(a|s_0) \right) V^{\widetilde{P},\pi_1}(s_0; \ell) - \sum_{a \in A} \pi_2(a|s_0) Q^{\widetilde{P},\pi_1}(s_0,a; \ell) \\
&\quad + \sum_{a \in A} \pi_2(a|s_0) \left( Q^{\widetilde{P},\pi_1}(s_0,a; \ell) - Q^{\widetilde{P},\pi_2}(s_0,a; \ell) \right) \\
&= \sum_{a \in A} q^{\widetilde{P},\pi_2}(s_0,a) \left( V^{\widetilde{P},\pi_1}(s_0; \ell) - Q^{\widetilde{P},\pi_1}(s_0,a; \ell) \right) \\
&\quad + \sum_{a \in A} \sum_{s' \in S_1} \pi_2(a|s_0) \widetilde{P}(s'|s_0,a) \left( V^{\widetilde{P},\pi_1}(s'; \ell) - V^{\widetilde{P},\pi_2}(s'; \ell) \right) \\
&= \sum_{a \in A} q^{\widetilde{P},\pi_2}(s_0,a) \left( V^{\widetilde{P},\pi_1}(s_0; \ell) - Q^{\widetilde{P},\pi_1}(s_0,a; \ell) \right) \\
&\quad + \sum_{a \in A} \sum_{s' \in S_1} q^{\widetilde{P},\pi_2}(s') \left( V^{\widetilde{P},\pi_1}(s'; \ell) - V^{\widetilde{P},\pi_2}(s'; \ell) \right) \\
&= \sum_{s \neq s_L} \sum_{a \in A} q^{\widetilde{P},\pi_2}(s,a) \left( V^{\widetilde{P},\pi_1}(s; \ell) - Q^{\widetilde{P},\pi_1}(s,a; \ell) \right),
\end{aligned}
$$

where the last step follows from recursively repeating the first three steps. $\qquad\square$

**Lemma D.3.3.** *(Occupancy Measure Difference, [Jin et al., 2021, Lemma D.3.1]) For any transition functions $P_1, P_2$ and any policy $\pi$,*

$$
\begin{aligned}
q^{P_1,\pi}(s) - q^{P_2,\pi}(s) &= \sum_{k=0}^{k(s)-1} \sum_{(u,v,w) \in W_k} q^{P_1,\pi}(u,v) \left( P_1(w|u,v) - P_2(w|u,v) \right) q^{P_2,\pi}(s|w) \\
&= \sum_{k=0}^{k(s)-1} \sum_{(u,v,w) \in W_k} q^{P_2,\pi}(u,v) \left( P_1(w|u,v) - P_2(w|u,v) \right) q^{P_1,\pi}(s|w),
\end{aligned}
$$

*where $q^{P',\pi}(s|w)$ is the probability of visiting $s$ starting $w$ under policy $\pi$ and transition $P'$.*

**Lemma D.3.4.** *([Dann et al., 2023, Lemma 17]) For any policies $\pi_1, \pi_2$ and transition function $P$,*

$$
\sum_{s \neq s_L} \sum_{a \in A} \left| q^{P,\pi_1}(s,a) - q^{P,\pi_2}(s,a) \right| \leq L \sum_{s \neq s_L} \sum_{a \in A} q^{P,\pi_1}(s) \left| \pi_1(a|s) - \pi_2(a|s) \right|.
$$

**Corollary D.3.5.** *For any policies $\pi_1$, mapping $\pi_2 : S \to A$ (that is, a deterministic policy), and transition function $P$, we have*

$$\sum_{s \neq s_L} \sum_{a \in A} \left| q^{P,\pi_1}(s,a) - q^{P,\pi_2}(s,a) \right| \leq 2L \sum_{s \neq s_L} \sum_{a \neq \pi_2(s)} q^{P,\pi_1}(s,a).$$

*Proof.* According to Lemma D.3.4, we have

$$\sum_{s \neq s_L} \sum_{a \in A} \left| q^{P,\pi_1}(s,a) - q^{P,\pi_2}(s,a) \right| \leq L \sum_{s \neq s_L} \sum_{a \in A} q^{P,\pi_1}(s) \left| \pi_1(a|s) - \pi_2(a|s) \right|.$$

Note that, for every state $s$, it holds that

$$\sum_{a \in A} \left| \pi_1(a|s) - \pi_2(a|s) \right| = \sum_{a \neq \pi_2(s)} \pi_1(a|s) + \left| \pi_1(\pi_2(s)|s) - 1 \right| = 2 \sum_{a \neq \pi_2(s)} \pi_1(a|s),$$

where the first step follows from the fact that $\pi_2(a|s) = 1$ when $a = \pi_2(s)$, and $\pi_2(b|s) = 0$ for any other action $b \neq \pi_2(s)$.

Therefore, we have

$$L \sum_{s \neq s_L} \sum_{a \in A} q^{P,\pi_1}(s) \left| \pi_1(a|s) - \pi_2(a|s) \right| \leq 2L \sum_{s \neq s_L} \sum_{a \neq \pi_2(s)} q^{P,\pi_1}(s,a),$$

which concludes the proof. $\qquad\square$

According to Lemma D.3.3, we can estimate the occupancy measure difference caused by the corrupted transition function $P_t$ at episode $t$.

**Corollary D.3.6.** *For any episode $t$ and any policy $\pi$, we have*

$$\left| q^{P,\pi}(s) - q^{P_t,\pi}(s) \right| \leq C_t^{\mathsf{P}}, \quad \forall s \neq s_L, \quad \text{and} \quad \sum_{s \neq s_L} \left| q^{P,\pi}(s) - q^{P_t,\pi}(s) \right| \leq L C_t^{\mathsf{P}}.$$

*Proof.* By direct calculation, we have for any episode $t$ and any $s \neq s_L$

$$\left| q^{P,\pi}(s) - q^{P_t,\pi}(s) \right| \leq \sum_{k=0}^{k(s)-1} \sum_{(u,v,w) \in W_k} q^{P,\pi}(u,v) \left| P(w|u,v) - P_t(w|u,v) \right| q^{P_t,\pi}(s|w)$$

$$\leq \sum_{k=0}^{k(s)-1} \sum_{u \in S_k} \sum_{v \in A} q^{P,\pi}(u,v) \left\| P(\cdot|u,v) - P_t(\cdot|u,v) \right\|_1$$

$$\leq C_t^{\mathsf{P}},$$

where the first step follows from Lemma D.3.3, the second step bounds $q^{P_t,\pi}(s|w) \leq 1$, and the last two steps follows from the definition of $C_t^{\mathsf{P}}$.

Moreover, taking the summation over all states $s \neq s_L$, we have

$$\sum_{s \neq s_L} \left| q^{P,\pi}(s) - q^{P_t,\pi}(s) \right|$$

$$\leq \sum_{s \neq s_L} \sum_{k=0}^{k(s)-1} \sum_{(u,v,w) \in W_k} q^{P,\pi}(u,v) \left| P(w|u,v) - P_t(w|u,v) \right| q^{P_t,\pi}(s|w)$$

$$= \sum_{k=0}^{L-1} \sum_{(u,v,w) \in W_k} q^{P,\pi}(u,v) \left| P(w|u,v) - P_t(w|u,v) \right| \sum_{h=k+1}^{L-1} \sum_{s \in S_h} q^{P_t,\pi}(s|w)$$

$$\leq L \sum_{k=0}^{k(s)-1} \sum_{u \in S_k} \sum_{v \in A} q^{P,\pi}(u,v) \left\| P(\cdot|u,v) - P_t(\cdot|u,v) \right\|_1$$

$$\leq L C_t^{\mathsf{P}},$$

where the third step follows from the fact that $\sum_{s \in S_h} q^{P_t,\pi}(s|w) = 1$ for any $h \geq k(w)$. $\qquad\square$

**Corollary D.3.7.** *For any policy sequence $\{\pi_t\}_{t=1}^T$ and loss functions $\{\ell_t\}_{t=1}^T$ such that $\ell_t \in S \times A \to [0,1]$ for any $t \in \{1, \cdots, T\}$, it holds that*

$$\sum_{t=1}^T \left|\left\langle q^{P,\pi_t} - q^{P_t,\pi_t}, \ell_t \right\rangle\right| \leq LC^{\mathsf{P}}.$$

*Proof.* By direct calculation, we have

$$
\begin{aligned}
\sum_{t=1}^T \left|\left\langle q^{P,\pi_t} - q^{P_t,\pi_t}, \ell_t \right\rangle\right| &\leq \sum_{t=1}^T \left\| q^{P,\pi_t} - q^{P_t,\pi_t} \right\|_1 \\
&= \sum_{t=1}^T \sum_{s \neq s_L} \sum_{a \in A} \left| q^{P,\pi_t}(s,a) - q^{P_t,\pi_t}(s,a) \right| \\
&= \sum_{t=1}^T \sum_{s \neq s_L} \sum_{a \in A} \left| q^{P,\pi_t}(s) - q^{P_t,\pi_t}(s) \right| \pi_t(a|s) \\
&= \sum_{t=1}^T \sum_{s \neq s_L} \left| q^{P,\pi_t}(s) - q^{P_t,\pi_t}(s) \right| \\
&\leq \sum_{t=1}^T LC_t^{\mathsf{P}} = LC^{\mathsf{P}},
\end{aligned}
$$

where the first step applies the Hölder's inequality; the third step follows form the fact that $q^{P',\pi_t}(s,a) = q^{P',\pi_t}(s)\pi_t(a|s)$ for any state-action pair $(s,a)$ and any transition function $P'$; the fifth step applies [Corollary D.3.6]; the last step follows from the definition of $C^{\mathsf{P}}$. $\qquad\square$

Following the same idea in the proof of [Dann et al., 2023, Lemma 16], we also consider a tighter bound of the difference between occupancy measures in the following lemma.

**Lemma D.3.8.** *Suppose the event $\mathcal{E}_{\mathrm{CON}}$ holds. For any state $s \neq s_L$, episode $t$ and transition function $P_t' \in \mathcal{P}_{i(t)}$, we have*

$$
\begin{aligned}
\left| q^{P_t',\pi_t}(s) - q^{P,\pi_t}(s) \right| \leq &\mathcal{O}\left( \sum_{k=0}^{k(s)-1} \sum_{(u,v,w) \in W_k} q^{P,\pi_t}(u,v) \sqrt{\frac{P(w|u,v)\log(\iota)}{\widehat{m}_{i(t)}(u,v)}} q^{P,\pi_t}(s|w) \right) \\
&+ \mathcal{O}\left( |S|^2 \sum_{s \neq s_L} \sum_{a \in A} q^{P,\pi_t}(s,a) \left( \frac{C^{\mathsf{P}} + \log(\iota)}{\widehat{m}_{i(t)}(s,a)} \right) \right).
\end{aligned}
\tag{69}
$$

*Proof.* According to [Lemma D.3.3], we have

$$\left| q^{P_t',\pi_t}(s) - q^{P,\pi_t}(s) \right|$$

$$\leq \sum_{k=0}^{k(s)-1} \sum_{(u,v,w) \in W_k} q^{P,\pi_t}(u,v) \left| P(w|u,v) - P_t'(w|u,v) \right| q^{P_t',\pi_t}(s|w)$$

$$\leq \sum_{k=0}^{k(s)-1} \sum_{(u,v,w) \in W_k} q^{P,\pi_t}(u,v) \left| P(w|u,v) - P_t'(w|u,v) \right| q^{P,\pi_t}(s|w)$$

$$+ \sum_{k=0}^{k(s)-1} \sum_{(u,v,w) \in W_k} q^{P,\pi_t}(u,v) \left| P(w|u,v) - P_t'(w|u,v) \right| \sum_{h=k+1}^{k(s)-1} \sum_{(x,y,z) \in W_k} q^{P,\pi_t}(x,y|w) \left| P(z|x,y) - P_t'(z|x,y) \right|$$

$$\leq \mathcal{O}\left( \sum_{k=0}^{k(s)-1} \sum_{(u,v,w) \in W_k} q^{P,\pi_t}(u,v) \sqrt{\frac{P(w|u,v)\log(\iota)}{\widehat{m}_{i(t)}(u,v)}} q^{P,\pi_t}(s|w) \right)$$

$$+ \mathcal{O}\left(\underbrace{\sum_{k=0}^{k(s)-1} \sum_{(u,v,w) \in W_k} q^{P,\pi_t}(u,v) \left(\frac{C^{\mathsf{P}} + \log(\iota)}{\widehat{m}_{i(t)}(u,v)}\right) q^{P,\pi_t}(s|w)}_{\text{TERM (A)}}\right)$$

$$+ \underbrace{\sum_{k=0}^{k(s)-1} \sum_{(u,v,w) \in W_k} q^{P,\pi_t}(u,v) \left|P(w|u,v) - P'_t(w|u,v)\right| \sum_{h=k+1}^{k(s)-1} \sum_{(x,y,z) \in W_h} q^{P,\pi_t}(x,y|w) \left|P(z|x,y) - P'_t(z|x,y)\right|}_{\text{TERM (B)}},$$

where the second step first subtracts and adds $q^{P'_t,\pi_t}(s|w)$ and then applies Lemma D.3.3 again for $|q^{P'_t,\pi_t}(s|w) - q^{P,\pi_t}(s|w)|$, and the third step follows from Lemma D.2.7 .

Clearly, we can bound term (a) as:

$$\sum_{k=0}^{k(s)-1} \sum_{(u,v,w) \in W_k} q^{P,\pi_t}(u,v) \left(\frac{C^{\mathsf{P}} + \log(\iota)}{\widehat{m}_{i(t)}(u,v)}\right) q^{P,\pi_t}(s|w) \leq |S| \sum_{s \neq s_L} \sum_{a \in A} q^{P,\pi_t}(s,a) \left(\frac{C^{\mathsf{P}} + \log(\iota)}{\widehat{m}_{i(t)}(s,a)}\right).$$

On the other hand, for term (b), we decompose it into three terms in Eq. (70), Eq. (71), and Eq. (72):

$$\sum_{k=0}^{k(s)-1} \sum_{(u,v,w) \in W_k} q^{P,\pi_t}(u,v) \left|P(w|u,v) - P'_t(w|u,v)\right| \sum_{h=k+1}^{k(s)-1} \sum_{(x,y,z) \in W_k} q^{P,\pi_t}(x,y|w) \left|P(z|x,y) - P'_t(z|x,y)\right|$$

$$\leq \mathcal{O}\left(\sum_{k=0}^{k(s)-1} \sum_{(u,v,w) \in W_k} q^{P,\pi_t}(u,v) \sqrt{\frac{P(w|u,v)\log(\iota)}{\widehat{m}_{i(t)}(u,v)}} \sum_{h=k+1}^{k(s)-1} \sum_{(x,y,z) \in W_h} q^{P,\pi_t}(x,y|w) \sqrt{\frac{P(z|x,y)\log(\iota)}{\widehat{m}_{i(t)}(x,y)}}\right) \tag{70}$$

$$+ \mathcal{O}\left(\sum_{k=0}^{k(s)-1} \sum_{(u,v,w) \in W_k} q^{P,\pi_t}(u,v) \sqrt{\frac{P(w|u,v)\log(\iota)}{\widehat{m}_{i(t)}(u,v)}} \sum_{h=k+1}^{k(s)-1} \sum_{(x,y,z) \in W_h} q^{P,\pi_t}(x,y|w) \left(\frac{C^{\mathsf{P}} + \log(\iota)}{\widehat{m}_{i(t)}(x,y)}\right)\right) \tag{71}$$

$$+ \mathcal{O}\left(\sum_{k=0}^{k(s)-1} \sum_{(u,v,w) \in W_k} q^{P,\pi_t}(u,v) \left(\frac{C^{\mathsf{P}} + \log(\iota)}{\widehat{m}_{i(t)}(u,v)}\right) \sum_{h=k+1}^{k(s)-1} \sum_{(x,y,z) \in W_k} q^{P,\pi_t}(x,y|w)\right). \tag{72}$$

According to the AM-GM inequality, we have the term in Eq. (70) bounded as

$$\sum_{k=0}^{k(s)-1} \sum_{(u,v,w) \in W_k} q^{P,\pi_t}(u,v) \sqrt{\frac{P(w|u,v)\log(\iota)}{\widehat{m}_{i(t)}(u,v)}} \sum_{h=k+1}^{k(s)-1} \sum_{(x,y,z) \in W_h} q^{P,\pi_t}(x,y|w) \sqrt{\frac{P(z|x,y)\log(\iota)}{\widehat{m}_{i(t)}(x,y)}}$$

$$= \sum_{k=0}^{k(s)-1} \sum_{(u,v,w) \in W_k} \sum_{h=k+1}^{k(s)-1} \sum_{(x,y,z) \in W_h} q^{P,\pi_t}(u,v) \sqrt{\frac{P(w|u,v)\log(\iota)}{\widehat{m}_{i(t)}(u,v)}} q^{P,\pi_t}(x,y|w) \sqrt{\frac{P(z|x,y)\log(\iota)}{\widehat{m}_{i(t)}(x,y)}}$$

$$\leq \sum_{k=0}^{k(s)-1} \sum_{(u,v,w) \in W_k} \sum_{h=k+1}^{k(s)-1} \sum_{(x,y,z) \in W_h} \frac{q^{P,\pi_t}(u,v) q^{P,\pi_t}(x,y|w) P(z|x,y)\log(\iota)}{\widehat{m}_{i(t)}(u,v)}$$

$$+ \sum_{k=0}^{k(s)-1} \sum_{(u,v,w) \in W_k} \sum_{h=k+1}^{k(s)-1} \sum_{(x,y,z) \in W_h} \frac{q^{P,\pi_t}(u,v) P(w|u,v) q^{P,\pi_t}(x,y|w)\log(\iota)}{\widehat{m}_{i(t)}(x,y)}$$

$$= \sum_{k=0}^{k(s)-1} \sum_{(u,v,w) \in W_k} \frac{q^{P,\pi_t}(u,v)\log(\iota)}{\widehat{m}_{i(t)}(u,v)} \left(\sum_{h=k+1}^{k(s)-1} \sum_{(x,y,z) \in W_h} q^{P,\pi_t}(x,y|w) P(z|x,y)\right)$$

$$+ \sum_{h=0}^{k(s)-1} \sum_{(x,y,z) \in W_h} \frac{\log(\iota)}{\widehat{m}_{i(t)}(x,y)} \left(\sum_{k=0}^{h-1} \sum_{(u,v,w) \in W_k} q^{P,\pi_t}(u,v) P(w|u,v) q^{P,\pi_t}(x,y|w)\right)$$

$$\leq L \sum_{k=0}^{k(s)-1} \sum_{(u,v,w)\in W_k} \frac{q^{P,\pi_t}(u,v)\log(\iota)}{\widehat{m}_{i(t)}(u,v)} + L \sum_{h=0}^{k(s)-1} \sum_{(x,y,z)\in W_h} \frac{q^{P,\pi_t}(x,y)\log(\iota)}{\widehat{m}_{i(t)}(x,y)}$$

$$\leq \mathcal{O}\left( L|S| \sum_{k=0}^{L-1} \sum_{s\in S_k} \sum_{a\in A} \frac{q^{P,\pi_t}(s,a)\log(\iota)}{\widehat{m}_{i(t)}(s,a)} \right),$$

where the first step follows from re-arranging the summation; the second step applies the fact that $\sqrt{xy} \leq x+y$ for any $x,y \geq 0$; the third step rearranges the summation order, and the fourth step follows form the facts that $\sum_{x\in S_h}\sum_{y\in A} q^{P,\pi_t}(x,y|w)P(z|x,y) = q^{P,\pi_t}(z|w)$ and $\sum_{(u,v,w)\in W_k} q^{P,\pi_t}(u,v)P(w|u,v)q^{P,\pi_t}(x,y|w) = q^{P,\pi_t}(x,y)$ for any $k$ and $(x,y,z)$.

Similarly, we have the term in Eq. (71) bounded as

$$\sum_{k=0}^{k(s)-1} \sum_{(u,v,w)\in W_k} q^{P,\pi_t}(u,v)\sqrt{\frac{P(w|u,v)\log(\iota)}{\widehat{m}_{i(t)}(u,v)}} \sum_{h=k+1}^{k(s)-1} \sum_{(x,y,z)\in W_h} q^{P,\pi_t}(x,y|w)\left(\frac{C^{\mathsf{P}}+\log(\iota)}{\widehat{m}_{i(t)}(x,y)}\right)$$

$$\leq \sum_{k=0}^{k(s)-1} \sum_{(u,v,w)\in W_k} \sum_{h=k+1}^{k(s)-1} \sum_{(x,y,z)\in W_h} \frac{q^{P,\pi_t}(u,v)q^{P,\pi_t}(x,y|w)\log(\iota)}{\widehat{m}_{i(t)}(u,v)}$$

$$+ \sum_{k=0}^{k(s)-1} \sum_{(u,v,w)\in W_k} \sum_{h=k+1}^{k(s)-1} \sum_{(x,y,z)\in W_h} q^{P,\pi_t}(u,v)P(w|u,v)q^{P,\pi_t}(x,y|w)\left(\frac{C^{\mathsf{P}}+\log(\iota)}{\widehat{m}_{i(t)}(x,y)}\right)$$

$$\leq \sum_{k=0}^{k(s)-1} \sum_{(u,v,w)\in W_k} \frac{q^{P,\pi_t}(u,v)\log(\iota)}{\widehat{m}_{i(t)}(u,v)} \left(\sum_{h=k+1}^{k(s)-1} \sum_{(x,y,z)\in W_h} q^{P,\pi_t}(x,y|w)\right)$$

$$+ \sum_{h=0}^{L-1} \sum_{(x,y,z)\in W_h} \sum_{k=0}^{k(s)-1} \left(\sum_{(u,v,w)\in W_k} q^{P,\pi_t}(u,v)P(w|u,v)q^{P,\pi_t}(x,y|w)\right) \frac{C^{\mathsf{P}}+\log(\iota)}{\widehat{m}_{i(t)}(x,y)}$$

$$\leq |S| \sum_{k=0}^{k(s)-1} \sum_{(u,v,w)\in W_k} \frac{q^{P,\pi_t}(u,v)\log(\iota)}{\widehat{m}_{i(t)}(u,v)} + \sum_{h=0}^{L-1} \sum_{(x,y,z)\in W_h} \sum_{k=0}^{k(s)-1} q^{P,\pi_t}(x,y)\left(\frac{C^{\mathsf{P}}+\log(\iota)}{\widehat{m}_{i(t)}(x,y)}\right)$$

$$\leq |S|^2 \sum_{s\neq s_L} \sum_{a\in A} \frac{q^{P,\pi_t}(s,a)\log(\iota)}{\widehat{m}_{i(t)}(s,a)} + L|S| \sum_{s\neq s_L} \sum_{a\in A} q^{P,\pi_t}(s,a)\left(\frac{C^{\mathsf{P}}+\log(\iota)}{\widehat{m}_{i(t)}(s,a)}\right),$$

where the first step follows from the fact that $\widehat{m}_{i(t)}(s,a) \geq C + \log(\iota)$ according to its definition; the third step follows from the facts that $\sum_{x\in S_h}\sum_{y\in A} q^{P,\pi_t}(x,y|w) \leq 1$ and $\sum_{(u,v,w)\in W_k} q^{P,\pi_t}(u,v)P(w|u,v)q^{P,\pi_t}(x,y|w) = q^{P,\pi_t}(x,y)$.

For the term in Eq. (72), we have

$$\sum_{k=0}^{k(s)-1} \sum_{(u,v,w)\in W_k} q^{P,\pi_t}(u,v)\frac{C^{\mathsf{P}}+\log(\iota)}{\widehat{m}_{i(t)}(u,v)} \sum_{h=k+1}^{k(s)-1} \sum_{(x,y,z)\in W_k} q^{P,\pi_t}(x,y|w)$$

$$= \sum_{k=0}^{k(s)-1} \sum_{u\in S_k} \sum_{v\in A} \sum_{w\in S_{k+1}} q^{P,\pi_t}(u,v)\frac{C^{\mathsf{P}}+\log(\iota)}{\widehat{m}_{i(t)}(u,v)} \left(\sum_{h=k+1}^{k(s)-1} \sum_{z\in S_{k+1}} 1\right)$$

$$\leq |S|^2 \sum_{u\neq s_L} \sum_{v\in A} q^{P,\pi_t}(s,a)\left(\frac{C^{\mathsf{P}}+\log(\iota)}{\widehat{m}_{i(t)}(u,v)}\right),$$

according to the fact that $\sum_{x\in S_h}\sum_{y\in A} q^{P,\pi_t}(x,y|w) \leq 1$.

Putting all the bounds for the terms in Eq. (70), Eq. (71), and Eq. (72) together yields the bound of TERM (B) that

$$\sum_{k=0}^{k(s)-1} \sum_{(u,v,w)\in W_k} q^{P,\pi_t}(u,v)\left|P(w|u,v)-P'_t(w|u,v)\right| \sum_{h=k+1}^{k(s)-1} \sum_{(x,y,z)\in W_h} q^{P,\pi_t}(x,y|w)\left|P(z|x,y)-P'_t(z|x,y)\right|$$

$$= \mathcal{O}\left( |S|^2 \sum_{u \neq s_L} \sum_{v \in A} q^{P,\pi_t}(u,v) \left( \frac{C^{\mathsf{P}} + \log(\iota)}{\widehat{m}_{i(t)}(u,v)} \right) \right).$$

Combining this bound with that of TERM (A) finishes the proof. $\qquad \square$

## D.4 Loss Shifting Technique with Optimistic Transition

**Lemma D.4.1.** *Fix an optimistic transition function $\widetilde{P}$ (defined in Section 5). For any policy $\pi$ and any loss function $\ell : S \times A \to \mathbb{R}$, we define function $g : S \times A \to \mathbb{R}$ similar to that of Jin et al. [2021] as:*

$$g^{\widetilde{P},\pi}(s,a;\ell) \triangleq \left( Q^{\widetilde{P},\pi}(s,a;\ell) - V^{\widetilde{P},\pi}(s;\ell) - \ell(s,a) \right),$$

*where the state-action and state value function $Q^{\widetilde{P},\pi}$ and $V^{\widetilde{P},\pi}$ are defined with respect to the optimistic transition $\widetilde{P}$ and $\pi$ as following:*

$$Q^{\widetilde{P},\pi}(s,a;\ell) = \ell(s,a) + \sum_{s' \in S_{k(s)+1}} \widetilde{P}(s'|s,a) V^{\widetilde{P},\pi}(s';\ell),$$

$$V^{\widetilde{P},\pi}(s;\ell) = \begin{cases} 0, & s = s_L, \\ \sum_{a \in A} \pi(a|s) Q^{\widetilde{P},\pi}(s,a;\ell), & s \neq s_L. \end{cases}$$

*Then, it holds for any policy $\pi'$ that,*

$$\left\langle q^{\widetilde{P},\pi'}, g^{\widetilde{P},\pi} \right\rangle = \sum_{s \neq s_L} \sum_{a \in A} q^{\widetilde{P},\pi'}(s,a) \cdot g^{\widetilde{P},\pi}(s,a;\ell) = -V^{\widetilde{P},\pi}(s_0;\ell),$$

*where $V^{\widetilde{P},\pi}(s_0;\ell)$ is only related to $\widetilde{P}$, $\pi$ and $\ell$, and is independent with $\pi'$.*

*Proof.* By the extended performance difference lemma of the optimistic transition in Lemma D.3.1, we have the following equality holds for any policy $\pi'$:

$$V^{\widetilde{P},\pi'}(s_0;\ell) - V^{\widetilde{P},\pi}(s_0;\ell) = \sum_{s \neq s_L} \sum_{a \in A} q^{\widetilde{P},\pi'}(s,a) \left( Q^{\widetilde{P},\pi}(s,a;\ell) - V^{\widetilde{P},\pi}(s;\ell) \right).$$

On the other hand, we also have

$$V^{\widetilde{P},\pi'}(s_0;\ell) = \sum_{s \neq s_L} \sum_{a \in A} q^{\widetilde{P},\pi'}(s,a) \ell(s,a).$$

Therefore, subtracting $V^{\widetilde{P},\pi'}(s_0;\ell)$ yields that

$$-V^{\widetilde{P},\pi}(s_0;\ell) = \sum_{s \neq s_L} \sum_{a \in A} q^{\widetilde{P},\pi'}(s,a) \left( Q^{\widetilde{P},\pi}(s,a;\ell) - V^{\widetilde{P},\pi}(s;\ell) - \ell(s,a) \right).$$

The proof is finished after using the definition of $g^{\widetilde{P},\pi}$. $\qquad \square$

Therefore, we have the following result for the FTRL framework which is similar to [Jin et al., 2021, Corollary A.1.2.].

**Corollary D.4.2.** *The FTRL update in Algorithm 4 can be equivalently written as:*

$$\widehat{q}_t = \operatorname*{argmin}_{q \in \Omega(\widetilde{P}_i)} \left\langle q, \sum_{\tau=t_i}^{t-1} (\widehat{\ell}_\tau - b_\tau) \right\rangle + \phi_t(q) = \operatorname*{argmin}_{x \in \Omega(\widetilde{P}_i)} \left\langle q, \sum_{\tau=t_i}^{t-1} (g_\tau - b_\tau) \right\rangle + \phi_t(q),$$

*where $g_\tau(s,a) = Q^{\widetilde{P}_i,\pi_\tau}(s,a;\widehat{\ell}_\tau) - V^{\widetilde{P}_i,\pi_\tau}(s;\widehat{\ell}_\tau)$ for any state-action pair $(s,a)$.*

## D.5 Estimation Error

**Lemma D.5.1.** *([Jin et al., 2020, Lemma 10]) With probability at least $1 - \delta$, we have for all $k = 0, \ldots, L - 1$,*

$$\sum_{t=1}^{T} \sum_{(s,a) \in S_k \times A} \frac{q^{P_t, \pi_t}(s, a)}{\max\left\{m_{i(t)}(s, a), 1\right\}} = \mathcal{O}\left(|S_k||A|\log(T) + \log\left(\frac{L}{\delta}\right)\right),$$

*and*

$$\sum_{t=1}^{T} \sum_{(s,a) \in S_k \times A} \frac{q^{P_t, \pi_t}(s, a)}{\sqrt{\max\left\{m_{i(t)}(s, a), 1\right\}}} = \mathcal{O}\left(\sqrt{|S_k||A|T} + |S_k||A|\log(T) + \log\left(\frac{L}{\delta}\right)\right).$$

*Proof.* Simply replacing the stationary transition $P$ with the sequence of transitions $\{P_t\}_{t=1}^{T}$ in the proof of [Jin et al., 2020, Lemma 10] suffices. □

**Proposition D.5.2.** *Let $\mathcal{E}_{\text{EST}}$ be the event such that we have for all $k = 0, \ldots, L - 1$ simultaneously*

$$\sum_{t=1}^{T} \sum_{(s,a) \in S_k \times A} \frac{q^{P_t, \pi_t}(s, a)}{\max\left\{m_{i(t)}(s, a), 1\right\}} = \mathcal{O}\left(|S_k||A|\log(T) + \log(\iota)\right),$$

*and*

$$\sum_{t=1}^{T} \sum_{(s,a) \in S_k \times A} \frac{q^{P_t, \pi_t}(s, a)}{\sqrt{\max\left\{m_{i(t)}(s, a), 1\right\}}} = \mathcal{O}\left(\sqrt{|S_k||A|T} + |S_k||A|\log(T) + \log(\iota)\right).$$

*We have $\Pr[\mathcal{E}_{\text{EST}}] \geq 1 - \delta$.*

*Proof.* The proof directly follows from the definition of $\iota$, which ensures that $\iota \geq L/\delta$. □

Based on the event $\mathcal{E}_{\text{EST}}$, we introduce the following lemma which is critical in analyzing the estimation error.

**Lemma D.5.3.** *Suppose the event $\mathcal{E}_{\text{EST}}$ defined in Proposition D.5.2 holds. Then, we have for all $k = 0, \ldots, L - 1$,*

$$\sum_{t=1}^{T} \sum_{(s,a) \in S_k \times A} \frac{q^{P, \pi_t}(s, a)}{\widehat{m}_{i(t)}(s, a)} = \mathcal{O}\left(|S_k||A|\log(T) + \log(\iota)\right),$$

*and,*

$$\sum_{t=1}^{T} \sum_{(s,a) \in S_k \times A} q^{P, \pi_t}(s, a)\left(\frac{C^P + \log(\iota)}{\widehat{m}_{i(t)}(s, a)}\right) = \mathcal{O}\left(\left(C^P + \log(\iota)\right)|S_k||A|\log(\iota)\right),$$

*Proof.* According to Corollary D.3.6, we have

$$\sum_{t=1}^{T} \sum_{(s,a) \in S_k \times A} \frac{q^{P, \pi_t}(s, a)}{\widehat{m}_{i(t)}(s, a)}$$

$$\leq \sum_{t=1}^{T} \sum_{(s,a) \in S_k \times A} \frac{q^{P_t, \pi_t}(s, a)}{\widehat{m}_{i(t)}(s, a)} + \sum_{t=1}^{T} \sum_{(s,a) \in S_k \times A} \frac{\left|q^{P, \pi_t}(s, a) - q^{P_t, \pi_t}(s, a)\right|}{\widehat{m}_{i(t)}(s, a)}$$

$$\leq \sum_{t=1}^{T} \sum_{(s,a) \in S_k \times A} \frac{q^{P_t, \pi_t}(s, a)}{\max\left\{1, m_{i(t)}(s, a)\right\}} + \sum_{t=1}^{T} \sum_{(s,a) \in S_k \times A} \frac{C_t^P}{C^P + \log(\iota)}$$

$$\leq \mathcal{O}\left(|S_k||A|\log(T) + \log(\iota)\right),$$

where the second step follows from the definitions of $C^{\mathsf{P}}$ and $\widehat{m}_{i(t)}(s,a)$, and the last step applies Lemma D.5.1.

Similarly, we have

$$
\sum_{t=1}^{T} \sum_{(s,a) \in S_k \times A} q^{P,\pi_t}(s,a) \left( \frac{C^{\mathsf{P}} + \log(\iota)}{\widehat{m}_{i(t)}(s,a)} \right)
$$

$$
\leq \sum_{t=1}^{T} \sum_{(s,a) \in S_k \times A} \left( \left| q^{P_t,\pi_t}(s,a) - q^{P,\pi_t}(s,a) \right| + q^{P_t,\pi_t}(s,a) \right) \left( \frac{C^{\mathsf{P}} + \log(\iota)}{\widehat{m}_{i(t)}(s,a)} \right)
$$

$$
\leq \sum_{t=1}^{T} \sum_{(s,a) \in S_k \times A} \left( C_t^{\mathsf{P}} + q^{P_t,\pi_t}(s,a) \left( \frac{C^{\mathsf{P}} + \log(\iota)}{\widehat{m}_{i(t)}(s,a)} \right) \right)
$$

$$
\leq |S_k||A| \sum_{t=1}^{T} C_t^{\mathsf{P}} + \left( C^{\mathsf{P}} + \log(\iota) \right) \sum_{t=1}^{T} \sum_{(s,a) \in S_k \times A} \frac{q^{P_t,\pi_t}(s,a)}{\widehat{m}_{i(t)}(s,a)}
$$

$$
= \mathcal{O} \left( |S_k||A| \, C^{\mathsf{P}} + \left( C^{\mathsf{P}} + \log(\iota) \right) \left( |S_k||A| \log(T) + \log(\iota) \right) \right)
$$

$$
= \mathcal{O} \left( \left( C^{\mathsf{P}} + \log(\iota) \right) |S_k||A| \log(\iota) \right),
$$

where the first step adds and subtracts $q^{P_t,\pi_t}$; the second step follows from Corollary D.3.6; the fourth step uses Proposition D.5.2 due to the fact that $\widehat{m}_i(s,a) \geq \max\{m_i(s,a), 1\}$. $\qquad \square$

**Lemma D.5.4.** *(Extension of [Dann et al., 2023, Lemma 16] for adversarial transition) Suppose the high-probability events $\mathcal{E}_{\mathrm{EST}}$ (defined in Proposition D.5.2) and $\mathcal{E}_{\mathrm{CON}}$ (defined in Lemma D.2.1) hold together. Let $P_t^s$ be a transition function in $\mathcal{P}_{i(t)}$ which depends on $s$, and let $g_t(s) \in [0, G]$ for some $G > 0$. Then,*

$$
\sum_{t=1}^{T} \sum_{s \neq s_L} \left| q^{P_t^s,\pi_t}(s) - q^{P,\pi_t}(s) \right| g_t(s) = \mathcal{O} \left( \sqrt{L|S|^2|A| \log^2(\iota) \sum_{t=1}^{T} \sum_{s \neq s_L} q^{P,\pi_t}(s) g_t(s)^2} \right)
$$

$$
+ \mathcal{O} \left( G \left( C^{\mathsf{P}} + \log(\iota) \right) L^2 |S|^4 |A| \log(\iota) \right).
$$

*Proof.* By Lemma D.3.8, under the event $\mathcal{E}_{\mathrm{CON}}$, we have

$$
\sum_{t=1}^{T} \sum_{s \neq s_L} \left| q^{P_t^s,\pi_t}(s) - q^{P,\pi_t}(s) \right| g_t(s)
$$

$$
\leq \mathcal{O} \left( \sum_{t=1}^{T} \sum_{s \neq s_L} g_t(s) \sum_{k=0}^{k(s)-1} \sum_{(u,v,w) \in W_k} q^{P,\pi_t}(u,v) \sqrt{\frac{P(w|u,v) \log(\iota)}{\widehat{m}_{i(t)}(u,v)}} q^{P,\pi_t}(s|w) \right) \quad (73)
$$

$$
+ \mathcal{O} \left( G|S|^3 \sum_{t=1}^{T} \sum_{s \neq s_L} \sum_{a \in A} q^{P,\pi_t}(s,a) \left( \frac{C^{\mathsf{P}} + \log(\iota)}{\widehat{m}_{i(t)}(s,a)} \right) \right).
$$

By Lemma D.5.3, the last two terms can be bounded under the event $\mathcal{E}_{\mathrm{EST}}$ as

$$
\mathcal{O} \left( G|S|^3 \sum_{t=1}^{T} \sum_{s \neq s_L} \sum_{a \in A} q^{P,\pi_t}(s,a) \left( \frac{C^{\mathsf{P}} + \log(\iota)}{\widehat{m}_{i(t)}(s,a)} \right) \right)
$$

$$
\leq \mathcal{O} \left( G|S|^3 \sum_{k=0}^{L-1} \left( C^{\mathsf{P}} + \log(\iota) \right) |S_k||A| \log(\iota) \right)
$$

$$
= \mathcal{O} \left( G \left( C^{\mathsf{P}} + \log(\iota) \right) L|S|^4 |A| \log(\iota) \right).
$$

For the first term in Eq. (73), we first rewrite it as

$$\sum_{t=1}^{T}\sum_{s\neq s_L}\sum_{h=0}^{k(s)-1}\sum_{(u,v,w)\in W_h} q^{P,\pi_t}(u,v)\sqrt{\frac{P(w|u,v)\log(\iota)}{\widehat{m}_{i(t)}(u,v)}}q^{P,\pi_t}(s|w)g_t(s)$$

$$\leq \sum_{h=0}^{L-1}\sum_{t=1}^{T}\sum_{s\neq s_L}\sum_{(u,v,w)\in W_h} q^{P,\pi_t}(u,v)\sqrt{\frac{P(w|u,v)\log(\iota)}{\widehat{m}_{i(t)}(u,v)}}q^{P,\pi_t}(s|w)g_t(s).$$

For any $\theta > 0$ and layer $h$, conditioning on the event $\mathcal{E}_{\mathrm{EST}}$, we have

$$\sum_{t=1}^{T}\sum_{s\neq s_L}\sum_{(u,v,w)\in W_h} q^{P,\pi_t}(u,v)\sqrt{\frac{P(w|u,v)\log(\iota)}{\widehat{m}_{i(t)}(u,v)}}q^{P,\pi_t}(s|w)g_t(s)$$

$$=\sum_{t=1}^{T}\sum_{s\neq s_L}\sum_{(u,v,w)\in W_h}\sqrt{\frac{q^{P,\pi_t}(u,v)q^{P,\pi_t}(s|w)\log(\iota)}{\widehat{m}_{i(t)}(u,v)}\cdot q^{P,\pi_t}(u,v)P(w|u,v)q^{P,\pi_t}(s|w)g_t(s)^2}$$

$$\leq\sum_{t=1}^{T}\sum_{s\neq s_L}\sum_{(u,v,w)\in W_h}\left(\theta\cdot\frac{q^{P,\pi_t}(u,v)q^{P,\pi_t}(s|w)\log(\iota)}{\widehat{m}_{i(t)}(u,v)}+\frac{1}{\theta}\cdot q^{P,\pi_t}(u,v)P(w|u,v)q^{P,\pi_t}(s|w)g_t(s)^2\right)$$

$$=\theta\cdot\sum_{t=1}^{T}\sum_{u\in S_h}\sum_{v\in A}\frac{q^{P,\pi_t}(u,v)\log(\iota)}{\widehat{m}_{i(t)}(u,v)}\cdot\left(\sum_{w\in S_{h+1}}\sum_{s\neq s_L}q^{P,\pi_t}(s|w)\right)$$

$$+\frac{1}{\theta}\cdot\sum_{t=1}^{T}\sum_{s\neq s_L}\left(\sum_{(u,v,w)\in W_h}q^{P,\pi_t}(u,v)P(w|u,v)q^{P,\pi_t}(s|w)\right)g_t(s)^2$$

$$\leq\theta\cdot L|S_{h+1}|\log(\iota)\sum_{t=1}^{T}\sum_{u\in S_h}\sum_{v\in A}\frac{q^{P,\pi_t}(u,v)}{\widehat{m}_{i(t)}(u,v)}+\frac{1}{\theta}\cdot\sum_{t=1}^{T}\sum_{s\neq s_L}q^{P,\pi_t}(s)g_t(s)^2$$

$$\leq\mathcal{O}\left(\theta\cdot L|S_{h+1}|\log(\iota)\left(|S_h||A|\log(T)+\log(\iota)\right)+\frac{1}{\theta}\cdot\sum_{t=1}^{T}\sum_{s\neq s_L}q^{P,\pi_t}(s)g_t(s)^2\right)$$

$$=\mathcal{O}\left(\theta L\cdot|S_h||S_{h+1}||A|\log^2(\iota)+\frac{1}{\theta}\cdot\sum_{t=1}^{T}\sum_{s\neq s_L}q^{P,\pi_t}(s)g_t(s)^2\right),$$

where the second step uses the fact that $\sqrt{xy}\leq x+y$ for all $x,y\geq 0$, the fourth step follows from Lemma D.5.3.

Then, for any layer $h=0,\ldots L-1$, picking the optimal $\theta$ gives

$$\sum_{t=1}^{T}\sum_{s\neq s_L}\sum_{(u,v,w)\in W_h} q^{P,\pi_t}(u,v)\sqrt{\frac{P(w|u,v)\log(\iota)}{\widehat{m}_{i(t)}(u,v)}}q^{P,\pi_t}(s|w)g_t(s)$$

$$=\mathcal{O}\left(\sqrt{L|S_h||S_{h+1}||A|\log^2(\iota)\sum_{t=1}^{T}\sum_{s\neq s_L}q^{P,\pi_t}(s)g_t(s)^2}\right)$$

$$\leq\mathcal{O}\left((|S_h|+|S_{h+1}|)\sqrt{L|A|\log^2(\iota)\sum_{t=1}^{T}\sum_{s\neq s_L}q^{P,\pi_t}(s)g_t(s)^2}\right).$$

Finally, taking the summation over all the layers yields

$$\sum_{h=0}^{L-1}\sum_{t=1}^{T}\sum_{s\neq s_L}\sum_{(u,v,w)\in W_h} q^{P,\pi_t}(u,v)\sqrt{\frac{P(w|u,v)\log(\iota)}{\widehat{m}_{i(t)}(u,v)}}q^{P,\pi_t}(s|w)g_t(s)$$

$$\leq \mathcal{O}\left(\sqrt{L|S|^2|A|\log^2(\iota)\sum_{t=1}^{T}\sum_{s\neq s_L} q^{P,\pi_t}(s)g_t(s)^2}\right),$$

which finishes the proof. □

