# OpenReview forum: "No-Regret Online Reinforcement Learning with Adversarial Losses and Transitions"
_NeurIPS.cc/2023/Conference — NeurIPS 2023 poster_

### Official Review · Reviewer_fP6g · 2023-06-08

**Soundness:** 3 good
**Presentation:** 2 fair
**Contribution:** 3 good
**Rating:** 5
**Confidence:** 1

**Summary:**

This paper introduces the challenge of online learning in adversarial MDPs where the loss functions and transition functions are chosen by a malicious adversary. Although previous algorithms achieving $O(\sqrt{T})$ regret with fixed transition functions could not handle adversarial transitions, in this paper, the authors propose a new algorithms that can handle both adversarial losses and transitions, with regret increasing smoothly based on the degree of maliciousness $C^P$. The first algorithm achieves $O(\sqrt{T} + C^P)$ regret, where $C^P$ measures how adversarial the transitions are and can be at most $O(T)$. Second, a black-box reduction approach is introduced to remove the requirement of knowing $C^P$. The algorithm is further refined to adapt to easier environments and achieve $O(U + \sqrt{U C^L} + C^P)$ regret, where $U$ is a gap-dependent coefficient and $C_L$ represents the amount of corruption on losses.

**Strengths:**

- A variant of the UOB-REPS algorithm is suggested to achieve regret in completely adversarial environments when the adversarial transition parameter, $C^P$, is known. This is accomplished by using an enlarged confidence set with the log-barrier regularizer and introducing a novel amortized bonus term.
- The requirement of knowing the adversarial transition parameter, $C^P$, is eliminated by proposing a black-box reduction approach. This approach provides the same guarantee (up to logarithmic factors) even if $C^P$ is unknown.
- The algorithms are further refined to simultaneously adapt to the maliciousness of the loss functions and achieve low regret. This enables the algorithms to handle various degrees of adversarial behavior in the loss functions while maintaining their regret performance.

**Weaknesses:**

- The organization of the paper may be making it difficult for me to comprehend. I think it is recommended to have the main algorithm presented in the main paper for better clarity. Since the paper covers multiple algorithms, it would be helpful to focus on explaining how they differ from existing methods, the challenges they address, and the novelty they bring to the field. Providing a dedicated conclusion section would enhance the understanding of the paper by summarizing the key findings and contributions.

**Questions:**

-

**Limitations:**

Given that this paper is primarily focused on theoretical aspects, its direct impact on society is expected to be neutral.

---

> ### Author Rebuttal · Authors · 2023-08-09
>
> Thanks for your helpful feedback. Please see our response below:
> ***
> **Q:**  Issues of paper's organization.
>
> **A:** Thanks for the suggestion. In the submission phase, we were unable to squeeze the algorithms and the conclusion section in the main texts because of the space limit. We will use the extra page granted in the final version to address these issues.

---

### Official Review · Reviewer_P2en · 2023-07-02

**Soundness:** 4 excellent
**Presentation:** 2 fair
**Contribution:** 4 excellent
**Rating:** 7
**Confidence:** 3

**Summary:**

This paper studies online reinforcement learning in tabular MDPs when the losses and transitions can be adversarially changing from round to round. They show that one can achieve regret guarantees which are $O(\sqrt{T} + C^P)$ where $C^P$ measures the degree to which transitions are changing.

Specifically they
- devise an algorithm which achieves the aforementioned guarantee when the value of $C^P$ is known.
- apply further reductions to get an algorithm which does not need to know the value of $C^P$.
- get gap-dependent regret bounds when the value of $C^P$ is known.

**Strengths:**

- The results seem very impressive. To my knowledge this is the first paper which studies adversarial MDPs with changing transitions. The authors quantify the notion of changing transitions and prove a regret bound.
- The algorithms and techniques seem very novel, and might be useful for future work.
- References and connections to previous work are stated clearly.

**Weaknesses:**

- The writing is a bit unclear, specifically regarding the technical details. This paper is a very technical paper with long proofs, but I do think it would be helpful to state the algorithms (or some simplified versions of the algorithms) in the main text and relegate some of the finer details to the appendix.
- Upon first reading, the proof ideas in Section 3 onwards did not make much sense, because they were trying to cover a lot of fine details, even though the overall proof sketch was not really discussed much. I'd encourage the authors to put more effort into improving readability.

Typos:
- Line 116: should this be $S$ instead of $X$?

**Questions:**

1. This paper considers transitions and losses which are specified ahead of time by an adversary. Do you think it is possible to consider adversarial losses/transitions which the adversary can suggest after seeing a history of the learner's decisions? (a harder setting)
2. What is the role of the splitting into epochs feature of the algorithm? What goes wrong if the algorithm updates the empirical transition in every round, as opposed to in every epoch?
3. Is there some intuition for why the algorithm uses the upper occupancy bound following line 161, as opposed to just using the occupancy measure associated with $\bar{P}$?
4. What is the message of Lemma 3.1? Why is the inequality $\sum_t C_t^P/u_t(s) \le \sum_t b_t(s)$ useful?
5. This might be a standard technique, but why are the costs in Algorithm 3 offset by a reward $r_\tau$? What does this capture?

---

> ### Author Rebuttal · Authors · 2023-08-09
>
> Thanks for your helpful feedback. Please see our responses below:
> ***
> **Q1:** Issues of writing, organization, and presentation.
>
> **A1:** Thanks for your suggestions. We will consider re-organizing the content in the future version.
>
> ***
> **Q2:** Line 116: should this be $S$ instead of $X$?
>
> **A2:** We thank the reviewer for spotting the typo. It will be fixed in the final version.
>
> ***
> **Q3:** Do you think it is possible to consider adversarial losses/transitions which the adversary can suggest after seeing a history of the learner's decisions? (a harder setting)
>
> **A3:** We do believe that our algorithm can handle the standard adaptive adversary (i.e., decides the transition/loss in round $t$ based on the history up to $t-1$). We consider the oblivious adversary in the paper just for simplicity. For a stronger adversary that can decide the transition/loss in round $t$ based on the action chosen in round $t$, and the total corruption is unknown, previous work by [He et al., 2022] has shown that it is impossible to achieve $O(C^P)$ regret.
>
> [He et al., 2022] Jiafan He, Dongruo Zhou, Tong Zhang, Quanquan Gu. Nearly Optimal Algorithms for Linear Contextual
> Bandits with Adversarial Corruptions. NeurIPS 2022.
>
> ***
> **Q4:** What is the role of the splitting into epochs feature of the algorithm? What goes wrong if the algorithm updates the empirical transition in every round, as opposed to in every epoch?
>
> **A4:** Since FTRL does not deal with varying decision sets easily (as mentioned in lines 328-329), splitting time steps into epochs guarantee that the occupancy measures in a epoch are from the same decision set.
> ***
> **Q5:** Is there some intuition for why the algorithm uses the upper occupancy bound following line 161, as opposed to just using the occupancy measure associated with $\bar{P}$?
>
> **A5:** It is important to ensure that ``optimism'' holds (i.e., $q^{P,\pi_t}(s,a) \leq \mu_t(s,a)$) in the analysis.
> ***
> **Q6:** What is the message of Lemma 3.1? Why is the inequality $\sum_t C^P_t/u_t(s) \leq \sum_t b_t$ useful?
>
> **A6:** Because of the transition corruption, we have a regret overhead of $\mathbb{E}[\sum_t  \sum_s q^{P,\mathring{\pi}}(s)\frac{C^P_t}{u_t(s)}]$ (Line 183) that can be prohibitively large. By incorporating the bonus term $b_t$ into the policy update, we get an additional regret term $\mathbb{E}[\sum_t \langle q^{P,\pi_t} - q^{P,\mathring{\pi}},b_t\rangle]$. The first part of Lemma 3.1 is to show that the overhead term $\mathbb{E}[\sum_t  \sum_s q^{P,\mathring{\pi}}(s)\frac{C^P_t}{u_t(s)}]$ can be cancelled by the negative part of the additional regret $\mathbb{E}[\sum_t \langle  q^{P,\mathring{\pi}},b_t\rangle]$. The second part of Lemma 3.1 is to show that the positive part of the additional regret $\sum_{t} \langle q^{P, \pi_t}, b_t\rangle\approx \sum_t \langle \widehat{q}_t, b_t\rangle$ can be bounded by the order of $C^P\log T$.
> ***
>
> **Q7:** Why are the costs in Algorithm 3 offset by a reward $r_\tau$? What does this capture?
>
> **A7:** Introducing a ``bonus term'' $r_\tau$ is a standard technique in the model selection literature where the goal is to use a meta-bandit algorithm to learn over a set of base-bandit algorithms, and try to perform as well as the best base-bandit algorithm running alone. This bonus technique appears in [Foster et al., 2020] and [Luo et al., 2022]. The reason to introduce the bonus is that when running the model selection algorithm, every base algorithm is only chosen and updated with a certain probability (because at each round, the meta algorithm can only select one of the base algorithms to execute), which results in base algorithm's performance degradation compared to the case when it's running alone. To compensates this phenomenon, for base algorithms that are chosen with a smaller probability, the meta algorithm will add a larger bonus to them.
>
> [Foster et al., 2020] Dylan J. Foster, Claudio Gentile, Mehryar Mohri, Julian Zimmert. Adapting to Misspecification in Contextual Bandits. 2020.
>
> [Luo et al., 2020] Haipeng Luo, Mengxiao Zhang, Peng Zhao, Zhi-Hua Zhou. Corralling a Larger Band of Bandits: A Case Study on Switching Regret for Linear Bandits. 2022.

---

> > ### Comment · Reviewer_P2en · 2023-08-14
> > **Thanks**
> >
> > Thank you for your detailed answers. I believe this is a good work and I will keep my score.

---

### Official Review · Reviewer_7UE7 · 2023-07-05

**Soundness:** 3 good
**Presentation:** 4 excellent
**Contribution:** 3 good
**Rating:** 7
**Confidence:** 4

**Summary:**

The authors consider no-regret learning in adversarial MDPs, when the dynamics may change adversarially across episodes. They design an algorithm which provides a regret guarantee of $\tilde{O}(\sqrt{T} + C)$ where $C$ is the total deviation from some fixed transition function, and the benchmark is the best fixed Markov policy in hindsight. The authors show that this result can be obtained even when $C$ is unknown to the algorithm, by constructing a black-box reduction from an algorithm which knows $C$. The authors also consider the stochastically constrained adversarial setting and provide a gap-dependent regret guarantee, but in this setting knowing $C$ is required.

**Strengths:**

* The results established in the paper improve upon previous results in corruption-robust reinforcement learning, and in particular constitute the first $\tilde{O}(\sqrt{T} + C)$-type bounds even when the losses are adversarial.
* The techniques presented in the paper seem novel and interesting, and may be of independent interest in various RL scenarios.
* The authors provide an overview of the analysis presenting the main challenges and ideas, making it easier to understand their main contributions and high-level techniques.
* The black-box reduction from an algorithm which knows $C$ to one which doesn't need to know $C$ seems particularly interesting to me, and may be of interest in other non-stationary scenarios when trying to design algorithm that adapt to some corruption budget.

**Weaknesses:**

* Since the dynamics of the MDP changes adversarially across episodes, it makes less sense to set the benchmark as a Markov policy, as it is no longer the case that the optimal policy on a sequence of MDPs is WLOG Markov. The authors do not address this point in the current version of the paper, and I would like to hear from them why this assumption is needed and whether or not competing against a more general benchmark policy is hard.
* At first, it may seem that $C$ measures the total variation of the transition dynamics across time, as is the case in other non-stationary RL formulations. However, the authors define $C$ to be the sum of variations from a single fixed transition function. This is a weaker definition, and in particular the results presented in this paper do not include a setup where the dynamics have a drift of $1/ \sqrt{T}$ per episode, and for every fixed transition $P'$ the total deviation from it would be $\Omega(T)$. In such a setting, the authors' bounds would be meaningless even though obtaining sublinear regret (even dynamic regret) is indeed possible.

**Questions:**

See my remarks above under "Weaknesses" - I'd appreciate it if the authors could address the limited generality of the benchmark policy, as well as the problem of handling drift in the transitions.

**Limitations:**

Yes

---

> ### Author Rebuttal · Authors · 2023-08-09
>
> Thanks for your valuable feedback. Please see our responses below:
>
> ***
> **Q1:**
>  Since the dynamics of the MDP changes adversarially across episodes, it makes less sense to set the benchmark as a Markov policy, as it is no longer the case that the optimal policy on a sequence of MDPs is WLOG Markov. The authors do not address this point in the current version of the paper, and I would like to hear from them why this assumption is needed and whether or not competing against a more general benchmark policy is hard.
>
> **A1:** This is indeed a very good point. While we do not have a definite answer regarding how hard it is to compete with the best non-Markovian policy generally, we remark that in the case when corruption is ``binary'' (that is, an episode is either corrupted or not so $C^P$ counts the number of corrupted episodes), the best non-Markovian policy would not perform better than the best Markovian one on those uncorrupted episodes, and since we already have $\sqrt{T}+C^P$ regret against the best Markovian policy, the same bound would thus hold for competing against the best non-Markovian one as well. We also emphasize again that our regret notion, though restricted to competing with Markov policies only, is more general than the standard one used in the literature of learning corrupted MDPs, as we argue in L73-L78.
>
> ***
> **Q2:**  At first, it may seem that
>  measures the total variation of the transition dynamics across time, as is the case in other non-stationary RL formulations. However, the authors define $C$
>  to be the sum of variations from a single fixed transition function.
>
> **Q3:**  In particular, the results presented in this paper do not include a setup where the dynamics have a drift of $1/\sqrt{T}$ per episode, and for every fixed transition $P'$ the total deviation from it would be $\Omega(T)$. In such a setting, the authors' bounds would be meaningless even though obtaining sublinear regret (even dynamic regret) is indeed possible.
>
> **A2 and A3:** (For Q2 and Q3) We agree that the amount of variation $V^P\triangleq \sum_{t=2}^T \|P_t-P_{t-1}\|$ can be much smaller than our $C^P$. It is an interesting question whether the regret (compared to a fixed policy) can be just linear in $V^P$ instead of $C^P$. Previous work in non-staitonary RL shows that a dynamic regret of $(V^P)^{1/3}T^{2/3}$ is tight, which is incomparable to our static regret bound of $C^P$. However, there do exist examples (as the reviewer point out) where $(V^P)^{1/3}T^{2/3}$ is much smaller than $C^P$, so getting the best of both is also an interesting direction.

---

> > ### Comment · Reviewer_7UE7 · 2023-08-13
> >
> > I thank the authors for their detailed response.
> >
> > Q1: The fact that if the corruption is binary then competing against the best fixed Markov policy is sufficient does alleviate some of my concerns regarding the benchmark.
> >
> > I have no further questions about the paper.

---

### Official Review · Reviewer_tv4V · 2023-07-06

**Soundness:** 3 good
**Presentation:** 4 excellent
**Contribution:** 3 good
**Rating:** 7
**Confidence:** 3

**Summary:**

This paper studies the problem of reinforcement learning under adversarial reward and transitions. When evaluating the regret against the best fixed policy in hindsight, the algorithm proposed in this paper achieves the optimal regret O(\sqrt{T} + C^P), which is followed by other favorable extensions including being agnostic to the corruption amount $C^P$ and $C^L$, as well as gap-dependent regret bounds.

**Strengths:**

The theoretical results presented in this paper is solid. Several new techniques, including a new model selection framework that allows adversarial environments, is of independent interestis.

**Weaknesses:**

The main weakness is the lack of justification for the specific type of regret studied in this paper. To the best of my knowledge, this is the only paper that studies the regret against the best policy in hindsight in a mild-corruption setting, i.e. C^P \leq o(T), so some words justifying this new notion would be helpful. I'm not sure when such a regret notion is desirable to study in this scenario, since if C^P \leq o(T), the best policy in hindsight will converge to the optimal policy in the underlying uncorrupted MDP, so the two notions would converge anyways.

Minor:

There is a missing reference for corruption-robust RL

1. Zhang, X., Chen, Y., Zhu, X., & Sun, W. (2021, July). Robust policy gradient against strong data corruption. In International Conference on Machine Learning (pp. 12391-12401). PMLR.

**Questions:**

See above.

---

> ### Author Rebuttal · Authors · 2023-08-09
>
> Thanks for your helpful feedback. Please see our responses below:
> ***
> **Q1:** Lack of justification for the specific type of regret studied in this paper.
>
> **A1:** First, when the MDPs are time-varying, the ``underlying uncorrupted MDP'' is not always well-defined. On the other hand, the best-in-hindsight policy is always well-defined, so adopting it as the benchmark is more direct and requires no additional assumptions. Second, our regret bound that scales with $\sqrt{C^L}$ reflects that our algorithm is still doing a meaningful job (i.e., performing as well as the best fixed policy) even when $C^L=\Omega(T)$ and $C^P=o(T)$. In contrast, the regret notion in [Lykouris et al, 2019] and [Wei et al., 2022] will become vacuous in this scenario. Thus, our notion of regret can more clearly decouple and distinguish the hardness coming from adversarial loss and from adversarial transition. Lastly, using best-in-hindsight policy as the benchmark is a convention in the adversarial online learning literature. As mentioned in the paper (L73-L78), our regret notion always implies the other one, but not the other way around. Therefore, we never fail to capture scenarios that are captured in the other notion of regret.
>
> ***
> **Q2:** A missing reference [Zhang et al., 2021] for corruption-robust RL.
>
> **A2:** We thank the reviewer for highlighting this work. We will make sure to include it in the final version of this manuscript and have a more complete overview of the literature.
>
> [Zhang et al., 2021] Zhang, X., Chen, Y., Zhu, X., Sun, W. (2021, July). Robust policy gradient against strong data corruption. In International Conference on Machine Learning (pp. 12391-12401). PMLR.

---

> > ### Comment · Reviewer_tv4V · 2023-08-10
> > **Thank you for addressing my questions.**
> >
> > I have no more questions about the paper

---

### Official Review · Reviewer_ME4k · 2023-07-06

**Soundness:** 3 good
**Presentation:** 3 good
**Contribution:** 3 good
**Rating:** 7
**Confidence:** 3

**Summary:**

This paper studies learning algorithms for adversarial MDP with adversarial transition functions. The authors developed an algorithm that enjoys O(\sqrt{T} + C^P) regret where C^P measures how adversarial the transition functions are. The developed algorithm could work without knowing C^P. Finally, the authors show that further refinements of the algorithm can adapt to easier environments.

**Strengths:**

This paper tackles an important question of how to learn adversarial MDPs with adversarial transition functions. Prior work shows that learning with adversarial transition is information-theoretically impossible without paying exponential dependence on the episode length. This paper provides the first algorithm, based on a modification of UOB-REPS, which allows the regret bound to degrade mildly upon the amount of corruption. Furthermore, it also resolves the question of an unknown amount of adversarial transition and considers the question of adaptivity.

**Weaknesses:**

NA

**Questions:**

NA

---

> ### Author Rebuttal · Authors · 2023-08-09
>
> Thanks for your positive comments.

---

### Decision · Program_Chairs · 2023-09-21

**Decision:**

Accept (poster)

**Comment:**

This paper addresses the challenging problem of online learning in adversarial Markov Decision Processes (MDPs), where both loss and transition functions can be manipulated by a malicious adversary. It introduces novel algorithms that demonstrate remarkable adaptability to varying degrees of adversarial behavior, offering smooth regret bounds as a function of the maliciousness level. While the paper's organization and presentation have room for improvement, the significant contributions in tackling adversarial MDPs and the authors' commitment to addressing these issues in the final version warrant accepting the paper. Overall, this work advances the field of online learning in adversarial environments, making it a valuable addition to the conference.